# Analytical and Structural Tools of Lipid Hydroperoxides: Present State and Future Perspectives

**DOI:** 10.3390/molecules27072139

**Published:** 2022-03-25

**Authors:** Vassiliki G. Kontogianni, Ioannis P. Gerothanassis

**Affiliations:** 1Section of Organic Chemistry and Biochemistry, Department of Chemistry, University of Ioannina, GR-45110 Ioannina, Greece; 2International Center for Chemical and Biological Sciences, H.E.J. Research Institute of Chemistry, University of Karachi, Karachi 75270, Pakistan

**Keywords:** lipids, hydroperoxides, HPLC, MS, NMR, IR, Raman, fluorescence

## Abstract

Mono- and polyunsaturated lipids are particularly susceptible to peroxidation, which results in the formation of lipid hydroperoxides (LOOHs) as primary nonradical-reaction products. LOOHs may undergo degradation to various products that have been implicated in vital biological reactions, and thus in the pathogenesis of various diseases. The structure elucidation and qualitative and quantitative analysis of lipid hydroperoxides are therefore of great importance. The objectives of the present review are to provide a critical analysis of various methods that have been widely applied, and more specifically on volumetric methods, applications of UV-visible, infrared, Raman/surface-enhanced Raman, fluorescence and chemiluminescence spectroscopies, chromatographic methods, hyphenated MS techniques, NMR and chromatographic methods, NMR spectroscopy in mixture analysis, structural investigations based on quantum chemical calculations of NMR parameters, applications in living cells, and metabolomics. Emphasis will be given to analytical and structural methods that can contribute significantly to the molecular basis of the chemical process involved in the formation of lipid hydroperoxides without the need for the isolation of the individual components. Furthermore, future developments in the field will be discussed.

## 1. Introduction

Lipids are a diverse group of biomolecules of great importance in nutrition and health due to their functional roles as constituents of cellular membranes, energy sources in metabolic processes, and as signaling molecules [1,2,3]. Mono- and polyunsaturated lipids are particularly susceptible to free-radical process, known as lipid peroxidation. This critical molecular mechanism, which occurs widely both in plants and animals, results in the formation of hydroperoxides (ROOH, R is any organic group) as primary nonradical-reaction products [4,5,6,7]. Lipid peroxidation is implicated in damage of alimentary oils and fats, due to the development of off-flavors, degradation of nutrients and bioactives and the formation of potentiality toxic compounds [4,5,6,7,8,9,10]. In living organisms, lipids also undergo oxidation during normal aerobic metabolism or due to the effect of oxidizing agents, which results in the disruption of intracellular membranes and in the pathogenesis of neurodegenerative, gastric, and nutritional diseases [7,8,9].

One of the mechanisms that have been advanced to explain lipid peroxidation is the free-radical oxidation of the allylic C-H bond adjacent to a methylene group. This bond is relatively weak, especially in the case of bis-allylic hydrogens of polyunsaturated fatty acids, and thus the hydrogen is more susceptible to abstraction (Table 1). For methyl oleate, for example, hydrogen abstraction involves the allylic C-8 and C-11 (Figure 1). The unpaired electron on the carbon results in a carbon-centered radical which is stabilized by a molecular rearrangement. An oxygen attack on these intermediates produces a mixture of eight allylic hydroperoxides at C-8, C-10, C-9, and C-11. The relative amounts of the various isomers depend upon the conditions of oxidation and kinetic and thermodynamic factors [4,5]. An alternative reaction mechanism involves the critical role of the peroxyl-radical intermediate rather than the direct-radical isomerism, on the basis of experiments with α-tocopherol [6]. The O-O bond cleavage is thermodynamically more favored than the O-H bond (Table 1); therefore, hydroperoxides are more likely decomposed through cleavage of the O-O bond to form alkoxyl and hydroxyl radicals (Figure 2). Carbon–carbon cleavages (ii) and (iii) result in the formation of aldehyde esters and aldehydes [4]. Carbon–oxygen cleavage results in positional isomerism of unsaturated hydroperoxides. Further decomposition products include ketones, epoxides, alcohols, hydroxy compounds, oligomers, and polymers.

Hydroperoxides of unsaturated fatty acids and esters can also be produced with photooxidation in the presence of oxygen, light, and photosensitizer, such as chlorophyl, riboflavin, and hemoproteins [4,5,6,7,8,9,10,11]. The singlet-excited-state sensitizer, ^1^S_sens_, is converted to the triplet state, ^3^S_sens_, by absorption of light. ^3^S_sens_ has a longer lifetime and reacts with the lipid substrate by hydrogen atom or electron transfer to form radicals, which react with oxygen to produce hydroperoxides. In an alternative reaction mechanism, ^3^S_sens_ interacts with oxygen by energy transfer to produce nonradical singlet oxygen, ^1^O_2_. This highly reactive species interacts with unsaturated lipids by a concerted “ene” addition mechanism (Figure 3). The biological production of reactive species such as H_2_O_2_ and peroxide anion ($O_{2}^{-}$) can induce peroxidation effects in nucleic acids, proteins, and lipids. Lipid peroxidation has damaging effects on cell membranes since they are composed of polyunsaturated fatty acids, which are primary targets of reactive oxygen species (Figure 4). Lipid peroxidation can be enzymatic and nonenzymatic. Phospholipid hydroperoxides can be reduced to less-reactive hydroxides by phospholipid glutathione peroxidase (PHGPx) or hydrolyzed by phospholipase A2 (PLA2) releasing fatty-acid hydroperoxides (LOOH) [9]. These hydroperoxides can be reduced by glutathione peroxidases (GPx) or decomposed by metal ions, ONOO^−^ or HOCl, generating fatty-acid peroxyl (LOO**·**) and/or alcoxyl radicals (LO**·**). These radicals, by the Russell mechanism, can lead to the formation of lipid-fragmentation products, propagation of lipid peroxidation, and generation of ^1^O_2_ (Figure 5). Several of those secondary lipid-peroxidation products can be cytotoxic and result in oxidative damage and apoptosis.

Numerous mechanistic studies have been published on hydroperoxide formation of linoleate, conjugated linoleic acids (CLA), linolenate, arachidonate, *cis*-5,8,11,14,17-eicosapentaenoic acid (EPA), *cis*-4,7,10,13,16,19-docosahexaenoic acid (DHA), cholesterol, triacylglycerols, and phospholipids. Although many mechanistic questions remain to be answered, it may be concluded that the product distribution obtained depends on competition of hydrogen abstraction, β-scission, and cyclization pathways [5,6,10,11,12].

Over the past five decades, much emphasis has been given in the development of various analytical methods which can provide information on the level of oxidation in foods and biological samples by determining primary oxidation products, such as hydroperoxides, and/or secondary decomposition products, such as aldehydes, ketones, epoxides, etc. [7,8,9,10,11,12,13]. Nevertheless, several of the proposed methods are of limited specificity and precision and can provide only a crude indication of the molecular species involved in the process of lipid peroxidation. The basic aim of the review is to provide a critical evaluation of various structural and analytical methods that can be used in investigating lipid hydroperoxides (LOOH) since they are of primary importance in elucidating mechanisms of lipid peroxidation, effects of temperature, medium, and heterogeneous systems, the role of initiators and in clarifying the structures of decomposition products. More specifically, we will examine volumetric methods, applications of UV-visible (UV-Vis), infrared (IR), Raman/surface-enhanced Raman, fluorescence and chemiluminescence (CL) spectroscopies, chromatographic methods, hyphenated mass-spectrometry (MS) techniques, nuclear magnetic resonance (NMR) and chromatographic methods, NMR spectroscopy in mixture analysis, structural investigations based on quantum chemical calculations of NMR parameters, applications in living cells, and metabolomics. Much emphasis will be given to those analytical and structural methods which can contribute significantly to the molecular basis of the chemical processes involved in the formation of lipid hydrogen peroxides, without the need for the isolation of the individual components.

## 2. Volumetric Methods

Among the various methods employed, iodometry has been the most common and widely used method for the analysis of peroxides, owing to the simplicity of the experimental procedure. Although lipid extraction is required prior to the procedure, the method is rapid. In acidic medium, hydroperoxides and other peroxides react with the iodide ion to generate iodine, which is titrated using a sodium thiosulfate solution, in the presence of starch. Several steps are included in the procedure: precise weighing of the sample, dissolution in chloroform, acidification with acetic acid, potassium iodide addition, incubation for exactly 5 min, and titration with sodium thiosulfate.
$$\mathrm{ROOH}+2I^{-}+H_{2}O\;\to \mathrm{ROH}+2{\mathrm{OH}}^{-}+I_{2}$$
$$2S_{2}O_{3}^{2-}+{\;I}_{2}\;\to {\;S}_{4}O_{6}^{2-}+2I^{-}$$

There are two iodometric-titration methods, the AOAC Official Method [14] and the AOCS Official Method [15]. Peroxide value (PV) is defined as the amount of active oxygen (in meq) contained in 1 kg of lipid that can oxidize potassium iodide according to the above methods. It can alternatively be expressed as millimoles of active oxygen per kilogram of lipids by dividing it by two (SI units). For peroxide values below 10, the typical iodometric method requires a sample of 5 g, and roughly 1 g for higher peroxide values.

The PV test, although empirical, is simple, reliable, and reproducible when performed under standardized conditions. However, it has certain disadvantages, mostly due to high susceptibility of iodine to oxidation in the presence of molecular oxygen, which is amplified by light exposure. Moreover, spontaneous hydroperoxide formation can occur, resulting in iodine absorption by unsaturated fatty acids [16]. It also necessitates anhydrous systems to eliminate interference issues; therefore, lipids must be extracted, and this stage of the treatment increases oxygen interaction. Furthermore, because peroxides are generally further deteriorated, the peroxide value determination does not provide a true indication of oxidative degradation, hence simultaneous analysis of secondary products is recommended. Volumetric methods have issues with the high quantity of lipids required, the efficacy of the lipid-oxidation products’ interaction with the various reagents, and the susceptibility of these reactions to temperature, pH, solvents, oxygen, and other coexisting chemicals [17]. Nevertheless, the main drawback is the low accuracy, since higher PV values result in higher uncertainties of the titration endpoint. To overcome its main limitations, several alternative iodometric methods, based on the colorimetric detection of the triiodide chromophore [18,19] and on the endpoint potentiometric detection [20] have been proposed. 

## 3. UV-Visible (UV-Vis) Spectroscopy

Lipid hydroperoxides, under acidic conditions, can react quantitatively with potassium iodide to form iodine, which is then combined with starch to form a blue complex that can be measured at 550 nm. The accuracy, as with that of titration methods, is influenced by a number of experimental parameters such as reaction time, temperature, and light. The liberated iodine can react with highly unsaturated lipids, giving false results. To overcome this problem, excess of potassium iodide is allowed to react with the liberated iodine to form a triiodide anion, which can be measured at 353 nm [21]. 

For determination of peroxide content, the ferrous oxidation is an alternative method that is simpler to use than iodometry. The method involves the oxidation of Fe(II) to Fe(III), driven by hydroperoxide reduction in acidic conditions and in the presence of either thiocyanate or xylenol orange (FOX). The thiocyanate method, which is based on the International Dairy Federation’s Standard Method [22], has been utilized in milk-based products [23,24,25], fats and oils [26], and food-lipid extracts [27,28]. The complexes with the ferric ion have maximum absorbance peaks at 500 and 560 nm, respectively, which can be detected using a UV-Vis spectrophotometer. A red complex is formed when ferric salts react with potassium thiocyanate, and a Fe(^3+^)-xylenol orange complex is formed when ferric ions oxidize xylenol orange (FOX). The ferric thiocyanate method is more sensitive than the iodometric method and requires a smaller sample quantity (approximately 0.1 g). The values obtained by the ferric thiocyanate method, on the other hand, are 1.5 to 2 times higher than those by the iodometric method. The thiocyanate method uses a lot of solvent, and unlike the FOX, it only detects a narrow range of peroxide concentrations, and the molar absorptivity of the ferric xylenol orange complex changes depending on the synthetic methodology of the dye. The advantages, such as the high specificity of the reaction for hydroperoxides, as well as the disadvantages, such as spectrum interference by particular pigments, are rather similar for both methods [29].

The FOX method is cheap, rapid, and unaffected by oxygen or light. There is good agreement and precision between the FOX assay and the iodometric methodology; it also quantifies hydroperoxide levels as low as 0.1 meq O_2_/kg food [30]. However, the main advantage of the FOX method over the traditional iodometry is its wide applicability, since it has been employed in biological samples, fats and oils, lipid extracts, and homogenates from meat and vegetables. The main disadvantage of the FOX method, on the other hand, appears to be its limited linear range and low reproducibility, as it is influenced by several variables. Some of these factors are the type of sample, the extraction and/or purification steps, the presence of oxidizing/reducing agents in the samples, the contamination with pro-oxidant metals, and the strong dependency of the method on the pH values [30]. 

Another colorimetric method for measuring hydroperoxides in photo-oxidized oils utilizes cadmium acetate, and the absorbance is measured at 350 nm [31]. Hydroperoxide concentrations are calibrated using a standard solution of benzyl peroxide in ethanol. Aside from the ferric thiocyanate and FOX methods, the ferric ions can react and form colored complexes with N,N-dimethyl-p-phenylene diamine or N,N′-di(2-naphthyl)phenylene-1,4-diamine [17]. A modified ferrous oxidation-xylenol orange method, or FOX2, allows low hydroperoxide-concentration detection in the presence of nonperoxidated lipids [32]. 

## 4. Infrared (IR) Spectroscopy

Τhe distinctive –OO-H stretching vibration band of hydroperoxide moieties between 3600 and 3400 cm^−1^ [33] and the two overtones at 1417–1500 nm and 954–1011 nm [34], can be determined using Fourier transform IR (FT-IR) (Figure 6). The near-IR (NIR) spectral band at 1350–1480 nm, with a single-point baseline at 1514 nm, was shown to be a better criterion for predicting the peroxide value (PV). FT-IR spectroscopy has been effectively used to quantify hydroperoxides in neat fats and oils as an alternative to standard methods, conferring the advantages of analytical speed and automation [35]. A further improvement for predicting PV values employs triphenylphosphine (TPP), which reacts stoichiometrically with hydroperoxides to produce triphenylphosphine oxide (TPPO). Accurate quantitation of the TPPO can be achieved by measurement of its intense absorption band at 542 cm^−1^, which provides a simple means of determining PV. The FTIR method is a significant improvement over the standard AOCS method in terms of analytical time and effort; it avoids solvent and reagent disposal problems and is especially suited for routine quality-control applications in the industry of fats and oils (Figure 7) [36].

Li et al. [37] improved the FT-NIR method, originally designed to determine the PV values of triacylglycerols at levels of 10–100 PV, to allow for the analysis of PV between 0 and 10 PV; this range of PV values is of interest in the industry of edible oils. However, it is necessary to define the spectral changes that occur as oil oxidation progresses, assign wavelengths to more common molecular species formed, and investigate potential spectral interferences. Ruiz et al. [38] developed an automated, rapid, and highly precise method for determination of the PV in edible oils based on a continuous flow system and FT-IR spectroscopic detection. The sample stream was mixed with a solvent mixture consisting of 25% (*v*/*v*) toluene in hexanol which contained TPP. IR has been used to evaluate the PV in oxidized lipids [39], and to characterize the ageing of various edible oils [40].

Mahboubifar et al. [41] surveyed the long thermal oxidative kinetic and stability of four different edible oils with the use of attenuated total reflectance- FT-IR (ATR–FTIR) combined with multivariate-curve resolution alternative least-squares (MCR-ALS) method. Pizarro et al. [42] used Fourier transform mid-infrared (FT-MIR) spectroscopy combined with a variable selection method (the stepwise orthogonalization technique) coupled with multivariate calibration methods and preprocessing tools in order to develop models for the prediction of peroxide value in extra-virgin olive oils.

## 5. Raman Spectroscopy/Surface-Enhanced Raman Spectroscopy (SERS)

Raman spectral changes in several edible oils, taking place during thermally induced lipid oxidation, were investigated originally by Muik et al. [43]. However, the temperature used in their study mainly produced secondary oxidation products and only transient hydroperoxides, which rapidly decompose above 100 °C. Therefore, no correlation between the peroxide values and frequencies in the Raman spectra was reported. Guzman et al. [44] evaluated the combined use of a portable low-resolution Raman spectrometer, with some chemometric tools for determining quality parameters related to the oxidative state, such as the peroxide values, in virgin olive oil. Surface-enhanced Raman spectroscopy (SERS) was successfully applied to discriminate edible oil type, oxidation, and adulteration through sample pretreatment [45]. SERS, on the other hand, did not verify the change in a carbon–carbon double bond during the oxidation of edible oil. A SERS analyzer based on plasmonic metal liquid-like platform (PML-SERS) was developed for the rapid analysis and fingerprint identification of peak changes in oil samples. The peaks at 1265 and 1436 cm^−1^ which were assigned to the C-H bending and C-H bending (scissoring) vibrations, respectively, could be clearly discriminated. The characteristic peak of 1265 cm^−1^ declined steadily with the increase in oxidation degree, while the relatively stable peak of 1436 cm^−1^ was used as an internal standard to generate relative peak intensity I_1265/1436_ for quantitative analysis of the lipid-oxidation degree. The relative Raman intensity, I_1265/1436_, has a good correlation with peroxide value, which is used for quantitative detection [46]. Ottaway et al. [47] evaluated the use of near- and mid-infrared absorption and Raman-scattering spectroscopies on oil samples from different oil classes, including seasonal and vendor variations, to evaluate which measurement technique or combination thereof is optimum for predicting peroxide values. NIR was shown to be the most rugged to interclass variations associated with the collected spectra. Both MIR and NIR performed comparably for determining PV across three classes of olive oils, but an olive-oil-specific model could not reliably extrapolate PV values to non-olive-oil samples.

## 6. Fluorescence Spectroscopy

Akasaka et al. [48] reported the use of diphenyl-pyrenylphosphine (DPPP), that is a very sensitive and selective reagent for the determination of lipid hydroperoxides. This reagent is nonfluorescent; however, its oxide, which is produced by oxidation with lipid hydroperoxides, is highly fluorescent. Later, Miyazawa et al. [49] developed a fluorimetric assay for lipid hydroperoxides using pea peroxygenase, a hydroperoxide-dependent hydroxylase. In the reaction catalyzed by this enzyme, the hydroperoxide is converted to the alcohol and the substrate, 1-methyl indole, is hydroxylated. The above methodology allowed the determination of total hydroperoxides in homogenates of meat and fish products without a previous extraction of lipids (Figure 8). 

Akasaka et al. [50,51] used again the reagent DPPP to measure lipid hydroperoxides by both batch and HPLC postcolumn methods, then a year later reported a highly sensitive flow-injection analysis of lipid hydroperoxides using DPPP. Recently, Cropotova and Rustad [52] proposed modification of the DPPP-fluorimetric assay by the use of fluorescence microscopy as a rapid, noninvasive and cost-efficient method aiming not only to quantify lipid and protein hydroperoxides in muscle foods (fish samples during frozen storage) but also to visualize them by fluorescence imaging. 

## 7. Chemiluminescence (CL) Spectroscopy

Ultra-weak chemiluminescence (CL) is accompanied during oxidation of hydrocarbons and lipids, so the use of a light amplifier such as luminol (5-amino-2,3-dihydro-1,4-phthalazinedione) is required. The luminol-enhanced chemiluminescence involves oxidation of luminol with hydroperoxides in basic solution to form a luminol-derived product in excited state which relaxes to ground state, emitting strong blue light at 430 nm. Bunting and Gray [53] developed an automated flow-injection chemiluminescence system for measuring lipid-hydroperoxide concentrations in vegetable oils, with luminal and cytochrome c as the catalyst for the reaction. A luminol-enhanced CL method was also applied to directly measure the amount of hydroperoxides in lipid emulsions during oxidation, although it was not specific for fatty acids and lipid hydroperoxides (Figure 9) [54]. Szterk and Lewicki developed a method with CL reaction in a nonaqueous medium without the use of a catalyst [55]. The reaction in CL was based on lipid-hydroperoxide decomposition with HO^−^ ions in a DMF/KOH/oil mixture and with acridine in addition. Finally, for the routine clinical analysis of the total amount of lipid hydroperoxide in small volumes of biological fluids, a CL method was selected using a system based on microperoxidase and isoluminol [56].

## 8. Chromatographic Methods

One of the major disadvantages of the methods analyzed above is the limited specificity due to interference from minor compounds other than hydroperoxides. As a result of this constraint, several chromatographic methods for the separation and isolation of hydroperoxides have been developed in an effort to provide information on specific hydroperoxide structures. To separate and quantitatively analyze hydroperoxides and secondary products of lipid oxidation, adsorption (solid–liquid) and normal-phase liquid partition (liquid–liquid) column chromatography are commonly utilized. To purify and analyze the hydroperoxides of fatty acids and esters, as well as dimeric and polymeric products in vegetable oils, researchers combined adsorption and partition chromatography using a silicic-acid column with methanol as a partial “stationary” phase. Solid-phase extraction (SPE) columns became popular among lipid analysts when they became commercially available, with a variety of polar and nonpolar adsorbents having different functionality by bonding to the silanol groups on the surface of silica gel. A rapid and easy chromatographic process using a silica cartridge was used, for example, to separate pure hydroperoxides from autoxidized methyl linoleate, methyl linolenate, and hydroperoxy epidioxides from methyl linolenate in 20–40 mg quantities [10].

### 8.1. High-Performance Liquid Chromatography (HPLC)

When advanced HPLC systems become available, adsorption chromatography with a microporous 5 μm silica-acid column was used to partially separate four *cis*,*trans* and *trans*,*trans* isomeric 9- and 13-hydroperoxides from autoxidized methyl linoleate [10]. Preparative reversed-phase HPLC was used to separate autoxidized trilinolein into mono-, bis-, and tris-hydroperoxides using a 5 μm C-18 column with UV at 235 nm and refractive-index detectors. By using normal-phase HPLC on a 5 μm silica column with UV detection, the mono-hydroperoxides of trilinolein were further separated into various positional isomers [10].

HPLC has been frequently used for the analysis of a wide range of hydroperoxides (with different characteristics of volatility, molecular weight, or polarity) of fatty acids [57], triacylglycerols [58,59], and fatty-acid methyl esters [60]. However, sample preparation can be time-consuming and often necessitates prior lipid extraction. HPLC operates at room temperature, which is one of its major advantages over GC. This also reduces the possibility of artifact formation and permits applications without the need for prior derivatization. Quantitative analysis, on the other hand, is difficult to achieve due to the lack of a universal detector. Various lipid hydroperoxides can be separated using normal-phase or reversed-phase HPLC coupled to UV, post-column derivatization-UV/visible, chemiluminescence, electrochemical, or mass-spectrometry detectors. UV detection at 230–235 nm was employed by Park et al. [61] for the analysis of hydroperoxides formed by autoxidation of vegetable oils. UV and chemiluminescence (CL) detectors were found to be more sensitive and selective than the evaporative light-scattering (ELS) detector for measuring hydroperoxides from trilinolein at peroxide values ranging between 3 and 8 [58]. At higher peroxide values (22–63), detection by CL and ELS was more sensitive than by UV [58]. A fast and sensitive HPLC method for the simultaneous determination of cholesterol hydroperoxides and other major oxysterols, using two different detection systems (ultraviolet at 210 nm and light scattering) was also described [62]. Liquid chromatography can also be used to determine specific hydroperoxides generated from sterols. Säynäjoki et al. used a normal-phase column and two types of detectors (UV and fluorescence) to determine stigmasterol hydroperoxides [63]. Gotoh et al. [64] developed a method for measuring the peroxide value in colored lipids on the basis of the reaction with triphenylphosphine, forming a compound which absorbs at 260 nm. The samples then underwent HPLC separation and UV detection. An electrochemical detection of triacylglycerol hydroperoxides in oxidative deterioration of some vegetable oils was also developed [65].

The utilization of postcolumn reactions has been proven to be the most interesting approach for increasing the sensitivity and specificity of hydroperoxide analysis. Separation techniques such as HPLC avoid most of the interference compounds that affect several methods, especially those based on chemilumiscence or fluorescence reactions; this allows detection of hydroperoxides in complex systems at picomole levels. HPLC with postcolumn CL detection allows very sensitive and direct hydroperoxide analysis. By this technique, the HPLC effluent is mixed with different CL cocktails at a postcolumn tee and is monitored by a CL detector measuring light emitted by the reaction of hydroperoxides with a heme protein (cytochrome c or peroxidase) and an oxidized dye (luminol or isoluminol) [57,58,60]. Yang et al. [66] developed an improved hydroperoxide assay involving chromatographic separation by HPLC, followed by postcolumn reaction with I_2_ or luminal. Miyazawa et al. [67] used CL-HPLC equipped with a reversed-phase column to analyze mono-, bis-, and tris-hydroperoxides of triacylglycerols formed during autoxidation and photosensitized oxidation of vegetable oils (Figure 10). Using a cation-exchange gel column, a CL-HPLC method for determining H_2_O_2_ at picomolar levels was developed [68]. Christensen and Holmer developed two HPLC methods for measuring hydroperoxides based on a postcolumn CL detection to track the time-dependent changes in lipid peroxidation in different lipid classes in butter and dairy spreads [69]. The high sensitivity of the CL-HPLC method provides an excellent means for studying the peroxidation mechanism and flavor changes in vegetable oils.

The utilization of postcolumn fluorescence detection includes the reduction of hydroperoxides with DPPP and formation of the arylphosphine oxide, which is strongly fluorescent. Hydroperoxides of triacylglycerols and cholesterol esters were selectively determined, at picomole levels, with a fluorescence detector by postcolumn reaction with DPPP by Akasaka et al. [70]. Moreover, an HPLC-fluorescence method using postcolumn detection was effectively applied in the analyses of hydroperoxide mixtures containing conjugated and nonconjugated diene structures [71]. The simultaneous determination of different classes of lipid hydroperoxides by HPLC with post-column detection by a ferrous xylenol orange reagent (FOX) was also reported [72]. 

### 8.2. High-Performance Size-Exclusion (HPSEC) Spectroscopy

HPSEC separations can be used to investigate the oxidation of bulk material or on fractions generated by adsorption chromatography of fatty-acid methyl esters and triacylglycerols. The simplest approach is to separate nonpolar from polar lipids, which allows alkyl-chain dimers/polymers to be distinguished from oxygenated dimers/polymers using alkoxyl or peroxyl radicals. An important application of HPSEC includes analyses of dimers, trimers, oligomers, partial glycerides, and cyclic fatty acids in thermally oxidized and frying fats and fish oils [10]. This technique has become an accepted method for evaluating the quality of frying fats, crude and refined oils, and the effectiveness of processing. HPSEC with viscometric and refractometric detection was also used to examine the molecular weight distribution of macromolecular triglyceride polymers produced in frying oils [10]. A HPSEC-fluorometric method for the separation, detection, and quantification of neutral lipid hydroperoxides from dietary lipids and lipid systems, such as mayonnaises, was described [73]. 

## 9. Hyphenated Methods

### 9.1. Liquid Chromatography-Mass Spectrometry (LC-MS) in Mixture Analysis

Mass spectrometry coupled with normal-phase and reversed-phase HPLC is a useful method for determining the thermally labile hydroperoxides and nonvolatile high-molecular-weight secondary oxidation products of triacylglycerols, cholesterol, and phospholipids without the need for previous derivatization [10]. The utilization of thermospray, electrospray ionization (ESI), or atmospheric-pressure chemical ionization (APCI) interfaces has met the challenge of removing the solvent from these labile compounds coming from an HPLC column prior to MS. Synthetic isomers of triacylglycerol hydroperoxides of eicosapentaenoic acid, as well as other oxidation products such as hydroxides, epoxides, and triglyceride-core aldehydes, were analyzed using the ESI technique. The APCI technique was used to qualitatively analyze oxidation products from triacylglycerols isolated by reversed-phase HPLC. The protonated molecular ions, near-molecular ions, and molecular-ion adducts, as well as characteristic diacylglycerol-fragment ions, could be used to identify triacylglycerol hydroperoxides, epoxides, bis-hydroperoxides, epidioxides, and diepoxides from oxidized triolein, trilinolein, and trilinolenin [10]. 

HPLC was used to separate various lipid-oxidation products, and in combination with chemical ionization mass spectrometry (CI-MS) with a direct-exposure probe, to analyze the purified compounds. CI-MS yields the protonated molecular ion [M + H]^+^ with isobutane and the adduct ion [M + NH_4_]^+^ with ammonia as the reagent gases. Because chemical ionization produces ionic species with significantly less energy than electron impact ionization, fragmentation is greatly reduced, yielding useful structural information and more intense mass fragments with intact hydroperoxides, mono- and di-hydroxy fatty acids without derivatization. The direct-exposure CI technique with isobutane as reagent gas provided structural information on the position of the hydroperoxide group and fragmentation patterns for the isomeric methyl 9- and 13-hydroperoxides of linoleate purified by HPLC (Figure 11). Although both the 9- and 13-hydroperoxides produced strong CI ions with isobutene at *m*/*z* 309 and 311, the MS/MS daughters of these ions were significantly different. Intense daughter fragments at *m*/*z* 99 (presumably due to hexanal–H^+^) of the 13-hydroperoxide and *m*/*z* 185 (probably due to methyl 9-oxononanoate–H^+^) of the 9-hydroperoxide were obtained from the ion at *m*/*z* 309 (Figure 11).

Both 9- and 13-hydroperoxides showed a more stable molecular adduct ion (M + NH_4_)^+^ at *m*/*z* 344, with ammonia as the CI gas. The 9-hydroperoxide produced methyl 9-oxononanoate (*m*/*z* 204) and decadienal (*m*/*z* 170), and the 13-hydroperoxide produced methyl 13-tridecadienoate (*m*/*z* 256), as expected by homolytic cleavage. Hydroperoxy epidioxides and dihydroperoxides produced from methyl linoleate and linolenate were analyzed using the same analytical method. The fragmentation ions (*m*/*z* 213, 199, 155 and 169) were attributed to *α*-scission relative to the hydroperoxy group with loss of water. Previous CI-MS analyses of linoleate hydroperoxides and secondary oxidation products revealed ions with isobutane for protonated fragments produced by either homolytic *β*-scission or heterolytic (also known as Hock cleavage) degradation pathways established under thermal and acid conditions. Thus, the fragmentation ion *m*/*z* 213 from the 11-oleate hydroperoxide can be attributed to the protonated ion (212 + H^+^) due to methyl 11-oxo-9-undecenoate, the ion *m*/*z* 199 from the 10-oleate hydroperoxide to the protonated ion (198 + H^+^) due to methyl 10-oxo-8-decenoate, the ion *m*/*z* 155 from the 9-oleate hydroperoxide to the protonated ion (154 + H^+^) due to 2-decenal, and the ion *m*/*z* 169 from the 8-oleate hydroperoxide to the protonated ion (168 + H^+^) due to 2-undecenal.Τhe CI ion at *m*/*z* 328 with ammonia produced intense daughter fragments at *m*/*z* 185 and 99 from the 9- and 13-hydroperoxides, respectively, as in the case of CI ions with isobutene (Figure 11). Therefore, tandem mass spectrometry and CI-MS can be used to determine the position of hydroperoxides and other complex oxygenated functions in isomeric mixtures of oxidized unsaturated lipids without the need for time-consuming derivatization [10].

As previously emphasized, the structure of regioisomers of lipid hydroperoxides cannot be determined by conventional methods (e.g., colorimetry); thus, the development of new methods can be of considerable importance. Ag^+^ coordination ion-spray mass spectrometry (CIS-MS) based on Hock fragmentation was proven to be a powerful tool to analyze regioisomers of hydroperoxides [74]. According to Milne and Porter [75] CIS–MS can be coupled with the RP–HPLC method by the addition of AgBF_4_ to the mobile phase or to the HPLC effluent postcolumn, in order to identify complex mixtures of phospholipid hydroperoxides. In the same direction, the application of LC-CIS–MS in the analysis of the primary hydroperoxides of the methyl and cholesterol esters of docosahexaenoic acid and those of 1-palmitoyl-2-docosahexaenoyl- *sn*-glycero-3-phosphocholine was also reported [76]. LC-MS/MS analysis made it possible to elucidate the mechanism of lipid peroxidation and its main components. The ozonation of 1-palmitoyl-2-oleoyl-*sn*-glycero-phosphocholine (POPC) in an ethanol-containing solvent was investigated using CL-HPLC with on-line ESIMS, and its identification was based on MS in high-resolution fast atom bombardment mode and NMR spectroscopy. A considerable amount of a novel ethoxy hydroperoxide molecule, which is a potentially reactive ozonized lipid present in food and biological tissues, was produced as a result of the reaction [77]. Zeb and Murkovic [78] found that the isocratic HPLC-ESI–MS is a useful method for the identification and characterization of oxidized species of triacylglycerols (TAGs). Among the oxidized species of TAGs, mono-hydroperoxides, bis-hydroperoxides, epoxy-epidioxides, and epoxides were the major compounds identified.

Ito et al. [79] in order to identify the position of the hydroperoxy group in lipid hydroperoxides (LOOHs), prepared hydroperoxyl octadecadienoic acid (HPODE) and hydroperoxyl eicosatetraenoic acid (HPETE) and analyzed them by quadrupole-time-of-flight MS/MS in the absence and presence of alkali metals. The presence of alkali-metal—especially sodium—collision-induced dissociation (CID) of all LOOH isomers yielded structure-diagnostic fragment ions that were highly useful in identifying the position of the hydroperoxy group, even in the presence of background contaminants such as other oxidation products and unoxidized lipids. For instance,13-9*Z*,11*E*-HPODE yielded a characteristic fragment ion (*m*/*z* 247.1 [M + Na − CH_3_(CH_2_)_4_ − OH]^+^) corresponding to a 88 Da loss from the sodiated molecular-ion mass (*m*/*z* 335.2 [M + Na]^+^) (Figure 12). It is noteworthy that this fragmentation pattern was observed for all HPODE and HPETE isomers, except 12-9*Z*,13*E*-HPODE and 14-5*Z*,8*Z*,11*Z*,15*E*-HPETE.

Cholesterol ester hydroperoxide (CEOOH), one of the main lipid-oxidation products contained in oxidized low-density lipoprotein (LDL), is closely related to several diseases, according to studies in oxidized LDL. Ito et al. [80] achieved diastereoselective separation of CEOOH-bearing 13RS-9*Z*,11*E*-hydroperoxy-octadecadienoic acid (13(RS)-HPODE CE) using LC-MS/MS equipped with a chiral column (Figure 13). This method can differentiate between enzymatic oxidation and other oxidation mechanisms (i.e., radical oxidation and singlet-oxygen-induced oxidation) of CE. 

Recently Kato et al. [81] investigated the effects of different alkali metals on the fragmentation of LOOHs. From the analysis of PC 16:0/18:2;OOH (phosphatidylcholine) and FA 18:2;OOH (fatty acid), fragmentation pathways and ion intensities were found to depend largely on the binding position and type of alkali metals (i.e., Li^+^, Hock fragmentation; Na^+^ and K^+^, *α*-cleavage (Na^+^ > K^+^); Rb^+^ and Cs^+^, no fragmentation). Furthermore, it was suggested that this method can be applied to determine the hydroperoxyl group position of esterified lipids (e.g., phospholipids and cholesterol esters) as well as in polyunsaturated fatty acids (PUFAs) including ω-3, ω-6, and ω-9 FA. 

Different oxidation methods, such as photo-oxidation or auto-oxidation, produce characteristic lipid-hydroperoxide isomers (Figure 14). Ito et al. [82] developed a chiral stationary-phase LC-MS/MS (CSP-LC-MS/MS) method to analyze the positional and *cis*/*trans* isomers of lipid hydroperoxides with the use of a chiral column and alkali metals (i.e., sodium ion). The combination of the CSP-LC-MS/MS with lipase enabled the understanding of lipid-oxidation mechanisms (i.e., photo- and auto- oxidation) in oxidized edible oils. By adding sodium, fragmentation patterns similar to previous studies [79] were obtained during CSP-LC-MS/MS analysis (Figure 15).

Triacylglycerol hydroperoxide (TGOOH) isomers in canola oil were analyzed by preparing authentic isomer reference compounds and performing LC-MS/MS (MS/MS/MS) analysis [83]. This was the first report of the analysis of TGOOH at the hydroperoxide positional/geometrical isomer level. The same team developed a HPLC-MS/MS method for the analysis of linoleic-acid hydroperoxide (LAOOH) and linoleic-acid ethyl ester hydroperoxide (ELAOOH) isomers according to the previous method [82] with slight modifications and applied to food (liquor) and cosmetic (skin cream) samples [84]. ELAOOH isomers were subjected to MS/MS analysis in the presence of the sodium ion, which yielded the same fragment patterns with LAOOH. For example, 13-9Z, 11*E*-ELAOOH produced a characteristic fragment ion at *m*/*z* 275.1 ([M + Na − C_5_H_12_O]^+^) corresponding to a 88 Da loss from the sodiated molecular ion (*m*/*z* 363.2 [M + Na]^+^).

The formation of 9- and 13-hydroperoxy octadecadienoic acid (9-HpODE and 13-HpODE), was quantified by means of stable isotope-dilution analysis–liquid chromatography–mass spectroscopy [85], after short-term heating and conditions representative of long-term domestic storage of linoleic acid, canola, sunflower, and soybean oil. Briefly, the major regioisomers of autoxidized linoleic acid comprise 9- and 13-hydroperoxy linoleic acid (9-HpODE and 13-HpODE) (Figure 16). A temperature-dependent distribution of HpODE regioisomers in vegetable oils suggests their application as markers of lipid oxidation in oils used for short-term heating.

A novel liquid chromatography–tandem-mass-spectrometry method was developed to detect hydroperoxidized and epoxidized triacylglycerols (TAGs) without derivatization or hydrolyzation of food samples [86]. Under the chosen high-resolution LC-MS conditions, the oxidized TAGs were primarily detected as [M + NH_4_]^+^ adduct ions, which were also selected as precursor ions for MS/MS (Figure 17). Pure squalene hydroperoxide (SQOOH) isomers were synthesized, and a quadrupole/linear ion trap (QTRAP) MS/MS system was used as an analytical approach for SQOOH isomers [87,88]. Collision-induced dissociation produced distinct fragment ions for each SQOOH isomer, allowing multiple reaction monitoring (MRM) between various SQOOH isomers. Individual SQOOH isomers could be separated and detected with a sensitivity of 0.05 ng/injection using LC-MS/MS with MRM in a lipid extract from human forehead skin. The technique of LC-MS/MS was used to establish a method for quantifying PCOOH molecular species (1-palmitoyl-2-hydroperoxy-octadecadienoyl-*sn*-glycero-3-phosphocholine, 16:0/HpODE PC) focusing on isomers such as 16:0/13-HpODE PC and 16:0/9-HpODE PC in plasma from healthy subjects and patients with angiographically significant stenosis. Sodiated PCOOH ([M + Na]^+^, *m*/*z* 812), produced not only a known product ion (*m*/*z* 147), but also characteristic product ions (*m*/*z* 541 for 16:0/13-HpODE PC and *m*/*z* 388 for 16:0/9- HpODE PC) [79,89]. 

Kato et al. [90] in order to determine the oxidation mechanisms of human plasma lipoproteins before illness, developed novel analytical methods using LC-MS/MS for 1-palmitoyl-2-linoleoyl-*sn*-glycero-3-phosphocholine hydroperoxide (PC 16:0/18:2;OOH) and cholesteryl linoleate hydroperoxide (CE 18:2;OOH) isomers (partially including the *cis–trans* isomer) without any derivatizations (Figure 18). The predominant PC 16:0/18:2;OOH and CE 18:2;OOH isomers in LDL and HDL were found to be PC 16:0/18:2;9OOH, PC 16:0/18:2;13OOH, CE 18:2;9OOH, and CE 18:2;13OOH, which means that PC and CE in LDL and HDL are mainly oxidized by radical and/or enzymatic oxidation. Sodiated PC 16:0/18:2;OOH isomers (*m*/*z* 813 ([M + Na]^+^)) provided product ions specific of the position of the hydroperoxyl group:*m*/*z* 388 for PC16:0/18:2;9OOH, *m*/*z* 684 for PC16:0/18:2;10OOH, *m*/*z* 683 for PC 16:0/18:2;12OOH, and *m*/*z* 541 for PC 16:0/18:2;13OOH. Similarly, sodiated CE 18:2;OOH isomers (*m*/*z* 704 ([M + Na]^+^)) also provided specific product ions, *m*/*z* 195 for CE 18:2;9OOH, *m*/*z* 576 for CE 18:2;10OOH, *m*/*z* 575 for CE 18:2;12OOH, and *m*/*z* 247 for CE 18:2;13OOH (Figure 19). 

### 9.2. Gas Chromatography-Mass Spectrometry (GC-MS)

Numerous studies have been published to identify hydroperoxides and secondary oxidation products from various unsaturated lipids using GC-MS. Quantitative GC-MS analyses, however, necessitate rigorous standardization with stable derivative compounds with well-defined structures, such as hydroxy fatty-acid derivatives, which may be synthesized. Because the EI method produces significant fragmentation of molecular ions, it is not appropriate for molecules with high molecular weight. With lipid hydroperoxides and other labile oxidation products, chemical ionization (CI) and other “soft” ionization MS techniques induce less fragmentation. GC and GC-MS cannot be used to analyze allylic unsaturated hydroperoxides and their hydroxy esters. The thermally labile hydroperoxides disintegrate quickly, and EI-MS does not offer useful molecular weight information [10]. Under GC conditions, the allylic hydroxy derivatives are also labile and thermally dehydrate into conjugated polyenes. To improve the volatility of the hydroxyl groups for GC separation, they must be acetylated or trisilylated. Double bonds in the hydroperoxide molecules are typically hydrogenated and fatty-acid moieties are converted to methyl esters. For GC and GC-MS, the trimethylsilyl (TMS) derivatives of hydroxy octadecenoates, hydroxy octadecadienoates, and hydroxy octadecatrienoates from the corresponding hydroperoxides are most suitable. These derivatives produce intense ions due to *α*-scission (a) and a much less intense fragment (b) allylic to the single double bond of oleate (1), or the conjugated diene systems of linoleate (2) and linolenate (3) (Figure 20) [10].

Terao and Matsushito, [91,92] isolated hydroperoxides from photosensitized oxidation products of unsaturated triacylglycerol and determined the isomeric distribution of hydroperoxy fatty-acid components by using hydrogenation, methanolysis, and GC-MS analysis. The structure of isomeric hydroperoxides formed by autoxidation of vegetable oils was also clarified by the same procedure. Trioleoylglycerol, trilinoleoylglycerol, trilinolenolyglycerol, soybean oil, and olive oil were selected and subjected to autoxidation and the isomeric compositions of hydroperoxy fatty-acid components produced were determined [93]. Guido et al. [94] developed a method for quantifying lipid peroxidation using GC-MS. The method involved analysis of methyl-ester and trimethylsilyl-ether derivatives of various hydroxyeicosatetraenoic acids. It was demonstrated that structurally diverse hydroperoxides can be converted to trimethylsilyl (TMS) peroxides and analyzed by GC-MS without significant thermal decomposition. The hydroperoxides investigated included cumyl hydroperoxide, 2-phenylethyl-hydroperoxide, 13-hydroperoxy-octadeca-9,11-dienoic acid, 2,6-di-tert-butyl-4-hydroperoxy-4-methylcyclo-hexadienone, and 2,4,6-trimethyl-4-hydroperoxycyclohexadienone [95]. Hydroperoxyl-group positions of lipid hydroperoxides, specifically pentafluorobenzyl ester-trimethyl silyl-ether derivatives of hydroxy-substituted fatty acids were determined by GC-EI-MS, which also provides fragment ions related to α-cleavage [96]. Finally, the TMS peroxides/esters of the fatty-acid hydroperoxides (9*S*,10*E*,12*Z*)-9-hydroperoxy-10,12-octadecadienoic acid and (9*Z*,11*E*,13*S*,15*Z*)-13-hydroperoxy-9,11,15-octadecatrienoic acid were subjected to GC-MS and the products of the thermal rearrangements were identified [97]. GC-MS can disclose structural information of various substituents on hydroperoxides, such as epoxy or hydroxyl groups, as well as to identify specific hydroperoxide structures and improve the sensitivity of the analysis. The disadvantage of GC is that the hydroperoxides must be reduced and derivatized before analysis. This fact, along with the prior lipid extraction, makes the method time-consuming and inefficient.

## 10. NMR Spectroscopy of Chromatographically Isolated Hydroperoxides and Derivatives

There is extensive literature on the use of NMR spectroscopy in structure elucidation of chromatographically isolated lipid hydroperoxides and their derivatives. Emphasis, therefore, will be given only to those publications which contributed to the general applicability of NMR methods. In early studies, Frankel et al. [98] utilized ^13^C-NMR to investigate allylic hydroperoxides resulting from the oxidation of methyl oleate which were concentrated by solvent partition [98]. The allylic methylene carbon atoms were found to be strongly deshielded, due to the effect of the OOH group, at ~81.1 and 86.9 ppm for the *cis* and *trans* geometric isomers, respectively. The mixture was then reduced with sodium borohydride to produce the corresponding allylic alcohols. The *cis* and *trans* fractions were separated by TLC and were hydrogenated over palladium prior to GC-MS analysis as triethyl silyl (TMS)-ether derivatives. Again, a significant ^13^C chemical-shift difference was found for the *cis* C-OH (δ = 67.5–67.8 ppm) and *trans* C-OH (δ = 73.1 ppm). Quantitative GC-MS data were found to be in very good agreement with the % composition of the original mixture of hydroperoxides, based on ^13^C-NMR integration data. It was concluded that ^13^C-NMR chemical shifts of the allylic methylene carbons can provide a convenient method for the determination and quantification of the geometric isomerism of oxidation that occurs symmetrically around the double bond [10,98]. 

Frankel et al. [99] utilized a combination of HPLC and ^13^C-NMR to establish the positional and geometric stereoisomers of methyl linoleate hydroperoxides. Since hydroperoxide isomers isolated by HPLC are subject to decomposition, the more stable isomeric dienols, prepared by sodium borohydride reduction of hydroperoxides, were analyzed by preparative HPLC. The ^13^C-NMR assignments were based on literature data of unsaturated fatty acids and lanthanide shift reagents. The NMR signals of the olefinic carbons of the four geometric isomers (Table 2) clearly show that the resonances of the 13-OH *cis*-9, *trans*-11 and 9-OH *trans*-10, *cis*-12 cover a range of 136.1–125.5 ppm, which is wider than that of the 13-OH *trans*-9, *trans*-11 and 9-OH *trans*-10, *trans*-2 (135.9–129.6 ppm). In all cases, the ^13^C chemical shifts of the methine C-OH were found to be identical (δ = 72.9 ppm), which clearly demonstrates that they are adjacent to the *trans* bond.

The stereochemistry of hydroperoxides which are formed during the oxidation of the 9-*cis*, 11-*trans*-conjugated linoleic acid (CLA) methyl ester, was investigated in the presence of α-tocopherol, which is a hydrogen-atom donor [100]. The hydroperoxides were separated as hydroxyl derivatives by HPLC. GC-MS of the trimethyl silyl-ether derivatives was used to characterize the position of the hydroxyl group. The geometric isomerism and the position of the double bonds of the hydroxyl-CLA methyl esters were determined by the combined use of ^1^H, ^13^C and 2D ^1^H-^1^H COSY, and TOCSY and ^1^H-^13^C experiments. The methodology utilized will be exemplified in the case of the major (~50%) 13-hydroxy-9-*cis*, 11-*trans* CLA methyl ester. The ^1^H-NMR spectrum shows very characteristic multiplets in the conjugated diene region. Two deshielding resonances at 6.48 ppm (dd) and 5.97 ppm (dd) are due to the two “inner” protons H-11 and H-10, respectively (Figure 21). The two shielded multiplets at δ = 5.67 ppm (dd) and δ = 5.43 (d,t) are due to the “outer” protons H-12 and H-9, respectively. The geometric isomerism was confirmed with the use of vicinal coupling constants of ^3^*J* = 15.2 Hz and 11.0 Hz for the *trans* and *cis* isomers, respectively. The ^13^C resonance at δ = 72.90 ppm is characteristic of an allylic methine CH-OH carbon which is adjacent to a *trans* double bond. The ^13^C resonance at δ = 27.68 ppm was assigned to the allylic methylene C-8, which is adjacent to a *cis* double bond. Critical assignments were also made with the use of gradient heteronuclear multiple-bond-correlation (gHMBC) technique which provides correlations through ^n^*J*(^13^C,^1^H) couplings, where *n* = 2–4. The terminal H-18 methyl protons show strong correlations with C-16 and C-17. H-13 and H-14,14′ show correlations with C-15 and C-16, thus confirming the position of the double bond. The quantitative results clearly show that the reaction is stereoselective in favor of one major geometric isomer. The mechanism proposed involves allylic hydrogen abstraction under the formation of *cis*, *trans*-pentadiene and *trans*, *trans*-pentadiene-radical intermediates and geometric isomerization of peroxyl radicals [100] (Figure 22).

High-resolution ^1^H-NMR (600 MHz) was used to investigate oxidation products of ethyl ester all-*cis* 4, 7, 10, 13, 16, 19-docosahexaenoic acid (DHA), with and without added α-tocopherol. Correlations were found between primary oxidation products (PV and conjugated dienes) and ^1^H resonances in the region of 8.0–10.5 ppm. The peaks from the aldehyde hydrogens were only observed in the absence of α-tocopherol. With the addition of α-tocopherol, eight very sharp resonances of the hydroperoxide protons (–OOH) were observed in the region of 8 to 9 ppm. In the absence of α-tocopherol, several additional weaker signals due to hydroperoxide protons were also observed. It was concluded that the identification of the –OOH protons could not be performed on the basis of ^1^H-NMR chemical shifts [101]. The great potentialities of ^1^H-^13^C HSQC and HMBC experiments to resolve ambiguities of resonance assignments, not only of chromatographically isolated hydroperoxides but also in mixture analysis, will be discussed in Section 11.4.

## 11. NMR Spectroscopy in Mixture Analysis

### 11.1. 1D ^1^H and 2D ^1^H-^1^H-NMR Experiments

The ^1^H nucleus is the most sensitive NMR probe and is therefore appropriate for investigating minor lipid-peroxidation products [102,103,104,105,106]. The technique has, in principle, two distinct advantages: (i) it does not require any chemical-derivatization process, contrary to various spectrophotometric techniques outlined above; and (ii) changes in the starting lipid material, as well as the evolution of the various oxidation products, can be monitored in a short experimental time. In early studies, Saito [107] investigated lipid peroxidation of fish oil stored at 40 °C with the use of ^1^H-NMR. A significant increase in the integral of olefinic protons compared to the aliphatic protons was observed. The integral ratio of aliphatic (δ = 0.6–2.5 ppm) to olefinic protons (δ = 5.1–5.6 ppm) and aliphatic to diallylmethylene protons (δ = 2.6–3.0 ppm) was suggested as an index of oxidative degradation of oil samples [108]. It was suggested that ^1^H-NMR is more appropriate than the PV method for estimating oxidative deterioration in lipids containing relatively high amounts of phospholipids.

Grootveld et al. [109,110] investigated lipid-oxidation products in culinary oils which were subjected to various frying and cooking processes. Several *trans*-2-alkenals, *trans*-*trans* and *cis*-*trans* alka-2,4-dienals and *n*-alkanals were assigned and quantified using various 2D ^1^H-^1^H-NMR techniques. Silwood and Grootveld [111] performed the first detailed analysis, with the combined use of 2D ^1^H-^1^H relayed coherent transfer (RCT) and 2D ^1^H-^1^H TOCSY experiments, of the olefinic proton multiplets in the spectral region of 5.4 to 6.6 ppm and the protons bonded to the methine CH-O-O-H carbons in the region of 4.30 to 4.35 ppm. The linoleoylglycerol 1,3-dilinolein compound was used as a model which was allowed to oxidize in the presence of atmospheric O_2_ at ambient temperature for 2.0 h. The 2D ^1^H-^1^H RCT spectrum, recorded with a delay period of 50 ms, showed several highly diagnostic cross-peaks (Figure 23). More specifically, the most deshielded multiplet of the olefinic proton c (6.57 ppm) shows cross-peak connectivities with protons d (6.02 ppm), b (5.57 ppm), a (4.35 ppm), and e (5.48 ppm). Similarly, cross-peak connectivities were observed between the deshielded multiplet of the olefinic proton h (6.27 ppm), and protons i (6.05 ppm), g (5.76 ppm), j (5.46 ppm), and f (4.30 ppm). The above results allowed the unambiguous assignment of olefinic, allylic, and neighboring aliphatic protons of both *cis*, *trans*, and *trans*, *trans* isomers of 9- and 13-hydroperoxy octadecadienyl glycerol species. Similar results were obtained with the closely related 2D ^1^H-^1^H TOCSY experiment, with a typical spin-lock pulse of 70 ms. The experimental assignment was found to be in excellent agreement with the computer-simulated spectrum (Figure 24). The above assignments are also in agreement with recent DFT calculations of ^1^H-NMR chemical shifts of conjugated linoleic acid (18:2 ω-7) and model compounds which showed that the “inside” olefinic protons of the conjugated double bonds are deshielded than the “outside” protons due to the effect of through space van der Waals interactions [112]. Furthermore, the signals of the *cis* bonds are more deshielded than those of the *trans* bonds (see Section 12).

Guilllen at al. [102,106,113,114,115,116,117,118,119,120,121,122,123,124] pioneered the application of ^1^H-NMR to study the thermal oxidation of food lipids as a function of composition of the original food lipid and thermo-oxidation conditions used. Table 3 presents chemical shifts and *J* couplings in CDCl_3_ of several conjugated hydroperoxides, aldehydes, ketones, alcohols, and epoxides. Figure 25 shows selected regions of ^1^H-NMR spectra of corn oil (CO) after storage at room temperature in a closed receptable for 0 h and 121 months and of sunflower (SF) oil subjected to thermal oxidation at 70 °C for 72 h and at 100 °C for 9 h. As expected, different primary oxidation compounds are formed depending on the degradative conditions. Similarly, several sunflower oil samples stored at room temperature in closed receptacles under different oxidation conditions (Table 4, Figure 26) were investigated in the ^1^H-NMR spectral regions of 5.5 to 7.2 ppm and 7.9 to 10.0 ppm. The –OOH and the conjugated diene signals of sample S6 can be hardly distinguished from the noise level. In sample S7 and especially in S17 and S23, the presence of two signals in the hydroperoxide-proton region (8.4–8.6 ppm) and two multiplets (6.48 and 5.98 ppm) of *cis*, *trans*-conjugated olefinic protons, are clearly observed. In the samples S18 and S23 two further multiplets near the major multiplets at 6.48 and 5.98 ppm are observed, which were attributed to *cis*, *trans*-conjugated double bonds of hydroxy derivatives. Furthermore, in samples S18 and S23, the two additional multiplets at 6.2 and 5.7 ppm, were attributed to *trans*, *trans*-conjugated double bonds. Furthermore, the authors statistically analyzed relationships between oxidation conditions and the oxidation level of the samples. It was concluded that initially, hydroperoxy (*Z*,*E*)-conjugated dienic systems are formed, while at advanced oxidation stages, hydroperoxy (*E*,*E*)-conjugated dienic compounds are produced. According to literature data on model compounds, the formation of hydroperoxy (*Z*,*E*)-conjugated dienic systems is kinetically controlled at low temperatures [125]. The hydroperoxyl (*E*,*E*)-conjugated dienic compounds have greater thermodynamic stability and are formed at higher temperatures or after prolonged oxidation.

Guillen et al. [106], based on the ^1^H-NMR chemical-shift differences of the olefinic protons of *Z*,*E-* and *E*,*E-*isomers of CD-OOH and CD-OH, derived the following equations for the quantitative determination of primary oxidation products in digested lipid samples, expressed as mmol/mol of the total concentration of AG and FA present [123,124]:*E*,*E-*CD-OH (mmol/mol AG + FA) = 1000×(Pc×I_6_._18_)/N_AG+FA_(1)
*E*,*E-*CD-OOH (mmol/mol AG + FA) = 1000×(Pc×I_6_._27_)/N_AG+FA_(2)
*Z*,*E-*CD-OH (mmol/mol AG + FA) = 1000×(Pc×I_6_._45_)/N_AG+FA_(3)
*Z*,*E-*CD-OOH (mmol/mol AG + FA) = 1000×(Pc×I_6_._58_)/N_AG+FA_(4)
where I_6.18_ is the integral of the signal at 6.18 ppm corresponding to one proton of the *E*,*E*-conjugated double bond on chains also having a hydroxy group; I_6.27_ is the integral of the signal at 6.27 ppm due to one proton of the *E*,*E-*conjugated double bond on chains also having a hydroperoxy group; I_6.45_ is the integral of the signal at 6.45 ppm corresponding to one proton of the *Z*,*E-*conjugated double bond on chains also having a hydroxy group; I_6_._58_ is the integral of the signal at 6.58 ppm due to one proton of the *Z*,*E-*conjugated double bond on chains having also a hydroperoxy group (see Table 3), and N_AG+FA_ can be obtained as follows:N_AG+FA_ = 3×N_TG_ + 2×N_1,2-DG_ + 2×N_1,3-DG_ + N_2-MG_ + N_1-MG_ + N_FA_
(5)
where N_TG_, N_1,2-DG_, N_1,3-DG_, N_2-MG_ and N_1-MG_ denote the number of moles (N) of triglycerides, 1,2-diglycerides, 1,3-diglycerides, 2-monoglycerides, and 1-monogycerides, respectively. Furthermore, it was concluded that using Equation (5), N_AG+FA_ may be overestimated when the lipolyzed sample has a considerable number of other compounds containing methylenic protons bonded to carbon atoms in *a*-position in relation to the carbonyl/carboxyl group (modified AG or FA, aldehydes, ketones, etc.) which also contribute to the area of overlapped signals. Nevertheless, this approach, unlike chromatographic methods, provides significant information in a simple and fast way, without any prior chemical modification of the samples.

More recently, Gresley et al. [126] investigated lipid-oxidation products during a 300 min thermal degradation of culinary oils. The concentration of primary oxidation products was low or undetectable in coconut oil. This was attributed to saturated fatty acids at levels of ca. 90% *w*/*w* which are nearly completely resistant to thermal oxidation. Sunflower oils showed the highest concentration of conjugated diene hydroperoxydienes, hydroxymonoenes, and olefinic resonances of α,β-unsaturated aldehydes in the region of 5.4–7.1 ppm (Figure 27). This was attributed to the high content of polyunsaturated fatty acids, which are highly susceptible to peroxidation. Both (*E*,*E*) and (*Z*,*E*)-hydroperoxides were detected and their signal intensities increased as a function of heating time up to 210 min; then, a gradual decrease in signal intensities was observed, which was attributed to temperature-induced transformation to secondary oxidation products. It should be emphasized that despite the use of a high field instrument, some multiplets due to conjugated olefinic protons (denoted with m and n) were overlapped due to the presence of a,β-unsaturated aldehydes. Nevertheless, with the use of 2D experiments, the aldehyde proton doublet at 9.52 ppm and the olefinic resonances at 7.07, 6.30, 6.20 and 6.04 ppm were assigned to (*E*,*E*)-alka-2,4-dienals and the doublet at 9.48 ppm and the multiplets at 6.85 and 6.10 ppm to C3 and C2 of (*E*)-2-alkenals.

### 11.2. 1D ^1^H-NMR in Binary CDCl_3_/DMSO-d_6_ Solvents

Charisiadis et al. [127] reported a rapid and low micromolar 1D ^1^H-NMR method for the quantification of hydrogen peroxide in plant extracts. The method was based on the strongly deshielded ^1^H-NMR signal of H_2_O_2_ at ~10.30 ppm (288 K) in DMSO-*d_6_* and the reduction in the proton-exchange rate with the combined use of picric or TFA acids and low temperatures near the freezing point of the solution. The method was also extended for the determination and monitoring of the evolution of H_2_O_2_ in aqueous extracts with the use of H_2_O/DMSO-*d_6_* binary mixtures at low temperatures (~260 K) and optimum pH values for minimum proton-exchange rates [128]. A further analytical method for the detection and quantification of H_2_O_2_ in aqueous solution was based on the chemical-exchange saturation-transfer (CEST) technique in order to amplify the H_2_O_2_ signal by a factor of 10^3^, compared to the direct NMR method [129]. Quantification was achieved by the standard addition method. 

Skiera et al. [130] utilized a similar methodology to that in [127] in order to investigate in detail the effect of solvents with significantly different dielectric constants and solvation ability (benzene-*d_6_*, CDCl_3_, acetone-*d_6_* and DMSO-*d_6_*) and their binary mixtures, on the chemical shifts and linewidths of the hydroperoxide protons. CDCl_3_/DMSO-*d_6_* (5:1 *v*:*v*) appears to be an optimum binary mixture, which results in a significant deshielding in the range of 10.1–11.00 ppm and narrowing of the OOH signals. A similar effect has already been investigated in detail for phenol-type OH groups of natural products [131,132,133,134]. Figure 28 shows excellent resolution in the hydroperoxide spectral region for various oils with high content in oleic acid (olive oil), linoleic acid (sunflower oil), and linolenic acid (linseed oil). The assignment was based on the comparison with the oxidation products of oleic, linoleic, and linolenic methyl esters (Figure 29). Of particular interest are three strongly deshielded signals in the spectral region of 10.2–10.4 ppm of olive oil (Figure 30). These signals were shown to be due to squalene, which is a functional lipid and a precursor of sterols and terpenoids. Squalene is also susceptible to oxidation due to the presence of six nonconjugated double bonds. Comparison of the ^1^H-NMR method with the classical VP analytical method was performed using two artificial samples: an oxidized methyl linolenate standard and a triacylglyceride consisting of a saturated fatty acid in the middle chain. It was concluded that both methods had similar analytical performance characteristics (Figure 31). Comparison of the two methods with the use of 290 edible oil samples showed relatively good agreement for sunflower oils, corn oils, and thistle oils (Figure 32). On the contrary, significant deviations were observed for nut oils, black seed oils, pumpkin seed oils, and olive oils. The authors, furthermore, presented a critical evaluation of the discrepancies between the ^1^H-NMR and PV methods. The high PV values of black seed oils were attributed to the presence of natural oxidizing compounds in the oil matrix, such as thymoquinone. In the case of olive oils, the low PV values, compared with the NMR method, were attributed to the presence of phenolic compounds with ^1^H-NMR resonances of the OH groups in the same spectral region to those of hydroperoxide functional groups. A ^1^H-^13^C HMBC NMR technique could, in principle, be utilized to distinguish phenol-OH and hydroperoxide-OOH resonances, due to significantly different magnitudes of ^3^*J*(^13^C-C-O-H) and ^3^*J*(^13^C-O-O-H) coupling constants. This aspect will be discussed in Section 12.

### 11.3. Band-Selective 1D-NOESY and TOCSY in Binary CDCl_3_/DMSO-d_6_ Solvents

The excellent resolution of the hydroperoxide protons obtained with the use of CDCl_3_/DMSO-*d_6_* binary mixtures enabled Merkx et al. [135] to utilize band-selective 1D-NOESY and TOCSY pulse sequences to obtain specific assignments of hydroperoxide signals. Since the resonances of the oxidation products could be several orders of magnitude weaker than the abundant signals of the nonoxidized lipids, the use of 1D-NOESY band-selective gradient pulse, with a Gaussian inversion pulse, provides a significant improvement of the receiver gain, and thus in the achievable S/N ratio (Figure 33). The 1D-NOESY experiment was used for the selective, through-space, magnetization transfer from the hydroperoxide proton (OOH) to the neighboring allylic CH proton (Figure 34). Application of selective 1D-TOCSY on the identified allylic proton results in the magnetization transfer to the neighboring olefinic protons. The magnitude of the ^3^*J* coupling of the CH=CH moiety was utilized for the identification of the *cis*/*trans* geometric isomerism since ^3^*J*_trans_ (15–17 Hz) is significantly larger than the ^3^*J*_cis_ (11–12 Hz). The above assignment procedure was performed for oleic, linoleic, and α-linolenic acids (Figure 35 and Table 5). The chemical shifts of the allylic protons are not significantly affected by the position in the fatty-acid alkyl chain; therefore, the assignment of the hydroperoxide in the 8- and 11- position of oleic acid and 9- and 13-position of linoleic acid could not be elucidated. For α-linolenic esters, the identification of cyclic and “inner” hydroperoxides was achieved with the use of α-tocopherol, which inhibits cyclization mechanisms.

Furthermore, Merkx et al. [135] investigated, with unsupervised modeling by PCA, the oxidative stability of a conventional mayonnaise formulation stored at 22 °C for 22 days and at 50° for 53 days. The positive loadings for PC2 (Figure 36B) were assigned to the major hydroperoxides and the negative loadings to aldehydes. At ambient temperatures, the formation of hydroperoxides exceeded the formation of aldehydes. At 50 °C the opposite trend was observed due to instabilities of hydroperoxides. The authors, furthermore, investigated the effect of storage temperature on the *trans*-*trans* LA-OOH vs. *cis*-*trans*-LA-OOH (Figure 37B). At 50 °C the formation of *trans*-*trans*-LA-OOH exceeds that of *cis*-*trans*-LA-OOH, while at 22 °C the opposite trend was observed. This was attributed to temperature-induced *trans* isomerization of the native *cis* bonds of LA.

### 11.4. 2D ^1^H-^13^C HMBC Experiments in Mixture Analysis

Sharp hydroperoxide-proton resonances have been reported in the case of oxidation of DHA ethyl ester [101] and conjugated linoleic-acid (CLA) methyl ester [136]; however, no attempt has been made to utilize the -OOH resonances as structural and analytical tools. Ahmed et al. [137] achieved very sharp resonances Δν_1/2_ ≤ 2.5 Hz, of the hydroperoxide protons in CDCl_3_ of a sample of methyl oleate, which was oxidized in the presence of atmospheric oxygen in a breaker at 70 °C for 135 h. The excellent resolution of the C-O-O-H protons, without the use of CDCl_3_/DMSO-*d_6_* binary mixtures, was attributed to the absence of acidic groups, contrary to the case of the free fatty acids, which accelerate intermolecular proton transfer and thus result in linewidth broadening. Optimization of the ^1^H-^13^C HMBC experiments for ^3^*J*(^13^C-O-O-^1^H) coupling of ~4 Hz resulted in two sets of ^13^C connectivities at 86.7 and 81.1 ppm with excellent resolution and sensitivity (Figure 38), despite the extremely low concentration of the hydroperoxides (3.9% to 0.8%). The deshielding ^13^C connectivities at 86.7 ppm can be attributed to the *trans* geometric isomers and the connectivities at 81.1 ppm to the *cis* isomers, in excellent agreement with literature data of isolated analytes. It can therefore be concluded that Δδ(^13^C) *trans*/*cis* ~ 5 to 6 ppm of the methine CH-O-O-H carbons can be of high diagnostic value for the identification of *trans*/*cis* geometric isomerism of hydroperoxides, contrary to ^1^H chemical shifts of the hydroperoxide protons. Table 6 shows a very good agreement of the quantification data of Ahmed et al. [137] with those of isolated hydroperoxides from the literature. Similarly, the ^13^C chemical shifts of *trans*/*cis* hydroperoxides of methyl oleate, using the ^1^H-^13^C HMBC experiments, were shown to be in excellent agreement with those obtained from the literature with the use of chromatographically isolated hydroperoxides (Figure 39).

Further, ^1^H-^13^C HMBC experiments of oxidation products of methyl linoleate, after heating the sample at 70°C for 8 h, not only showed correlation of the hydroperoxide protons to allylic carbons through the ^3^*J*(^13^C-O-O-^1^H) coupling constant, but also to the next aliphatic carbon, through ^4^*J*(^13^C-O-O-^1^H) at ~ 31.5 ppm. This connectivity is very important, since this would further facilitate the assignment of C(3)-H (Figure 40). Again, the integration data (Table 7) and ^13^C chemical shifts (Figure 41) were shown to be in very good agreement with those obtained from the literature with the use of HPLC-isolated hydroperoxides. 

### 11.5. Combination of ^1^H-^13^C HMBC and 1D-TOCSY NMR Experiments in Mixture Analysis—Limitations of ^1^H-^13^C HMBC Experiments

Oxidation of methyl linolenate in atmospheric oxygen in a glass vial for 48 h at 40 °C resulted in a significant number of sharp (Δν_1/2_ ≤ 2.0 Hz) of the hydroperoxide protons in the region of 7.7 to 9.6 ppm [138] (Figure 42). Addition of 1 to 2 microdrops of D_2_O resulted in the elimination of several of these signals, including those in the strongly deshielded region of the aldehyde protons (Figure 43). The ^1^H-^13^C HMBC spectrum of the resonance at 8.04 ppm revealed the presence of ^13^C connectivity at δ = 87.5 ppm, due to the presence of a major *trans* isomer. Further, critical cross-peak connectivities of C-16 with H-16 (^1^*J*(^13^C^1^H), H-17 and H-18 confirm the presence of the 9-*cis*, 12-*cis*, 14-*trans*-16-OOH hydroperoxide (Figure 44A).

Of high diagnostic value are also the cross-peak connectivities of the strongly deshielded resonances at δ = 9.50 and 9.08 ppm with the CH-O-O-H protons at δ = 4.13 and 3.87 ppm, respectively (Figure 44B). These resonances were attributed to the presence of 16-OOH *endo*-hydroperoxides due to further cross-peak connectivities with strongly deshielded olefinic carbons. Thus, the H-16 (δ = 4.13 ppm) shows connectivities with C-16 (δ = 87.2 ppm), C-15 (δ = 83.35 ppm) and C-14 (δ = 40.7 ppm) which confirms the presence of a five-member *endo*-peroxide ring (Figure 45). Further, confirmation was achieved with the use of a selective 1D-TOCSY experiment. This technique is very important in establishing ^1^H-^1^H connectivities via scalar coupling in complex mixtures, provided that an isolated multiplet resonance can be utilized as the magnetization-transfer source. Since the allylic CH-O-O-H protons at δ = 4.13 and 3.87 ppm have significantly different chemical shifts, selective excitation of the multiplet at δ = 3.87 ppm results in TOSCY connectivities with H-7 (δ = 1.66 and 1.57 ppm), the CH_3_ group (δ = 1.07 ppm), H-15 (δ = 4.40 ppm), H-14 (δ = 2.88 and 2.23 ppm), and H-13 (δ = 4.79 ppm). Further connectivities were observed with the olefinic protons H-12 (δ = 5.58 ppm), H-11 (δ = 6.65 ppm), H-10 (δ = 6.00 ppm), and H-9 (δ = 5.54 ppm). A similar assignment procedure was used by the selective excitation of the multiplet at δ = 4.13 ppm (Figure 46).

The potentialities of the combined use of ^1^H-^13^C HMBC and selective 1D-TOCSY experiments are shown schematically in Figure 47. The resulting ^1^H-NMR quantification data of hydroperoxides and *endo*-hydroperoxides produced during oxidation of methyl linolenate (Table 8) and ^1^H and ^13^C chemical shifts of the two major diastereomeric 9-*cis*, 11-*trans*-16-OOH *endo*-peroxy (*erythro*) and 9-*cis*, 11-*trans*-16-OOH *endo*-peroxy (*threo*) hydroperoxide are in excellent agreement with those reported in the literature with HPLC isolated oxidation products (Table 9).

The successful implementation of the ^1^H-^13^C HMBC experiment depends critically on the size of ^3^*J*(^13^C-O-O-H) coupling constant and the linewidth of the labile hydroperoxide proton. Recently, Ahmed et al. [141] performed detailed DFT calculations of the ^3^*J*(^13^C-O-O-H) coupling constants of model hydroperoxides. The ^3^*J* values of the various low-energy conformers, weighted by the respective Boltzmann population factors, were found to be <$\left|1.6\right|$ Hz for the *cis*/*trans* isomers and <$\left|0.5\right|$ Hz for the *endo*-hydroperoxides. It was concluded that very sharp resonances of the hydroperoxide protons with Δν_1/2_ < 3 Hz are required for the successful implementation of the ^1^H-^13^C HMBC experiment. This condition is easily met in the case of methyl-ester derivatives; however, with free fatty acids and samples of interest, for example, in food chemistry it may be necessary to use DMSO-*d_6_*/CDCl_3_ as solvents [130] to sufficiently reduce the line widths of the -OOH protons.

## 12. Structural Investigations Based on Quantum Chemical Calculations of NMR Parameters—Can Diastereomeric Hydroperoxides Be Identified?

Quantum chemical calculations of NMR chemical shifts and coupling constants have been extensively utilized to confirm proposed structures or to aid reassignment of structures [142,143,144,145,146,147,148,149,150,151,152,153,154]. Nevertheless, in the field of free fatty acids and their hydroperoxide derivatives, only a limited number of computational studies have so far been reported [112,138,141,155,156]. Recently, Venianakis et al. [112] presented a density functional theory (DFT) study of the ^1^H- and ^13^C-NMR chemical shifts of the geometric isomers of 18:2 ω-7-conjugated linoleic acid (CLA) and nine model compounds, using five functionals and two basis sets. It was concluded that excellent linear correlations can be obtained between DFT-calculated and experimental ^1^H-NMR chemicals for the lowest-energy DFT-optimized single conformer for various functionals and basis sets, especially at the B3LYP/6-311++G(d,p) level. The other low-energy conformers have negligible effects on the computational ^1^H-NMR chemical shifts. Furthermore, the computational ^1^H-NMR chemical shifts can provide an unequivocal assignment of the geometric isomerism in conjugated systems of biological systems such as CLAs. The great sensitivity of ^1^H-NMR chemical shifts to geometric isomerism, and conformation of substituents can provide an excellent method for obtaining high-resolution structures in solution. Furthermore, the authors suggested a typical workflow for investigating resonance assignment and structures in solution, which includes the following steps:(i)The ^1^H-NMR spectra are recorded in solution, preferentially in CDCl_3_ and at 298 K, and the experimental chemical shifts, δ_exp_, are determined.(ii)The ^1^H-NMR chemical shifts are computed at the GIAO B3LYP/6-311+G(2d,p) level with the CPCM model with energy minimization using the B3LYP/6-311++G(d,p) method.(iii)A very good linear correlation between experimental NMR chemical shifts, δ_exp_, and calculated, δ_calc_, provides a strong indication that the resonance assignment and the resulting structures are correct.

Further DFT calculations of *cis*/*trans* geometric models of *cis*-OOH and *trans*–9-OOH oleate [141] showed significant chemical-shift differences of the methine CH-OOH protons with Δδ(^1^H)*_cis_*_/*trans*_ ≈ 0.55–0.65 ppm and methine CH-OOH carbons with Δδ(^13^C)*_trans_*_/*cis*_ ≈ 7.3–8.9 ppm. These values are in very good agreement with experimental chemical-shift data [10,98,99,100,137,138,157] and demonstrate that they are of high diagnostic value for the identification of the *cis*/*trans* geometric isomerism of hydroperoxides.

Quantum chemical calculations of NMR parameters have also been extensively used for the precise determination of diastereomerism of natural products with multiple asymmetric centers [138,141,142,143,158,159,160,161]. On the contrary, in the field of lipid hydroperoxides, only three papers have so far been reported [138,141,156]. It has been demonstrated that the oxidation of methyl linolenate results in the formation of four peroxyl radicals which lead to conjugated dienoic 9-, 12-, 13-, and 16-hydroperoxides [10,98,139,162] (Figure 48). The internal 12- and 13-hydroperoxides undergo 1,5-cyclization, which results in the formation of four pairs of enantiomers due to the presence of three stereocenters at C-13, C-15, and C-16. DFT calculations using the Austin–Frisch–Peterson functional with Dispersion (APFD)/6-31+G(d) basis set, showed that the *erythro*- and *threo*-diastereomeric pairs form intramolecular hydrogen-bond interactions of various strength, and thus can be distinguished on the basis of the OOH chemical shifts (Figure 49).

Further DFT calculations of various diastereomeric pairs of model *endo*-hydroperoxides (Figure 49) demonstrated that both the chemical shift of the hydroperoxide proton and δ(CH–OOH) and δ(H–5) of the epidioxide ring (Table 10) are highly diagnostic for structure elucidation of *erythro*- and *threo*-diastereomers of *endo*-hydroperoxides [141]. Further research is needed to investigate whether the various *syn*- and *anti*-stereoisomers can be distinguished on the basis of NMR parameters, such as NOE differences.

Recently, Khalifa et al. [156] isolated from human-skin-surface lipids a novel photoinduced squalene cyclic *endo* hydroperoxide. DFT calculations of ^13^C chemical shifts of several model compounds, at the ωB97X-V level of theory, using Boltzmann distribution of the various stereoisomers, showed that the most probable structure is the *cis* 1A (Table 11). The complimentary use of DFT calculations of ^1^H chemical shifts would be of interest, including those of the hydroperoxide proton, since as shown above, they can be highly diagnostic in structural analysis.

## 13. Analytical Applications in Living Cells

Methods for intracellular measurement of hydroperoxides are mainly based on the use of fluorescent-oxidation products. Such probes should be cell-permeable, selective over various oxygen-reactive species, and should remain trapped inside in the cell after reaction. The majority of the methods were focused on hydrogen peroxide (H_2_O_2_), which is a by-product of a wide range of biological processes, and can act as a messenger in cellular-signal transduction and participate in oxidation stress and damage. The 2′,7′-Dichlorodihydrofluorescein diacetate ester, D CFH-DA, is cell-permeable, where the two acetate groups are cleaved by intramolecular esterases to yield DCFH [163]. This product is colorless and nonfluorescent; however, two-electron oxidation by ROS results in the formation of DCF, which is highly fluorescent at ~530 nm on excitation at ~488 nm. DCFH derivatives are nonselective probes since they react with many oxidants such as hydroxyl radicals, nitric oxides, lipid peroxides, H_2_O_2_, etc. The disadvantages of the oxidation-based DCFH derivatives can be eliminated with the use of probes, which are based on the deprotonation mechanism of the boronate group. Of general interest is a platform for studying the chemistry of H_2_O_2_ by molecular imaging in living biological specimens. The strategy is based on the SNAP-tag technology for site-specific protein labeling with boronate-capped dyes [164]. Fusion of O^6^-alkylguanine-DNA alkyltransferase (AGT) with proteins that contain a signaling sequence allows the expression of AGT in subcellular compartments. Boronate cleavage due to the presence of H_2_O_2_, results in hybrid small molecule/protein reporters that give a fluorescent response (Figure 50) that is integrated between 500–650 nm (λ_exc_ = 488 nm).

A further significant step towards selective fluorescent probes for visualization of organic hydroperoxides (LOOHs) was the use of the Organic Hydroperoxide Resistance Repression (OhrR) protein [165]. OhrR is a transcriptional regulator in control of organic hydroperoxides in a wide range of bacteria, and senses LOOHs via oxidation of the highly concerned Cys-22 near the N terminus. The resulting conformational change leads to derepression of the corresponding genes that defend against LOOHs through reduction to ROH, which is less harmful. H_2_O_2_, due to its high polarity, is much less of an inducer of these genes than LOOHs.

A near-infrared fluorescent probe for LOOH detection in living cells was reported based on tricarbocyaninediphenylphosphine (TCP) [166]. The probe, which is nonfluorescent, exhibits a rapid fluorescence response in the presence of LOOHs (λ_ex_ = 747 nm, λ_em_ = 770 nm) and high selectivity over other ROS, with a limit of detection of 38 pM. The probe avoids background fluorescence interference in biological systems and is less cytotoxic. The application of the above methods in the detection and quantification of LOOH formation upon stress in living cells remains to be seen.

In the field of biological fluids, of particular interest is the NMR method developed by Kakeshpour and Bax [167], which is based on the fast hydrogen exchange of H_2_O_2_ protons with those of water [129] combined with selective excitation of the strongly deshielded H_2_O_2_ resonance. In this case, the longitudinal magnetization of H_2_O_2_ returns to its Boltzmann equilibrium value at the rate of hydrogen exchange, which is much faster than the T_1_ relaxation time. This rapid equilibrium permits very short interpulse delay (~51 ms) which results in S/N ratios of ca. 25:1 of a 40 nM H_2_O_2_ sample in H_2_O in 6 min using a 600 MHz instrument. This strong enhancement in sensitivity enabled Bax et al. [168] to quantitate H_2_O_2_ in rain air, exhaled breath, saliva, blood, and urine samples. The presence, however, of biological fluids of (i) phosphate ions that can catalyze the hydrogen-exchange process; (ii) catalase and glutathione peroxidase enzymes that regulate H_2_O_2_; and (iii) superoxide dismutase which catalyzes superoxide ($O_{2}^{-})$ anion radical into O_2_ and H_2_O_2_, necessitates specific sample preparation. The blood sample, for example, was frozen quickly (<3 min) into liquid isopentane (ca—140 °C). The H_2_O and H_2_O_2_ were sublimed from the frozen blood, thus removing molecules that can interfere with hydrogen exchange. The resulting concentration of H_2_O_2_ (~0.1 μM) was found to be significantly lower than most literature values [169]. The application of this methodology to lipid hydroperoxides remains to be seen.

## 14. Lipidomics of Oxidized Products

Lipidomics is a newly emerged subfield of metabolomics that focuses on the study of molecular lipids within a cell, tissue, and biofluids, including the analysis of lipid species that are related to lipid metabolism [170]. The most commonly applied analytical methods are based on MS, often combined with chromatographical separation, such as LC-MS [169,170,171,172]. Several studies have also used NMR in lipoprotein analysis in serum plasma and in investigating the authenticity of oils and lipid quality in dairy products and their geographical origin [173,174,175,176].

### 14.1. MS Techniques

Polyunsaturated fatty acids produce a variety of metabolites, including eicosanoids, docosanoids, and octadecanoids, when they are oxygenated by cyclooxygenases, lipoxygenases, and cytochrome P450 monooxygenases [177]. These metabolites (oxylipins) (Figure 51), comprise several hundreds of molecular species which are considered as key targets in disease metabolomics, as they are involved in a variety of mammalian pathologies and physiologies [178]. 

Liquid chromatography coupled to electrospray mass spectrometry was proven to be a powerful tool for their analysis, allowing specific and accurate quantitation of multiple species present in the same sample [180]. Kortz et al. [181] presented an overview of LC-MS/MS methods for the analysis of eicosanoids and related lipids, with emphasis on sample preparation strategies, chromatographic separation (including ultra-high-performance liquid chromatography (UHPLC) and chiral separation), as well as mass spectrometric detection using multiple reaction monitoring. The same year, Tsikas and Zoerner [182] reviewed LC-MS/MS and GC-MS/MS methods for the analysis of eicosanoids. A comprehensive quantification method for eicosanoids and related compounds by using LC/MS with high-speed (30 ms/cycle) continuous-ionization polarity switching was reported by Yamada et al. [183]. The method differentiated 137 targets either by chromatography or by selected reaction monitoring with lower limits of quantification, which ranged from 0.5 to 200 pg on column.

Recently, a combined analytical and computational method for the identification of oxylipins and fatty acids was presented [179]. A reversed-phase drift-tube-ion mobility coupled with high-resolution mass spectrometry (LC/DTIM-MS) workflow was able to profile and quantify (based on chromatographic peak area) the oxylipin and fatty-acid content of biological samples while simultaneously acquiring full-scan and product-ion spectra. In the same direction, Kij et al. [184] developed and validated a rapid, specific, and sensitive LC-MS/MS method for quantification of arachidonic acid-derived eicosanoids in plasma, including lipid mediators generated via COX-, LOX- and CYP450-dependent pathways. A high-resolution multiple-reaction monitoring (MRMHR) method, as a powerful tool for the simultaneous analysis of CYP450-eicosanoids from different biological samples, was also reported [185].

It can be expected that the considerable success of hyphenated MS techniques in the analysis of oxylipins will motivate applications in the field of lipid hydroperoxides as primary nonradical-reaction products, provided that internal standards of LOOHs will become commercially available.

### 14.2. NMR Techniques

Guillen et al. [186] reported a comprehensive ^1^H-NMR study of a number of oxylipins quantified in corn oil, which was subjected to mild oxidative conditions. Furthermore, data of the sequence and kinetics of formation of various oxylipins were also reported (Figure 52). Identification of the new hydroperoxide oxylipins (dihydroperoxy-nonconjugated-dienes and hydroperoxy-epoxy-monoenes) along with known oxylipins (hydroperoxy-dienes) was based on literature data. For example, the formation of the dihydroperoxy-nonconjugated-dienes was reported under different oxidation conditions [187]. These types of studies will be of importance not only in the field of food science but also in related scientific disciplines. The identification and specific resonance assignment of oxylipins, however, could be significantly facilitated, taking into consideration (i) the excellent resolution of the hydroperoxide protons obtained with the use of CDCl_3_/DMSO-*d_6_* binary mixtures [130,135], and (ii) the great potentialities of the use of band-selective 1D-NOESY, 1D-TOCSY, and 2D ^1^H-^13^C HMBC experiments [135,137,138] (see Section 11.2, Section Section 11.3 and Section Section 11.5).

## 15. Conclusions and Future Prospects

From the numerous techniques that are available as analytical and structural tools of lipid hydroperoxides, advanced separation–mass spectrometry and NMR techniques have driven the field, since they have contributed significantly to the molecular basis of the chemical processes involved in the formation of lipid hydroperoxide. This is contrary to several methods that can provide only a crude indication of the hydroperoxide species. NMR spectroscopy has the advantages of relatively simple sample preparation, high reproducibility and the use of separation techniques not being a requirement for several applications. MS spectroscopy has considerable sensitivity advantages, and the development of Na^+^/MS techniques provides structure-diagnostic fragment ions that can contribute to the facile identification of the position of the hydroperoxide group and of diastereomers using chiral column [79,80]. MS and NMR spectroscopic methods, therefore, will undoubtedly continue to be developed as primary structural and analytical tools in the field of lipid hydroperoxide research. It should be emphasized, however, that it appears unlikely that any single MS or NMR technique will be a panacea for the complete structural analysis of lipid hydroperoxides. More comprehensive and systematic comparisons, therefore, between NMR and MS analytical methods, should be performed in order to investigate their limitations, strengths, and weaknesses, as well as the advantages of their combined use, such as of automated hyphenated LC-SPE-NMR-MS platforms [188]. Cited below are selected current and potential areas where further research could be profitable.

In the field of NMR:(i)The excellent resolution and sensitivity advantages of the selective 1D-TOCSY and band-selective ^1^H-^13^C HSQC, ^1^H-^13^C HSQC-TOCSY and ^1^H-^13^C HMBC experiments show great potential in deciphering complex lipid-oxidation products [189,190,191]. For example, band-selective ^1^H-^13^C gHSQC experiments can achieve resolution in the ^13^C dimension <0.05 ppm, and thus comparable to that of the 1D ^13^C-NMR spectra.(ii)Broadband ^1^H homonuclear decoupled techniques result in highly resolved ^1^H-NMR spectra with collapsed singlets; this minimizes signal overlap, thus providing ultra-highly resolved NMR spectra [192,193]. Such methods have not been utilized in lipid hydroperoxide research, presumably due to sensitivity reasons.(iii)The latest developments, with respect to sensitivity, of the chemical-exchange-saturation technique (CEST) [129] and the selective excitation of the strongly deshielded hydroperoxide protons [166] may greatly facilitate the rapid identification of specific fatty-acid hydroperoxides in biological systems. Hyperpolarizable techniques in solution state by dissolution dynamic nuclear polarization (DNP) can also greatly improve sensitivity, although at the expense of resolution [194].(iv)Incorporation of the numerous literature ^1^H and ^13^C-NMR chemical shift and coupling constant data of lipid hydroperoxides into open-source NMR (web) databases. These data combined with prediction software will greatly facilitate structural information from ^1^H, ^13^C, ^1^H-^1^H COSY, ^1^H-^1^H TOCSY, ^1^H-^13^C HSQC, and ^1^H-^13^C HMBC NMR data [194,195,196,197].(v)DFT calculations of ^1^H-, and ^13^C- chemical shifts and *J* couplings can contribute significantly to the unequivocal resonance assignment, identification of *cis*/*trans* geometric isomers, and diastereomeric pairs of complex hydroperoxides and solvent effects [142,143,144,145,146,147,148,149,150,151,152,153,154,155,156,198,199]. Further, research is also required for the determination of chirality in solution [200].(vi)Development of officially recognized NMR methodologies for LOOHs in fats and oils through collaboration of various NMR groups.(vii)Saturation-transfer difference (STD), Tr-NOESY, and Interligand NOEs for Pharmacophore Mapping (INPARMA) techniques can be used to investigate interactions of fatty-acid hydroperoxides with full-length proteins [201,202,203], without the need of enriched protein with ^15^N and ^13^C isotopes or hydroperoxides enriched with ^13^C.(viii)In-cell NMR is a relatively new field that allows investigation of structural and dynamic properties of biological molecules in living cells [204,205,206]. Extension of the applicability of in-cell NMR in the field of LOOHs would be expected in the near future.

In the field of MS:(i)The use of Na^+^ during MS/MS analysis afforded characteristic fragment ions for diastereomers of cholesterol-ester hydroperoxides (CEOOH) separated using a chiral column. As the main lipid-peroxidation products contained in oxidized low-density lipoprotein (LDL) are cholesterol-ester hydroperoxide (CEOOH) and phosphatidylcholine hydroperoxide (PCOOH), this method can be expected to provide further understanding of the biochemical oxidation mechanisms in vivo and elucidate the involvement of CEOOH in the development of diseases [80].(ii)The collision-induced dissociation (CID) of the Na^+^ adduct, due to high sensitivity and selectivity, is expected to provide further understanding of the hydroperoxyl group position of esterified lipids (i.e., phospholipids and cholesterol esters) as well as of polyunsaturated fatty acids (PUFAs) including ω-3, ω-6, and ω-9 FA [81].(iii)Future research using alternate methods (e.g., ion-mobility MS analysis and/or extensive theoretical studies) is required, as the detailed mechanisms of lipid oxidation have yet to be established.(iv)Development of more sensitive LC-MS systems may greatly facilitate the quantitation of specific fatty-acid hydroperoxides in biological systems. Another future trend is the miniaturization of separation systems to be able to analyze lower amounts of sample material by using UPLC and nanocolumns.(v)Development of officially recognized MS methodologies for LOOHs in fats and oils through collaboration of various MS groups.(vi)In order to fully exploit MS detection, additional and orthogonal chromatographic separation is required to be able to distinguish between isomeric and isobaric structures, which cannot be differentiated by MS or MS/MS alone.(vii)Targeted LC-MS methods still have some significant limitations, which have hindered the development in the study of LOOHs, mainly due to the limited availability of standards.(viii)Specialized bioinformatic tools and databases for oxidized lipids are urgently needed for oxidative lipidomics. For example, LIPID MAPS [207] with its major changes, i.e., annotation of ring double bond equivalents and number of oxygens, the updated shorthand notation facilitates the report of newly delineated oxygenated-lipid species.(ix)Further development of the COLMAR lipids Web server for ultra-high-resolution methods for 2D NMR and MS-based lipidomics with the inclusion of data for LOOHs [208].

It would be expected that this review will stimulate and provide some guidance for future research in the exciting field of lipid hydroperoxides.

## Figures and Tables

**Figure 1 molecules-27-02139-f001:**
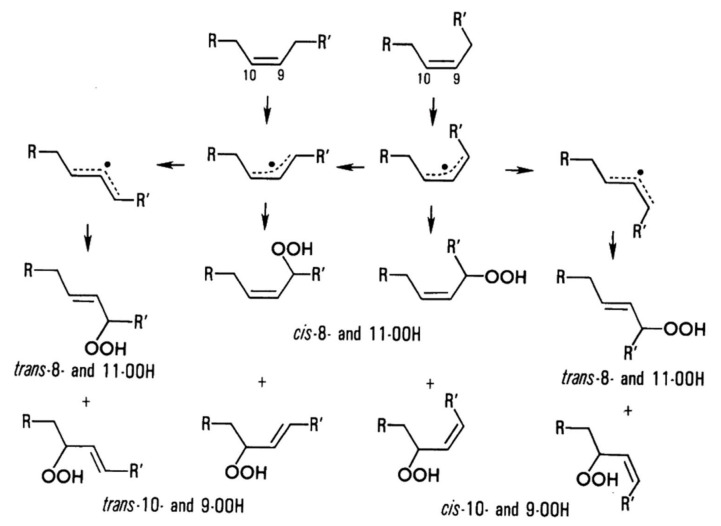
Mechanism of oleate oxidation. (R′: (CH_2_)_6_-COOH, R: (CH_2_)_6_-CH_3_). Adopted with permission from [10]. Copyright 2005, Elsevier Science; Woodhead Publishing.

**Figure 2 molecules-27-02139-f002:**
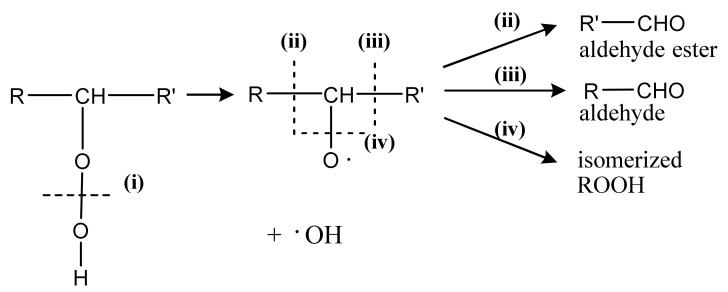
Decomposition products of unsaturated hydroperoxides [4].

**Figure 3 molecules-27-02139-f003:**
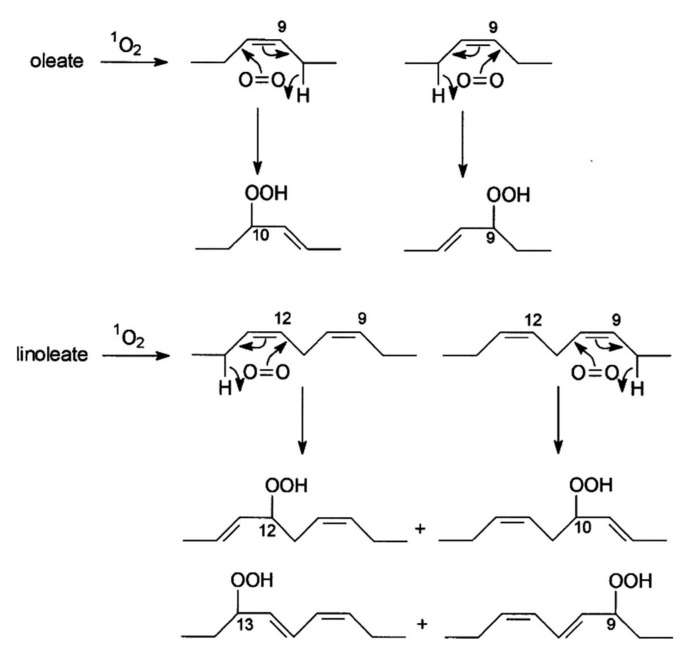
Oxidation of oleate and linoleate by singlet oxygen. Adopted with permission from [11]. Copyright 2005, Elsevier Science; Woodhead Publishing.

**Figure 4 molecules-27-02139-f004:**
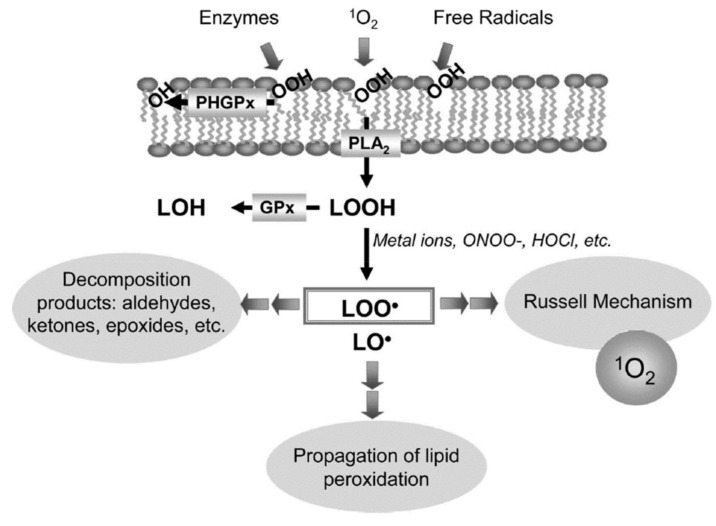
Generation and fate of lipid hydroperoxides in membranes. Adopted with permission from [9]. Copyright 2007, John Wiley & Sons.

**Figure 5 molecules-27-02139-f005:**
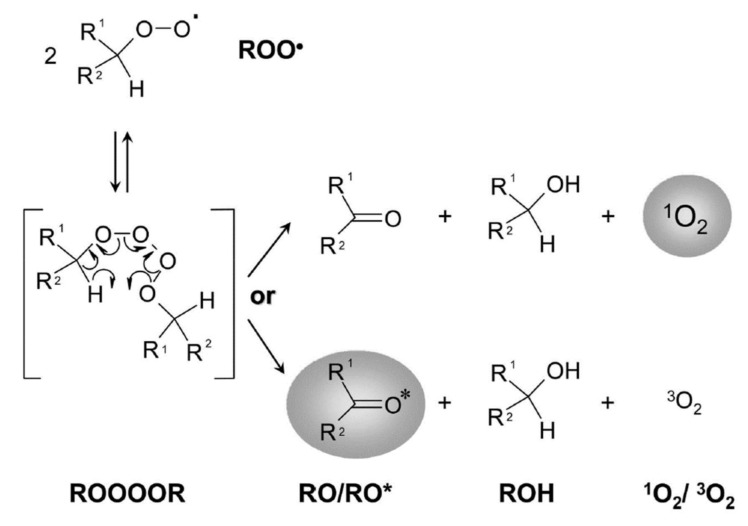
Russell mechanism for the bimolecular reaction of primary and secondary peroxyl radicals (ROO·). Adopted with permission from [9]. Copyright 2007, John Wiley & Sons.

**Figure 6 molecules-27-02139-f006:**
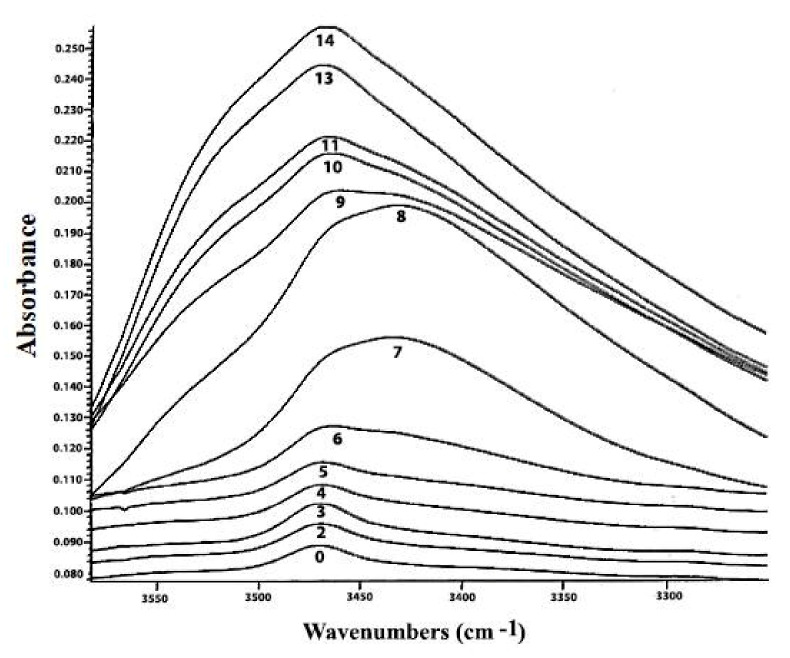
IR spectral changes in the region 3600–3250 cm^−1^ of sunflower oil on different days (0 to 14) of the oxidation process. Adopted with permission from [39]. Copyright 2007, American Chemical Society.

**Figure 7 molecules-27-02139-f007:**
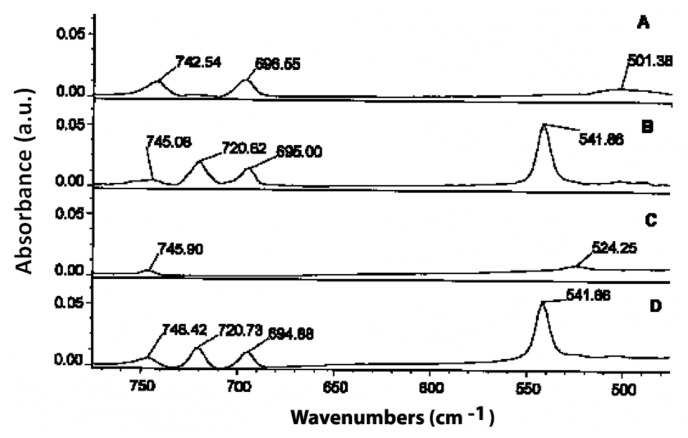
Differential spectra of TPP/hexanol (**A**), TPPO/hexanol (**B**), TBHP (**C**), and reacted TBHP and TPP/hexanol (**D**) in canola oil over the spectral range of 775–475 cm^−1^. Spectrum **D** of TBHP shows the presence of the band at 542 cm^−1^ due to TPPO formed by the reaction of TPP with TBHP in canola oil. Adopted with permission from [36]. Copyright 1997, AOCS.

**Figure 8 molecules-27-02139-f008:**
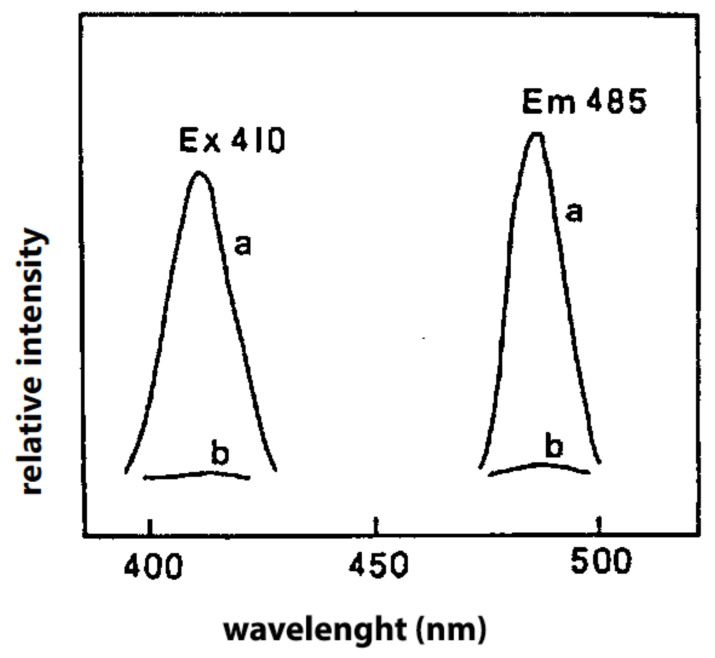
Fluorescence spectra of the hydroxylation product of l-methyl indole in pea peroxygenase reaction with methyl linoleate hydroperoxide (500 nmol); (**a**) in the presence and (**b**) in the absence of the enzyme. Adopted with permission from [49]. Copyright 1993, John Wiley & Sons.

**Figure 9 molecules-27-02139-f009:**
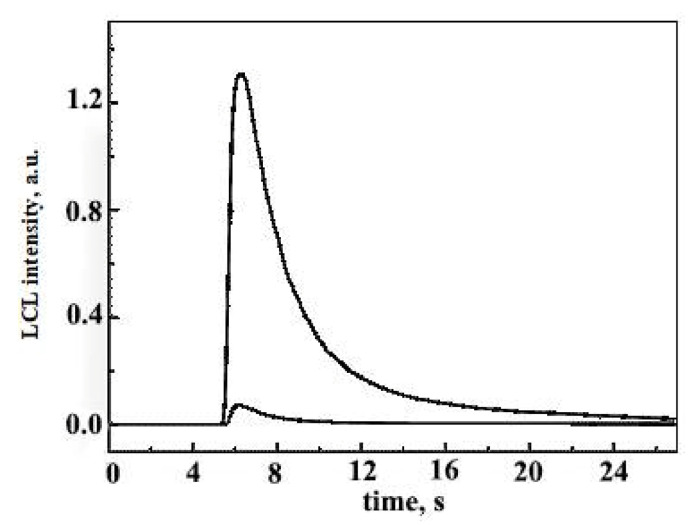
Typical luminol-enhanced chemiluminescence (LCL) signal. The specific experimental conditions were as follows: 100 pM of LOOH, luminol 0.1 mM, 5% methanol, 0.025% TRITON X-100, buffer CAPS 0.1 M, pH 10, final volume of 3 mL. The reaction started with addition of hemin 5 × 10^−6^ M. The lower curve shows emission in the absence of luminal. Adopted with permission from [54]. Copyright 2009, Elsevier.

**Figure 10 molecules-27-02139-f010:**
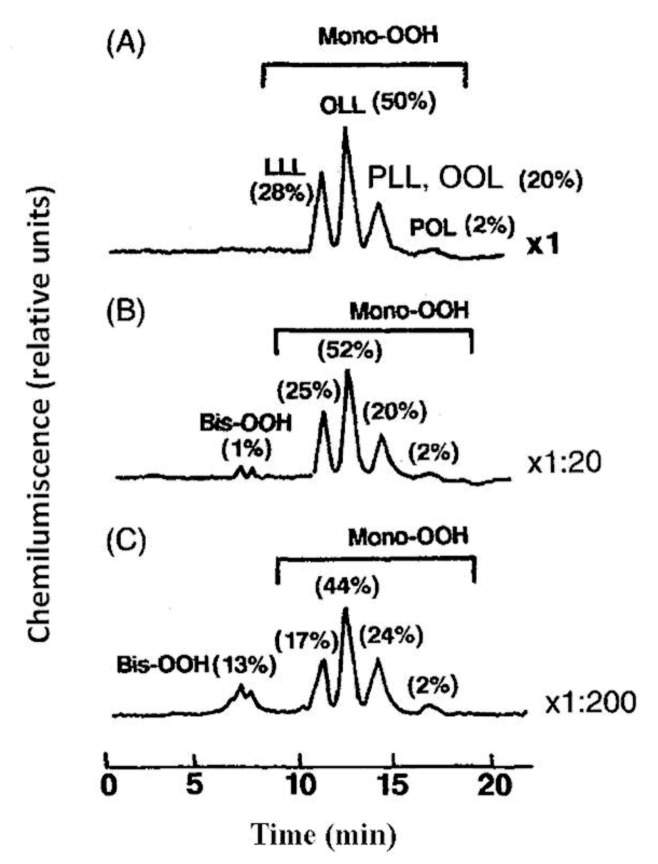
Chromatograms of chemiluminescence-detection high-performance liquid chromatography (CL-HPLC) of soybean oil ((**A**), peroxide value [PV] = 6 meq/kg; (**B**), PV = 110 meq/kg; (**C**), PV = 780 meq/kg) autoxidized at 25 °C. ×1, ×1:20, ×1:200, indicate the relative dilution ratios of oxidized oil. Trilinoleoylglycerol (LLL), oleoyl-linoleoyl-linoleoylglycero (OLL), palmitoyl-linoleoyl-linoleoylglycerol (PLL), oleoyl-oleoyl-linoleoylglycero (OOL), and palmitoyl-oleoyl-linoleoylglycerol (POL) depict the fatty-acid combination in the molecular species of triacylglycerol. Adopted with permission from [67]. Copyright 1995, AOCS.

**Figure 11 molecules-27-02139-f011:**
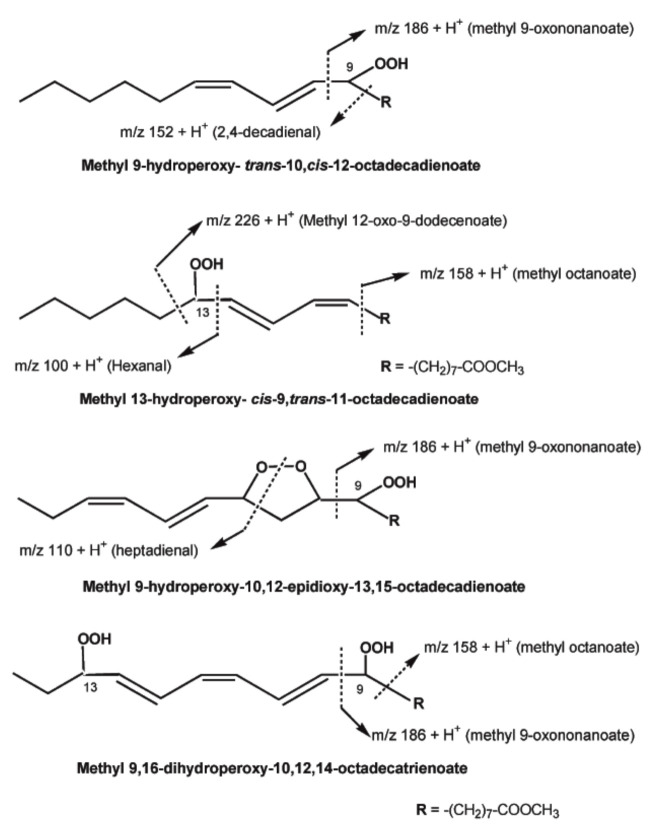
Mass-fragment ions obtained by chemical ionization mass-spectrometry (CI-MS) analysis of 9- and 13-hydroperoxides from oxidized methyl linoleate, and 9- hydroperoxy epidioxide and 9,16-dihydroperoxide from oxidized methyl linolenate. Adopted with permission from [10]. Copyright 2005, Elsevier Science; Woodhead Publishing.

**Figure 12 molecules-27-02139-f012:**
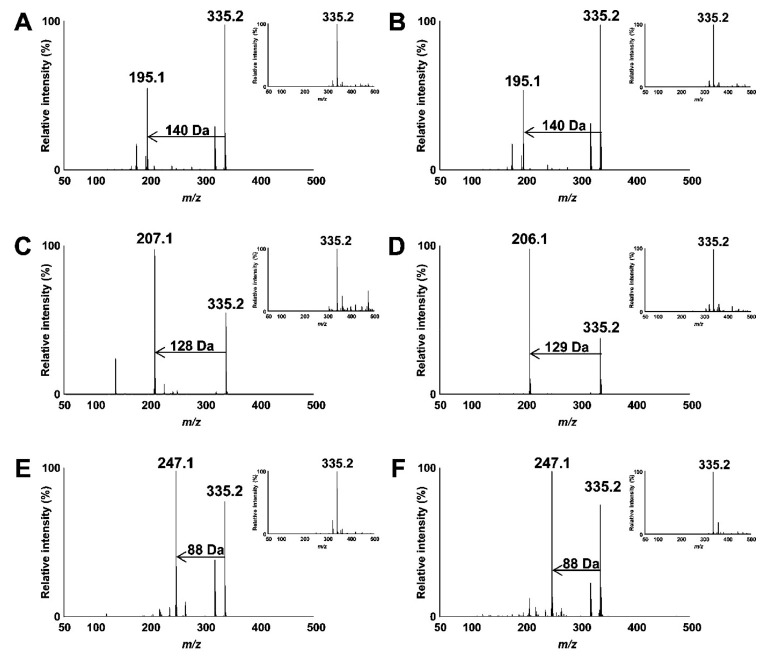
Mass spectra (Q1) and product-ion mass spectra (MS/MS) of HPODE isomers in the presence of sodium ion (positive mode). HPODE isomers were individually dissolved in methanol containing 0.1 mM sodium acetate. Each sample solution (37−89 μM) was infused directly into a microTOF-Q II mass spectrometer at a flow rate of 300 μL/h. Spectral data ((**A**), 9-10*E*,12*Z*-HPODE; (**B**), 9-10*E*,12*E*-HPODE; (**C**), 10-8*E*,12*Z*-HPODE; (**D**), 12-9*Z*,13*E*-HPODE; (**E**), 13-9*Z*,11*E*-HPODE; (**F**), 13-9*E*,11*E*HPODE) were obtained under the optimized conditions. Each spectrum is representative of at least a triplicate analysis. Reprinted with permission from [79]. Copyright 2015, American Chemical Society.

**Figure 13 molecules-27-02139-f013:**
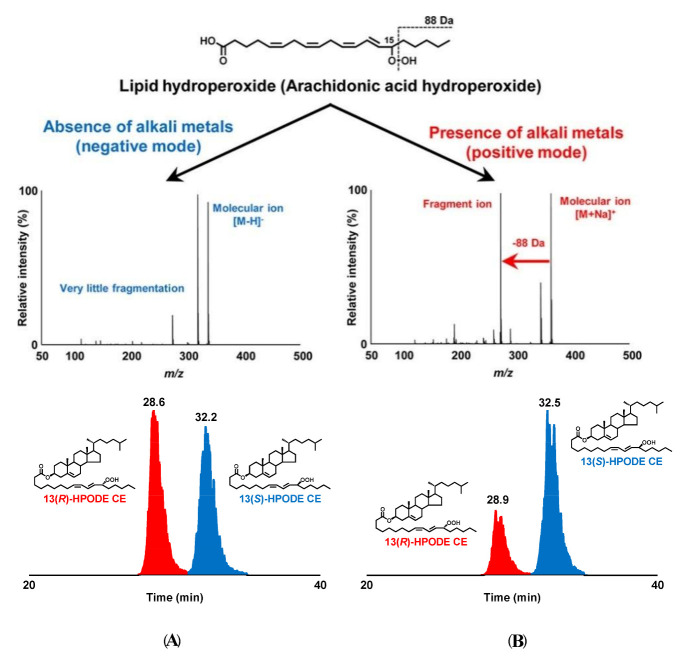
Diastereomeric analysis of synthesized 13(RS)-HPODE CE (**A**) and LOX-catalyzed oxidation product of CE (**B**). 13(RS)-HPODE CE was detected by structure-selective MRM (*m*/*z* 703 > 247). Reprinted with permission from [79,80]. Copyright 2015, American Chemical Society.

**Figure 14 molecules-27-02139-f014:**
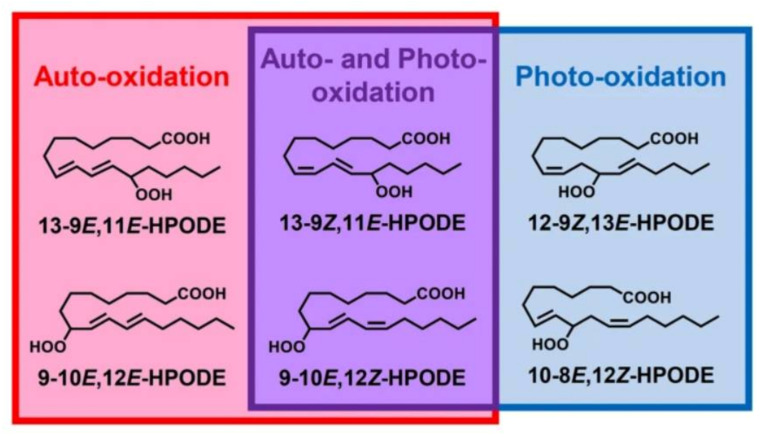
Chemical structures of six HPODE isomers (13-9*Z*,11*E*-HPODE, 13-9*E*,11*E*-HPODE, 12-9*Z*,13*E*-HPODE, 10-8*E*,12*Z*-HPODE, 9-10*E*,12*Z*-HPODE and 9-10*E*,12*E*-HPODE) [82].

**Figure 15 molecules-27-02139-f015:**
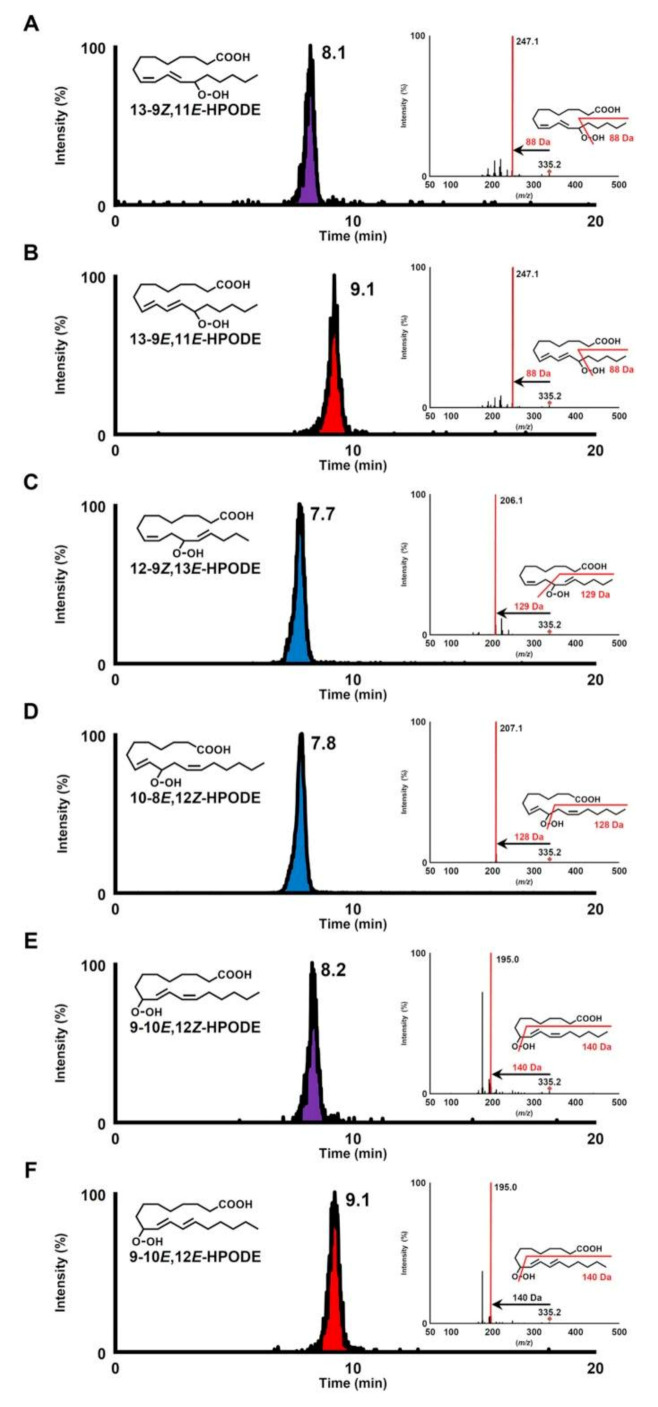
CSP-LC-MS/MS chromatogram and product-ion mass spectra (MS/MS) of standard HPODE isomers, using a selected reaction-monitoring (SRM) mode, as follows: 13-9*Z*,11*E*-HPODE and 13-9*E*,11*E*-HPODE, *m*/*z* 335.2 > 247.1 (**A**,**B**); 12-9*Z*,13*E*-HPODE, *m*/*z* 335.2 > 206.1 (**C**); 10-8*E*,12*Z*-HPODE, *m*/*z* 335.2 > 207.1 (**D**); 9-10*E*,12*Z*-HPODE and 9-10*E*,12*E*-HPODE, *m*/*z* 335.2 > 195.1 (**E**,**F**) [82].

**Figure 16 molecules-27-02139-f016:**
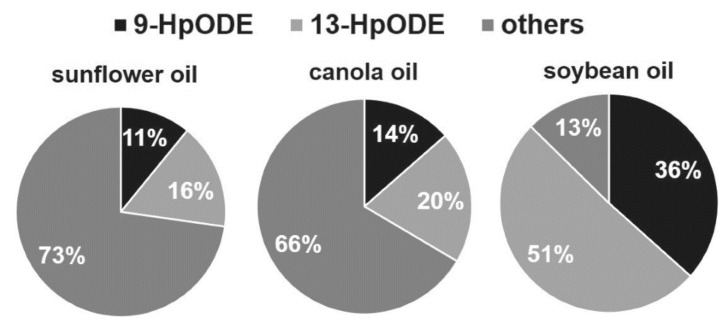
Percentage of 9-hydroperoxy octadecadienoic acid (9-HpODE) and 13-HpODEon total peroxides formed in sunflower, canola, and soybean oil stored under household conditions for 56 days. Data are expressed as the mean (*n* = 3) [85].

**Figure 17 molecules-27-02139-f017:**
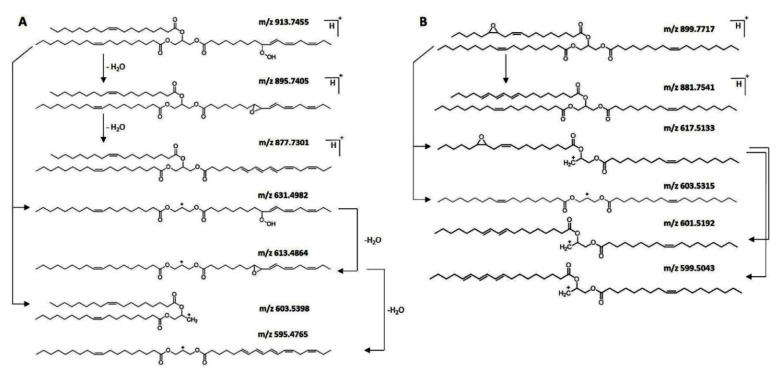
Proposed fragmentation pathway of (**A**) TAG 18:1/18:1/18:3 [OOH] and (**B**) TAG 18:1/18:1/18:1 [O] in canola oil. Reprinted with permission from [86]. Copyright 2019, American Chemical Society.

**Figure 18 molecules-27-02139-f018:**
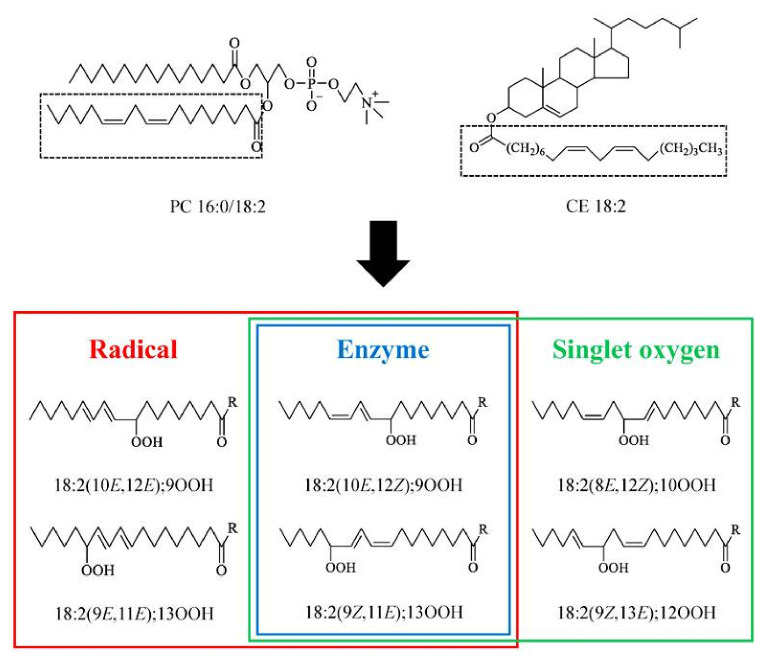
1-Palmitoyl-2-linoleoyl-*sn*-glycero-3-phosphocholine (PC 16:0/18:2) and cholesteryl linoleate (CE 18:2) peroxidation mechanisms and the structures of their hydroperoxide isomers. PC 16:0/18:2 and CE 18:2 were oxidized to 1-palmitoyl-2-linoleoyl-*sn*-glycero-3-phosphocholine hydroperoxide (PC 16:0/18:2;OOH) and cholesteryl linoleate hydroperoxide (CE 18:2;OOH) isomers, respectively, by radical, enzymatic, and singlet-oxygen (^1^O_2_) oxidation [90].

**Figure 19 molecules-27-02139-f019:**
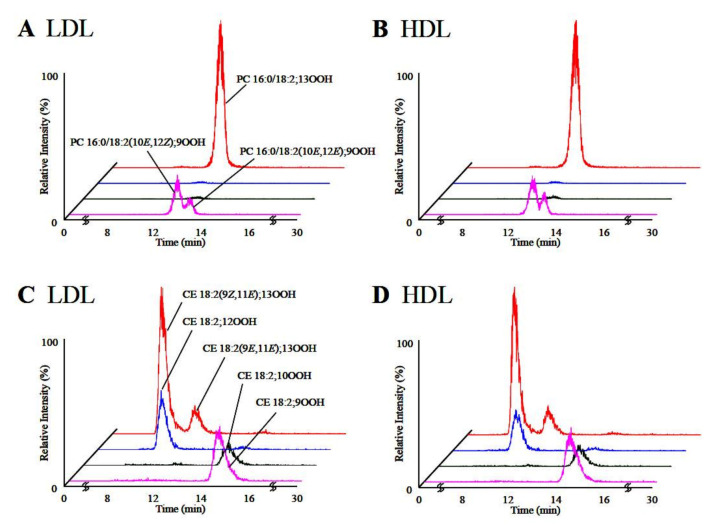
MRM chromatograms of PC 16:0/18:2;OOH isomers (**A**,**B**) and CE 18:2;OOH isomers (**C**,**D**). LOOH extracted from LDL (**A**,**C**) and HDL (**B**,**D**) [90].

**Figure 20 molecules-27-02139-f020:**
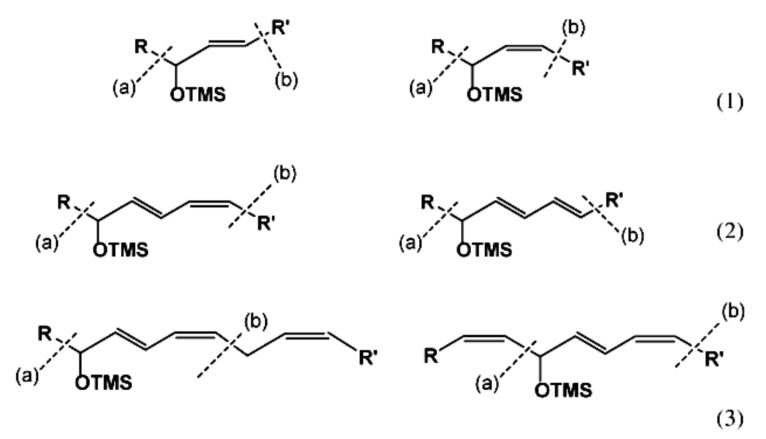
Mass-fragment ions obtained by GC-MC of trimethylsilyl (TMS) derivatives of oleate (**1**), linoleate (**2**), and linolenate (**3**) ((a) denotes *α*-scission and (b) allylic fragmentation). Reprinted with permission from [10]. Copyright 2005, Elsevier Science; Woodhead Publishing.

**Figure 21 molecules-27-02139-f021:**
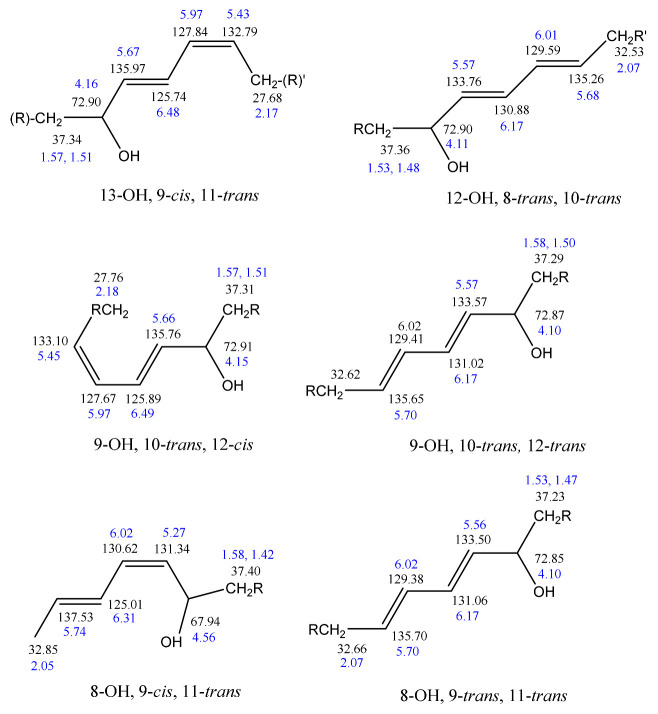
^1^H (blue color) and ^13^C (black color) chemical shifts of various hydroxy-CLA methyl esters.

**Figure 22 molecules-27-02139-f022:**
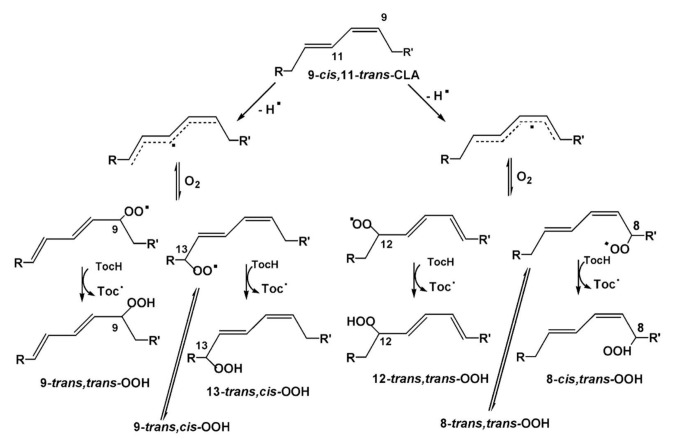
Mechanism proposed for the autoxidation of conjugated methyl 9-*cis*,11-*trans* octadecadienoate in the presence of α-tocopherol. Reprinted with permission from [10,100]. Copyright 2002, AOCS.

**Figure 23 molecules-27-02139-f023:**
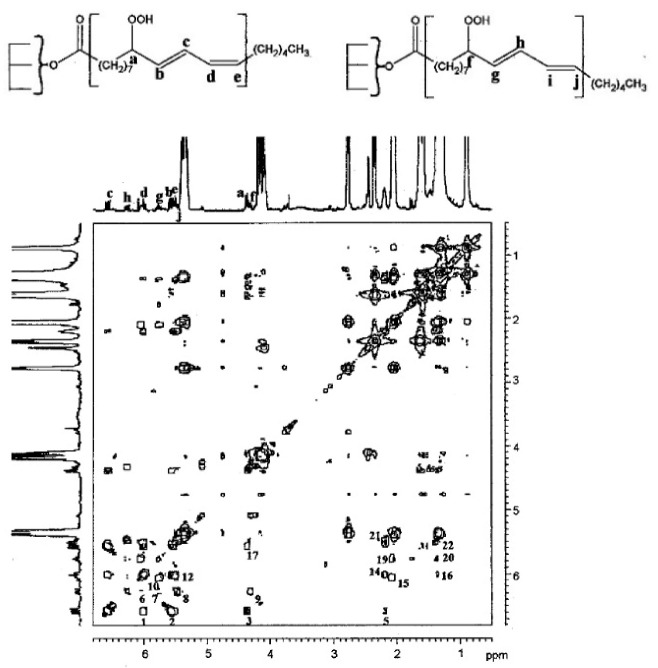
Selected 0.50–6.80 ppm region of a 400 MHz ^1^H-^1^H relayed coherence transfer (RCT) spectrum of an oxidized sample of 1,3-dilinolein in CDCl_3_ solution. Cross-peak assignments involving protons (**a**–**j**) of 9-hydroperoxy-*trans*-10,*cis*-12- (and in an analogous manner, 13-hydroperoxy-*cis*-9,*trans*-11-) and 9-hydroperoxy-*trans*-10,*trans*-12-(and in an analogous manner, 13-hydroperoxy-*trans*-9,*trans*-11-) octadecadienoylglycerol isomers are shown. Reprinted with permission from [111]. Copyright 1999, AOCS.

**Figure 24 molecules-27-02139-f024:**
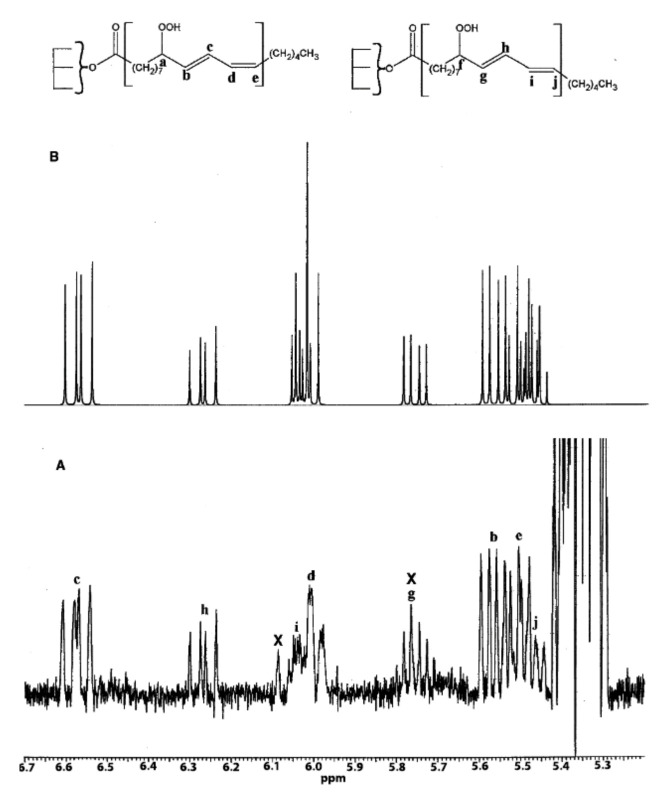
Selected 5.20–6.70 ppm regions of (**A**) an experimental 400 MHz ^1^H-NMR spectrum of an autoxidized sample of 1,3-dilinolein (in CDCl_3_ solution), and (**B**) a corresponding computer-simulated spectrum. ^1^H nuclei labels of isomeric conjugated hydroperoxyl dienes correspond to those given in the molecular structures depicted above the spectra. “x” indicates spectral artifacts. Reprinted with permission from [111]. Copyright 1999, AOCS.

**Figure 25 molecules-27-02139-f025:**
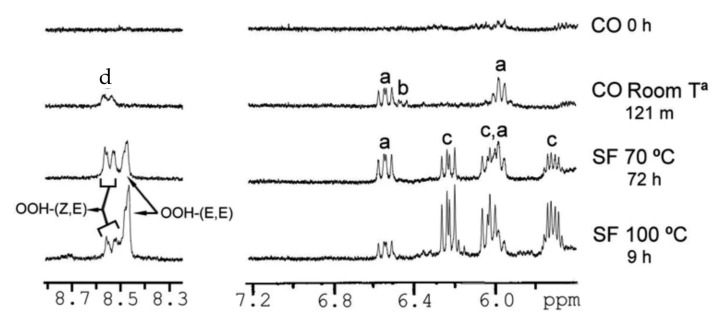
Selected regions of ^1^H-NMR spectra (400 MHz) of primary oxidation compounds formed in corn oil (CO), nonoxidized (0 h) and stored at room temperature in closed receptacles for 121 months; sunflower oil (SF) subjected (10 g oil, Petri dish) to thermal oxidation at 70 °C for 72 h and at 100 °C for 9 h. Letter abbreviations have as follows: (**a**), α-hydroperoxy (*Z*,*E*)-conjugated double bonds; (**b**), hydroxy (*Z*,*E*)-conjugated double bonds; (**c**), hydroperoxy (*E*,*E*)-conjugated double bonds; (**d**), hydroperoxide groups. Adopted with permission from [103]. Copyright 2014, Institute of Food Technologists.

**Figure 26 molecules-27-02139-f026:**
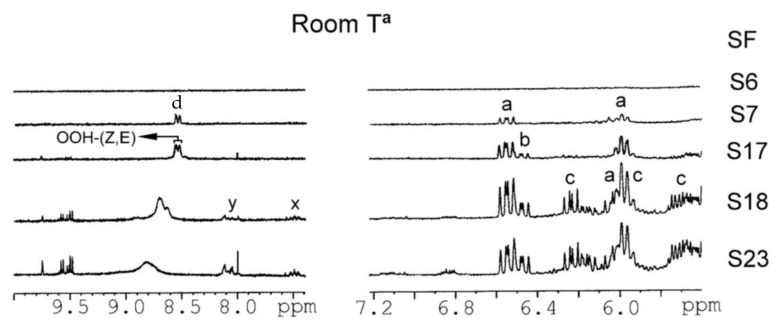
Selected regions of the ^1^H-NMR spectra (400 MHz) of sunflower oil (SF) samples stored at room temperature in closed receptacles for different oxidation conditions during storage (Table 4). Letter abbreviations have as follows: (**a**), α-hydroperoxy (*Z*,*E*)-conjugated double bonds; (**b**), hydroxy (*Z*,*E*)-conjugated double bonds; (**c**), hydroperoxy (*E*,*E*)-conjugated double bonds; (**d**), hydroperoxide groups. Adopted with permission from [103]. Copyright 2014, Institute of Food Technologists.

**Figure 27 molecules-27-02139-f027:**
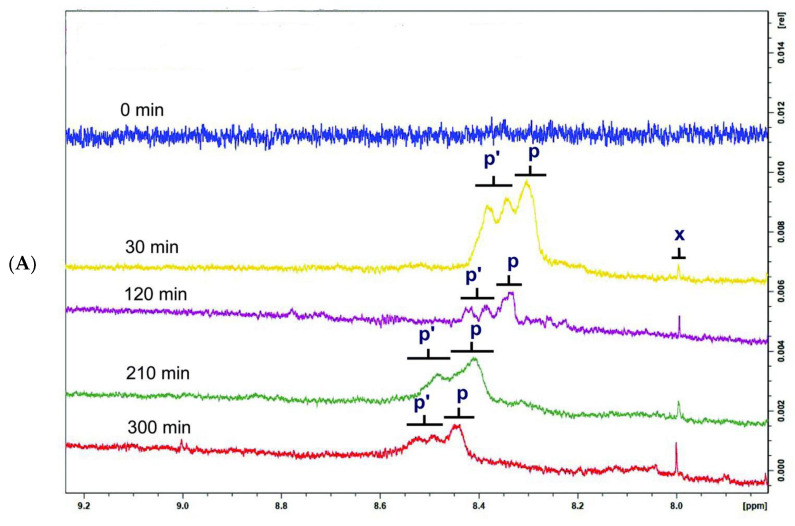
^1^H-NMR spectra (600 MHz) of (**A**) hydroperoxide group (primary LOPs) and methanoic acid present in the 7.5–9.2 ppm region. Signal p denotes the OOH–(*E*,*E*) hydroperoxide function, p′ the OOH–(*Z*,*E*) hydroperoxide function; and (**B**) of conjugated diene hydroperoxyl dienes and hydroxy monoenes (primary LOPs), and olefinic resonances of α,β-unsaturated aldehydes present in the 5.4–7.1 ppm regions of sunflower oil, thermally stressed continuously throughout a 300 min period. For m and n there is overlap, due to the presence of several unsaturated chain lengths; x denotes methanoic acid. Adopted with permission from [126]. Copyright 2019, Royal Society of Chemistry.

**Figure 28 molecules-27-02139-f028:**
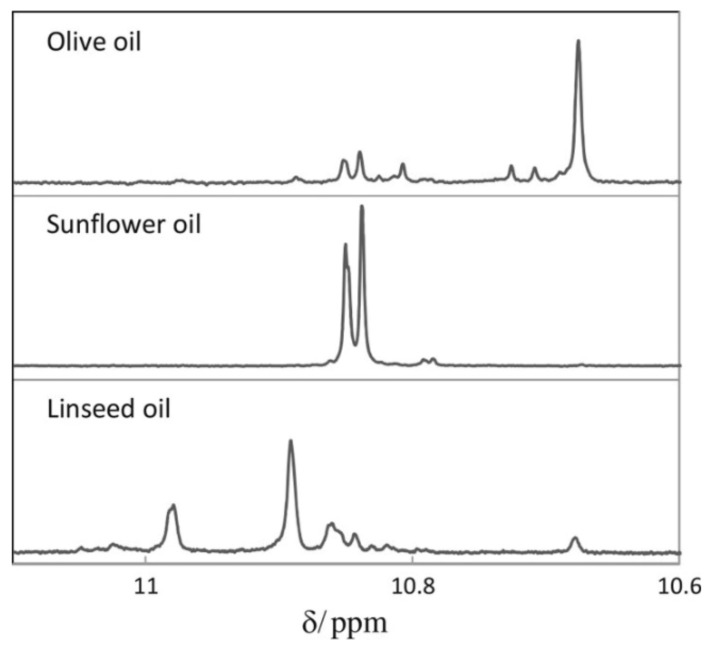
Hydroperoxide-proton region (10.1–11.0 ppm) of oils with high content in oleic (olive oil), linoleic (sunflower oil), and linolenic acid (linseed oil). Reprinted with permission from [130]. Copyright 2012, AOCS.

**Figure 29 molecules-27-02139-f029:**
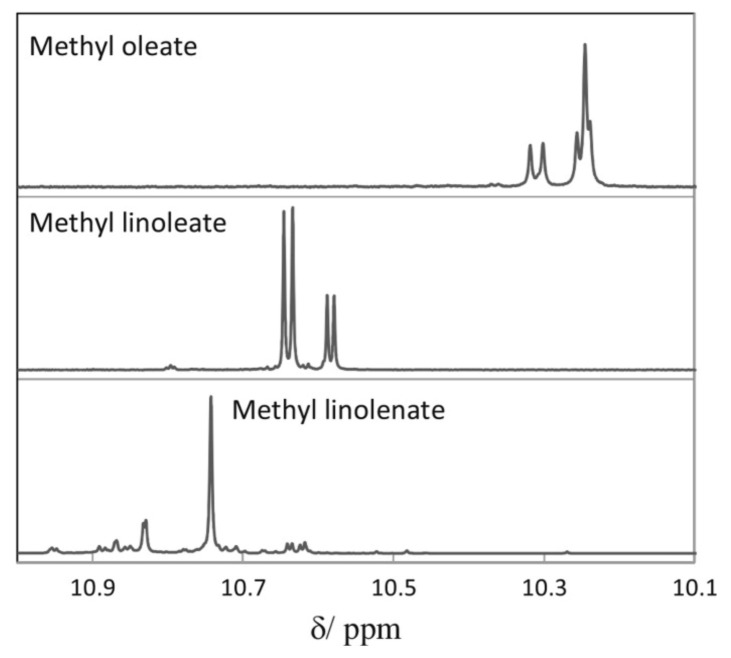
Hydroperoxide-proton region of oxidized oleic acid, linoleic acid, and linolenic acid standards. Reprinted with permission from [130]. Copyright 2012, AOCS.

**Figure 30 molecules-27-02139-f030:**
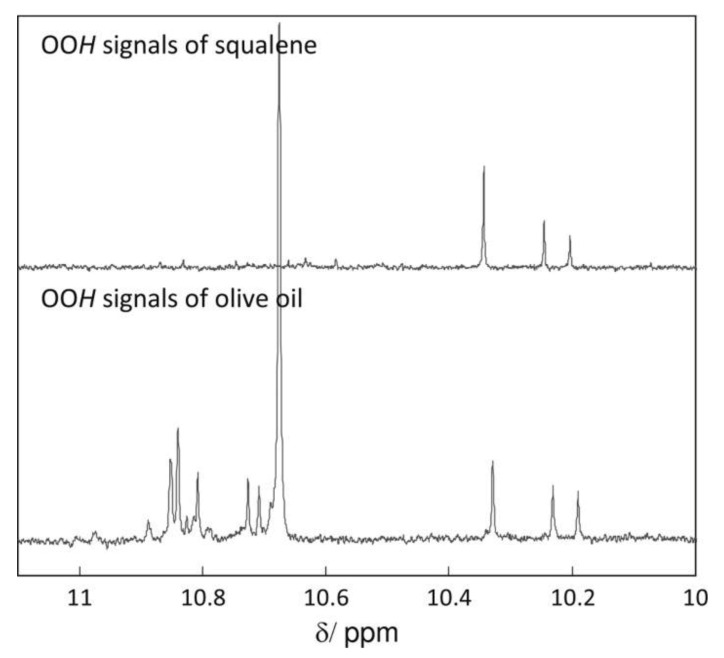
Hydroperoxide-proton region of olive oil compared to that of oxidized squalene. Reprinted with permission from [130]. Copyright 2012, AOCS.

**Figure 31 molecules-27-02139-f031:**
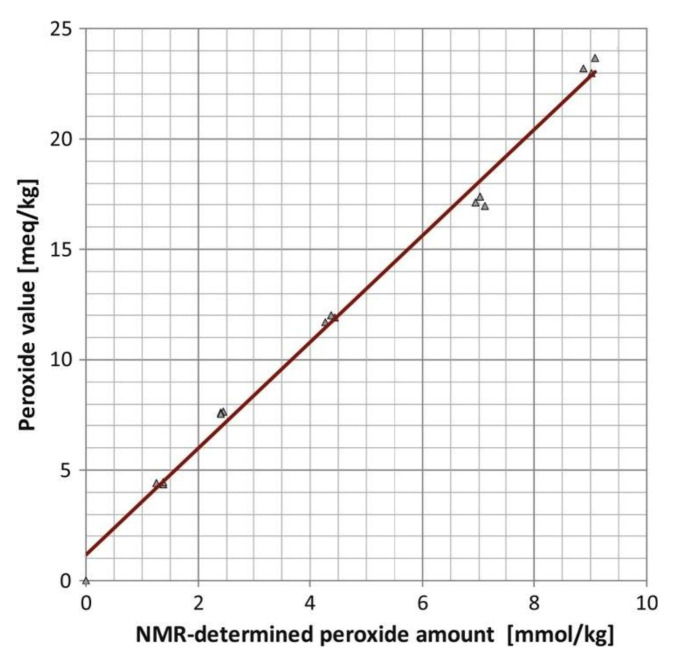
Peroxide value in mequiv/kg versus the NMR-determined peroxide amount in mmol/kg. Reprinted with permission from [130]. Copyright 2012, AOCS.

**Figure 32 molecules-27-02139-f032:**
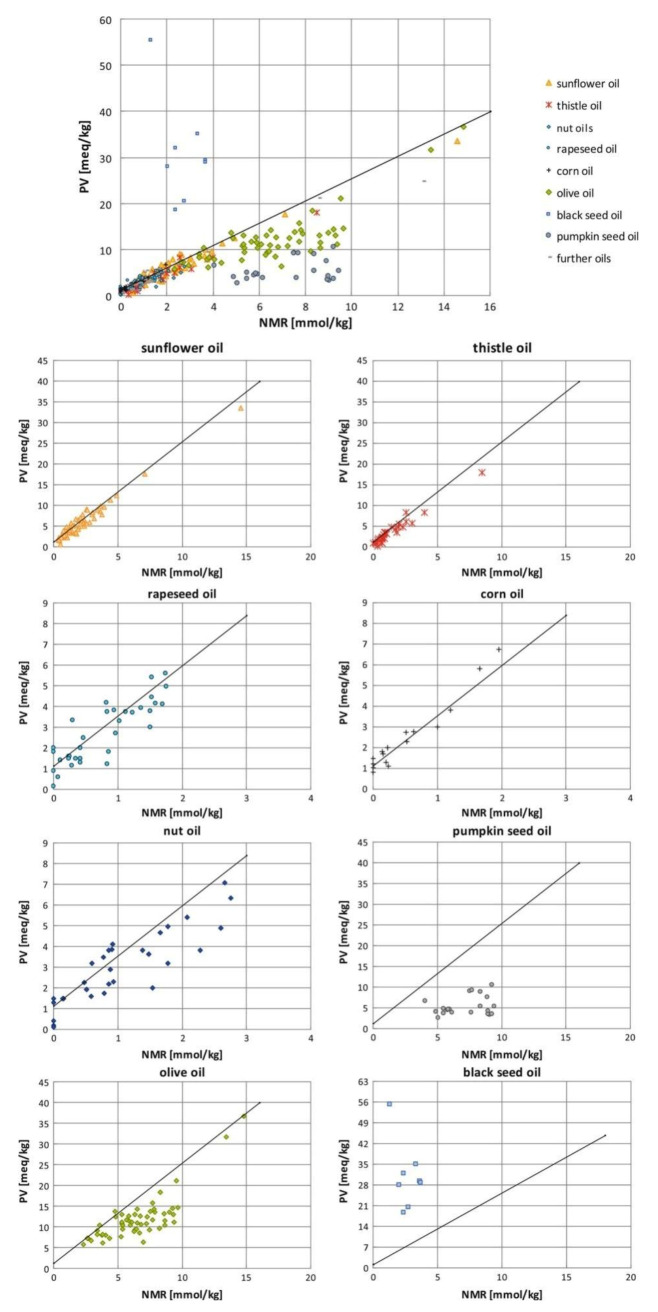
Plot of the PV versus the NMR-determined values of 290 oil samples. Reprinted with permission from [130]. Copyright 2012, AOCS.

**Figure 33 molecules-27-02139-f033:**
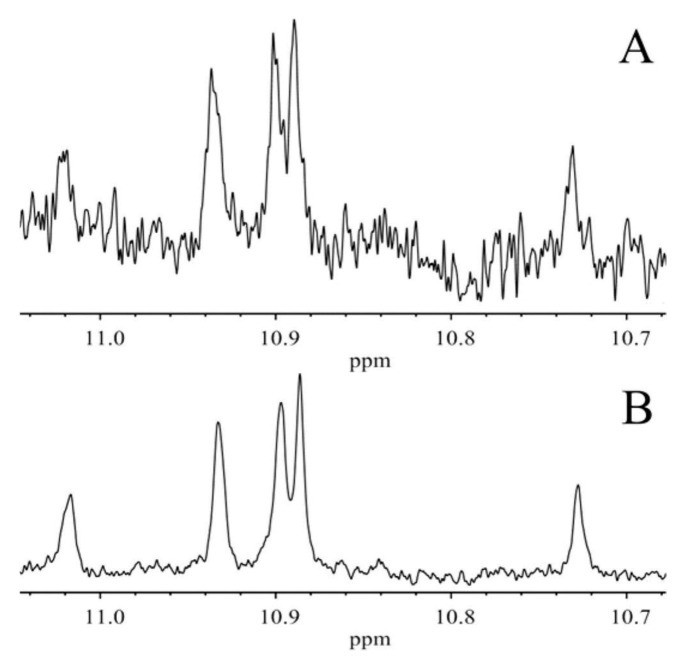
^1^H-NMR spectrum (700 MHz) of the hydroperoxide region of a CDCl_3_/DMSO-*d_6_* lipid extract prepared from a mildly oxidized mayonnaise sample. The top spectrum (**A**) is recorded with a regular excitation with a hard 90° pulse, and the bottom spectrum (**B**) was recorded with a band-selective pulse. Reprinted with permission from [135]. Copyright 2018, American Chemical Society.

**Figure 34 molecules-27-02139-f034:**
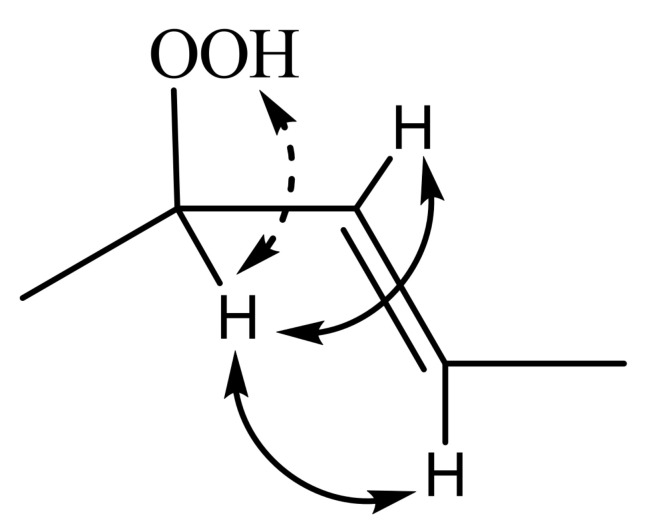
Signal assignment of hydroperoxides with the combined use of 1D NOE (
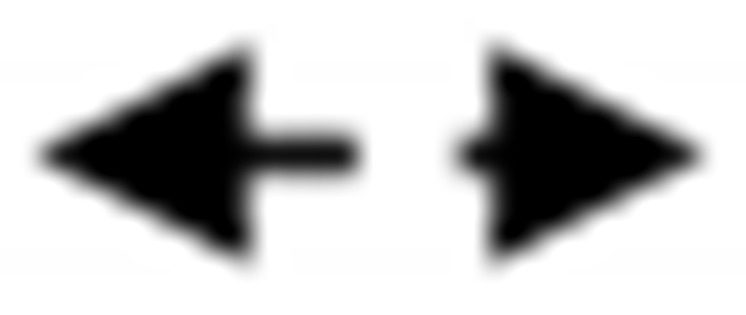
 ) and 1D TOCSY ( 
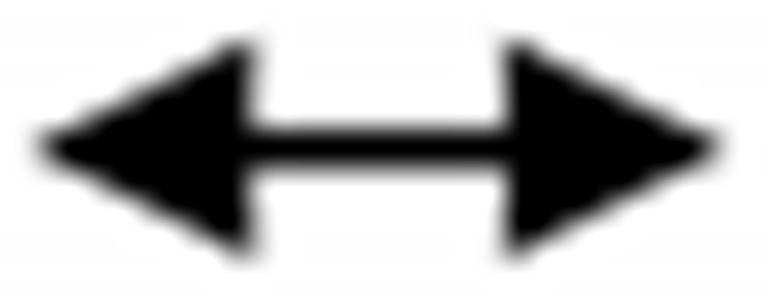
).

**Figure 35 molecules-27-02139-f035:**
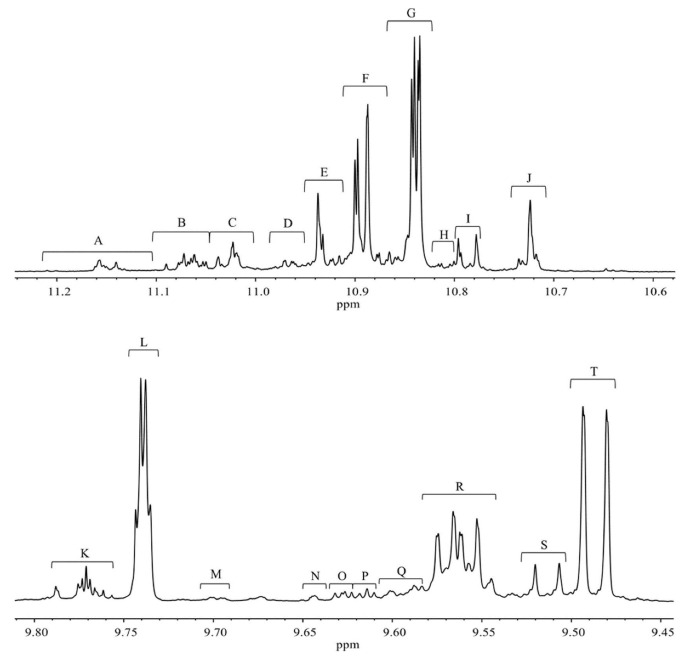
^1^H-NMR spectra (700 MHz) obtained with band-selective excitation, with assignment coding, according to Table 5. The sample contained 25% oxidized oil in 75% 5:1 CDCl_3_/DMSO-*d_6_*. Top: hydroperoxide region (δ 11.24−10.57 ppm); Bottom: aldehyde region (δ 9.81−9.44 ppm). Reprinted with permission from [135]. Copyright 2018, American Chemical Society.

**Figure 36 molecules-27-02139-f036:**
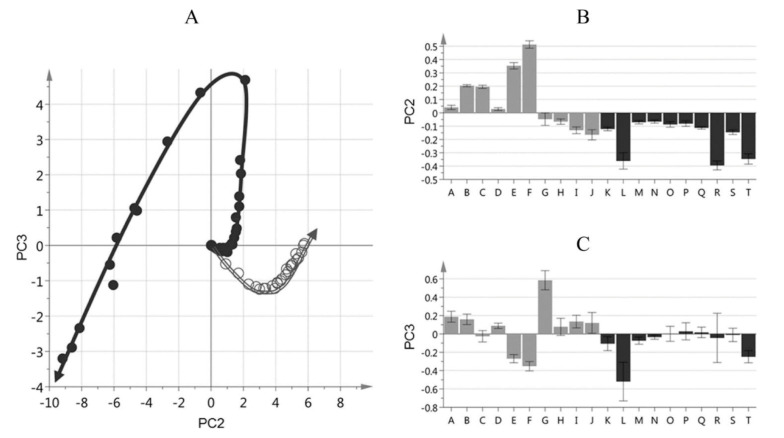
Oxidation-product profiles of mayonnaise stored at two different temperatures and followed as a function of time. (**A**) Score plot of PC 2 vs PC 3. The arrows in the score plots indicate the trajectory through time: the base of the arrow is time point zero, the arrowhead is located at the last time point. Samples at elevated (●) and ambient (○) temperature are respectively stored for 53 and 203 days. (**B**,**C**) Loading plots of PC2 and PC3, respectively, where gray bars represent hydroperoxides and black bars aldehydes. A–T correspond to the annotations in Table 5. Reprinted with permission from [135]. Copyright 2018, American Chemical Society.

**Figure 37 molecules-27-02139-f037:**
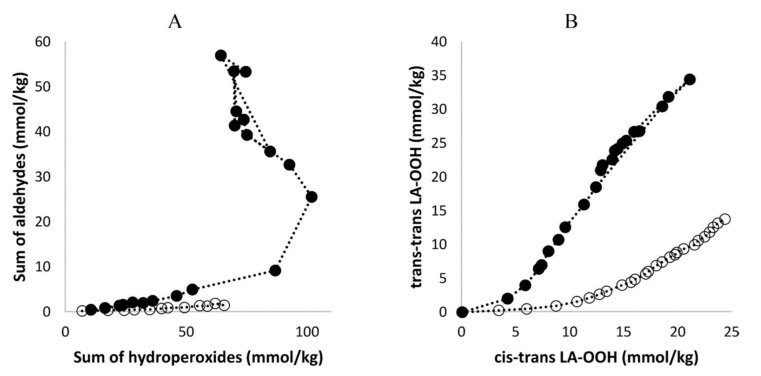
Map of oxidation trajectories at elevated (●) and ambient (○) temperature, stored for 53 and 203 days, respectively. (**A**) Plot of aldehydes *vs* hydroperoxides (PC2 visualized); (**B**) Plot of *trans*−*trans*-LA-OOH *vs. cis*−*trans*-LA-OOH (PC3 visualized). Reprinted with permission from [135]. Copyright 2018, American Chemical Society.

**Figure 38 molecules-27-02139-f038:**
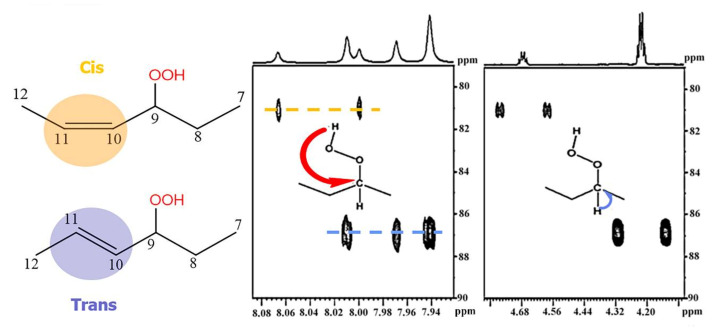
Selected region of the 800 MHz ^1^H–^13^C HMBC NMR spectrum of 45 mg of methyl oleate in CDCl_3_ subjected to heating at 70 °C for 135 h. Number of scans, 32; number of increments, 256; total experimental time, 5 h 53 min [137].

**Figure 39 molecules-27-02139-f039:**
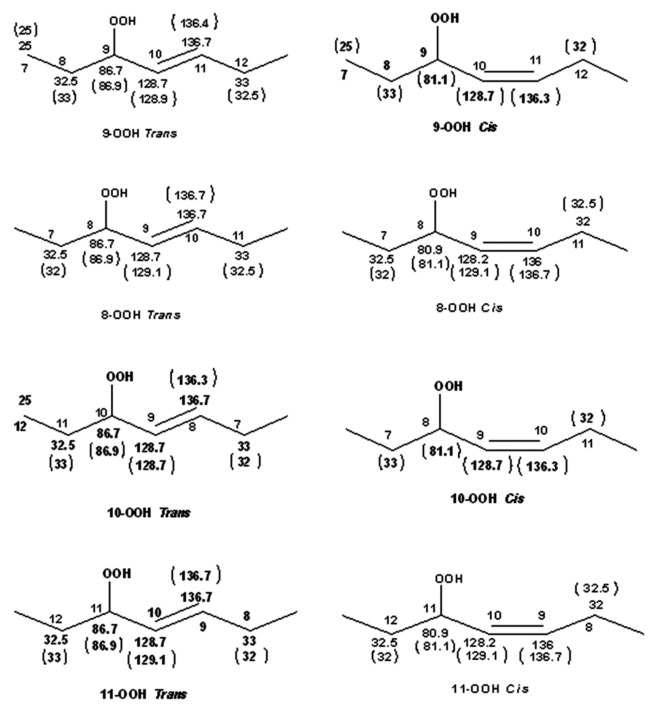
A comparison of the ^13^C-NMR chemical-shift data of geometric *cis*/*trans* hydroperoxides resulted from oxidation of methyl oleate with those obtained from the literature with the use of chromatographically isolated hydroperoxides (data in parentheses) [137].

**Figure 40 molecules-27-02139-f040:**
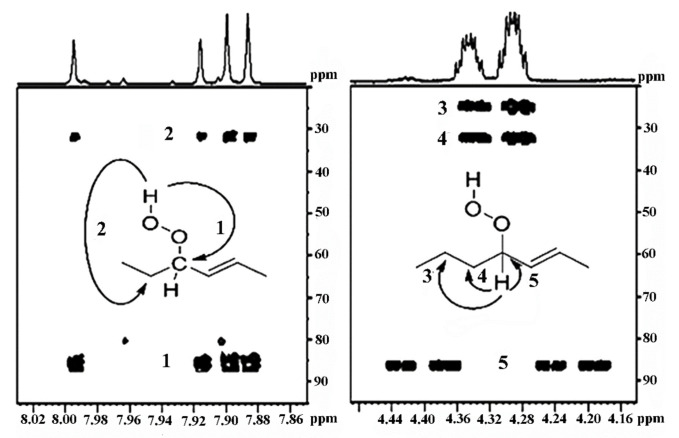
Selected region of 800 MHz ^1^H–^13^C HMBC NMR spectrum of 45 mg of methyl linoleate in CDCl_3_ subjected to heating at 70 °C for 8 h. Number of scans, 32; number of increments, 256; total experimental time, 11 h 46 min [137].

**Figure 41 molecules-27-02139-f041:**
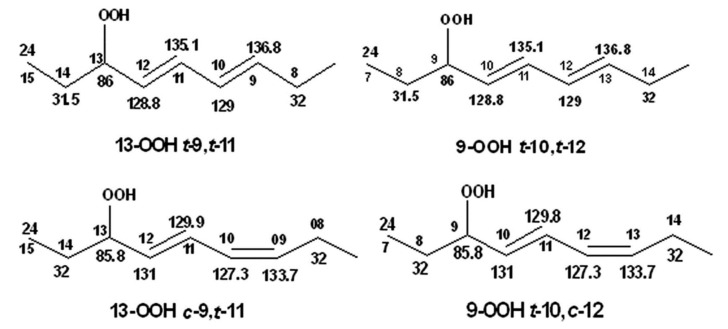
^13^C-NMR chemical shifts of the isomeric hydroperoxides resulting from oxidation of methyl linoleate heated at 70 °C for 8 h [137].

**Figure 42 molecules-27-02139-f042:**
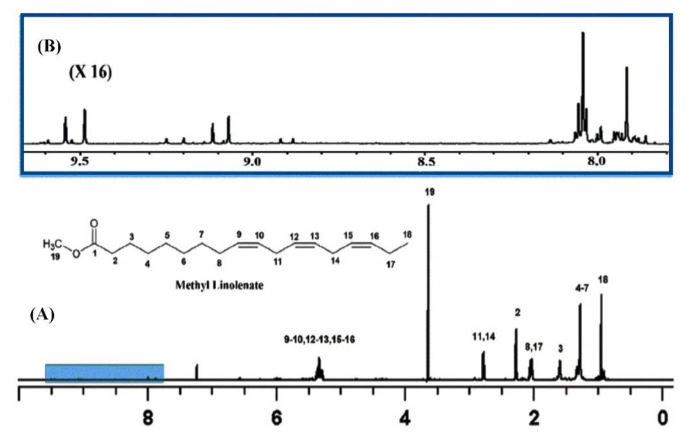
(**A**) 800 MHz ^1^H-NMR spectrum of 20 mg methyl linolenate in CDCl_3_ subjected to heating at 40 °C for 48 h. Number of scans = 64, acquisition time = 1.02 s, experimental time = 6.5 min, relaxation delay = 5 s, T = 298 K; (**B**) selected region of the C-O-O-H resonances of its primary oxidation products [138].

**Figure 43 molecules-27-02139-f043:**
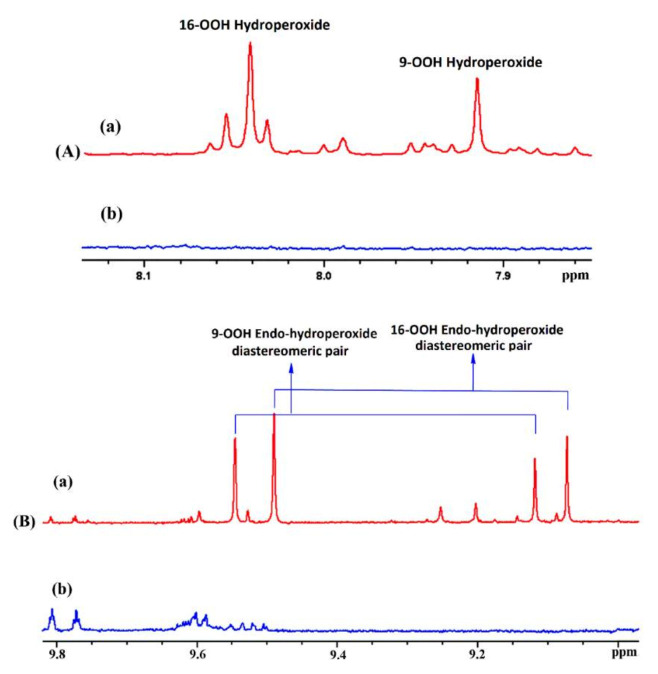
800 MHz ^1^H-NMR spectra of the solution of Figure 42 before the addition (**a**) and after the addition (**b**) of 2 microdrops of D_2_O. (**A**,**B**) are the selected regions of 7.86 to 8.14 and 8.96 to 9.82 ppm, respectively. For the method of assignment of the -OOH resonances, see text [138].

**Figure 44 molecules-27-02139-f044:**
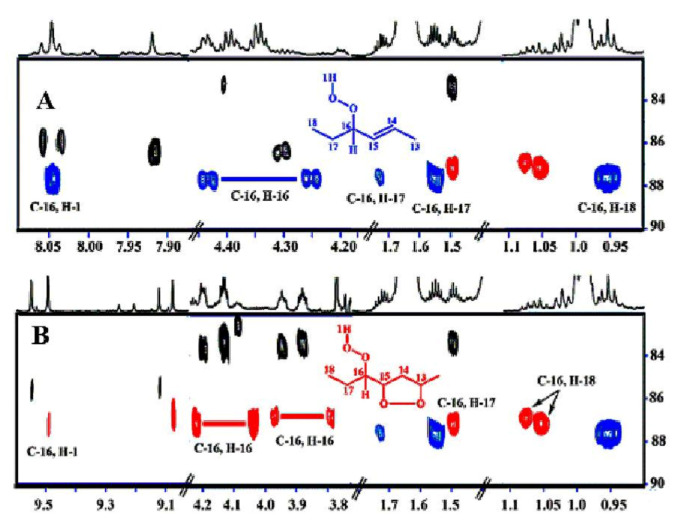
Selected regions of 800 MHz ^1^H-^13^C HMBC NMR spectrum of 20 mg methyl linolenate in CDCl_3_, subjected to heating at 40 °C for 48 h. Number of scans = 32, number of increments = 256, total experimental time = 12 h, T = 298 K. The critical cross-peak connectivities of C-16 with H-1(OOH), H-16 (^1^*J*(^13^C, ^1^H)), H-17, and H-18 of the 9-*cis*, 12-*cis*, 14-*trans*-16-OOH hydroperoxide (**A**), and two diastereomeric 9-*cis*, 11-*trans*-16-OOH *endo*-hydroperoxides (in red) (**B**), are illustrated [138].

**Figure 45 molecules-27-02139-f045:**
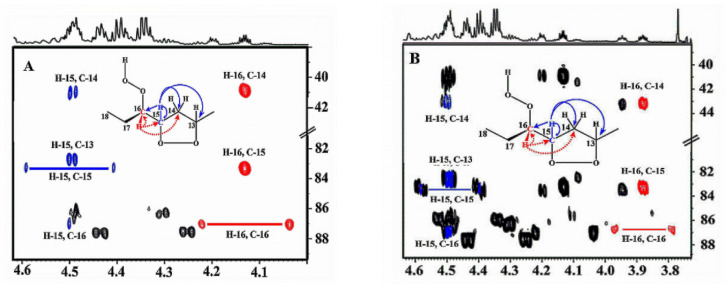
^1^H-^13^C HMBC correlations showing the assignments of the two diastereomeric 9-*cis*, 11-*trans*-16-OOH *endo*-hydroperoxides of methyl linolenate with OOH and CH-OOH resonances at 9.50 ppm and 4.3 ppm, respectively (**A**), and 9.08 ppm and 3.87 ppm, respectively (**B**). The critical connectivities of the H-15 with C-14, C-13, C-15 (^1^*J*(^13^C, ^1^H)), C-16, and connectivities of H-16 with C-14, C-15, and C-16 (^1^*J*(^13^C, ^1^H)) are illustrated [138].

**Figure 46 molecules-27-02139-f046:**
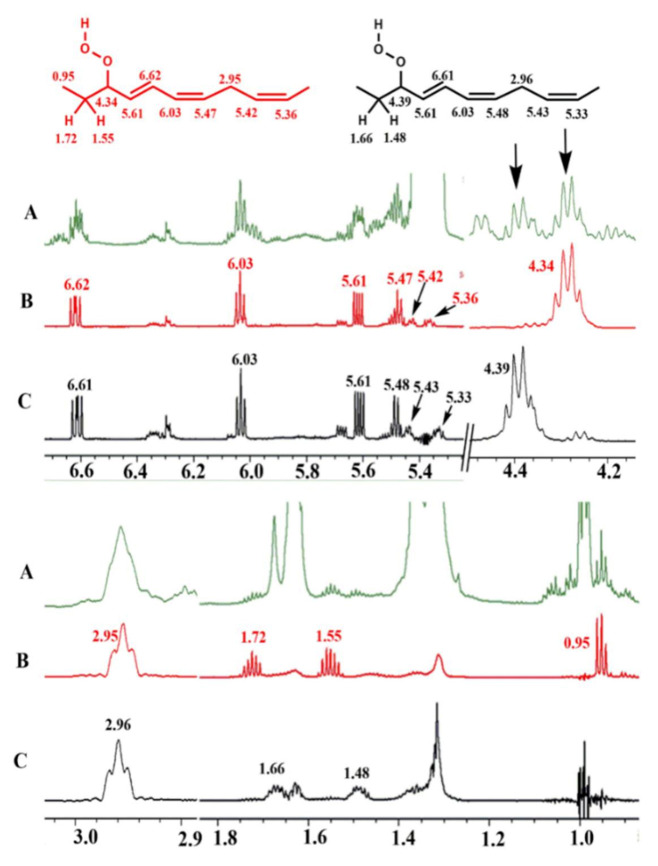
(**A**) 800 MHz 1D ^1^H-NMR spectrum of 20 mg methyl linolenate, subjected to heating at 40 °C for 48 h, in CDCl_3_ (acquisition time = 1.02 s, relaxation delay = 5 s, number of scans = 128, experimental time = 10 min), T = 298 K. (**B**,**C**) selective 1D-TOCSY spectra of the same solution using a mixing time of τ_m_ = 300 ms. The arrows denote the selected resonances that were excited at *δ* 4.34 (**B**) of the *cis*, *cis*, *trans*-16-OOH hydroperoxide and *δ* 4.39 (**C**) of the *trans*, *cis*, *cis*-9-OOH hydroperoxide. Number of scans = 256, experimental time = 20 min [138].

**Figure 47 molecules-27-02139-f047:**
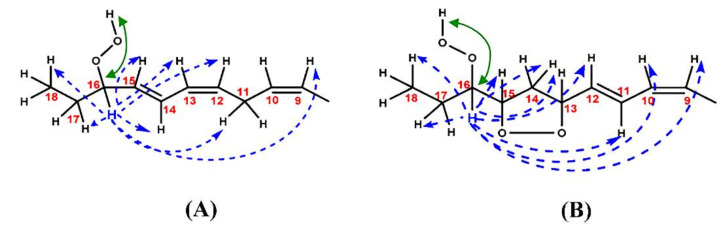
Schematic presentation of the combination of ^1^H-^13^C HMBC (
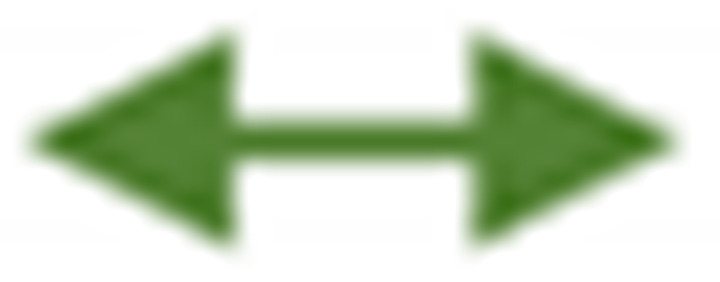
) and selective 1D TOCSY correlations (
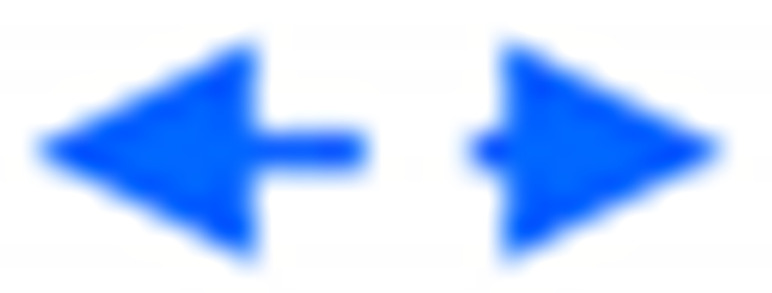
) which were observed in the case of the 9-*cis*, 12-*cis*, 14-*trans*-16-OOH hydroperoxide (**A**), and the 9-*cis*, 11-*trans*-16-OOH *endo*-hydroperoxide (**B**) of the methyl linolenate primary oxidation products.

**Figure 48 molecules-27-02139-f048:**
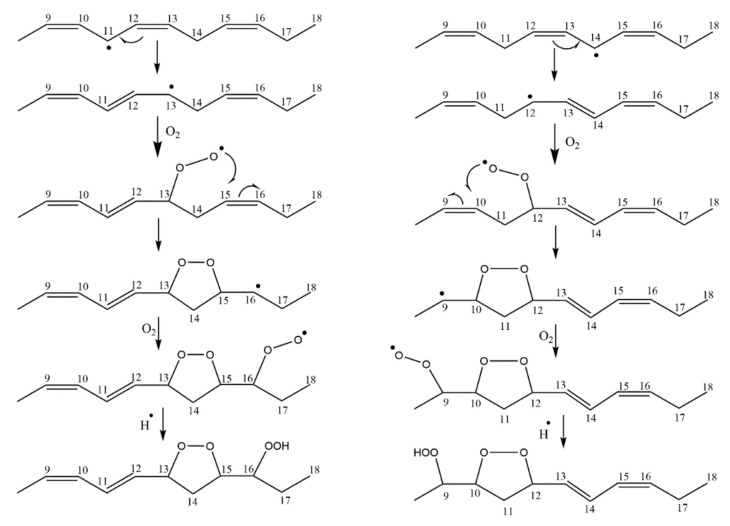
Proposed mechanism of the formation of two diastereomeric pairs of 9-*cis*, 11-*trans*-16-OOH, and 13-*trans*, 15-*cis*-9-OOH linolenate *endo*-hydroperoxides [138].

**Figure 49 molecules-27-02139-f049:**
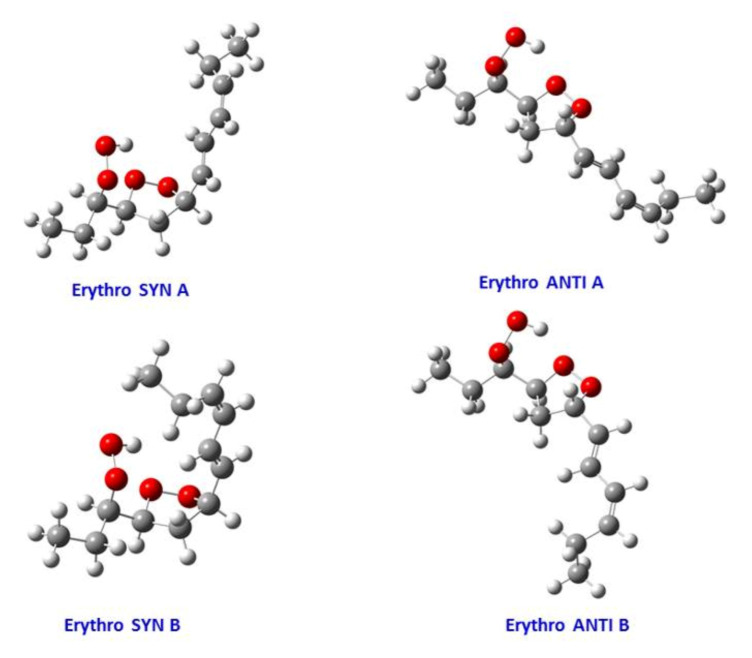
Structures of *erythro*- and *threo*-diastereomers of the *endo*-hydroperoxide with energy minimization at the DFT/B3LYL/6-31+G(d) level. The *anti* A and *anti* B conformers of (3S,5S)-3-((1*E*,3*Z*)-hexa-1,3-dien-1-yl)-5-((S)-1-hydroperoxypropyl)-1,2-dioxolanes are shown [141].

**Figure 50 molecules-27-02139-f050:**
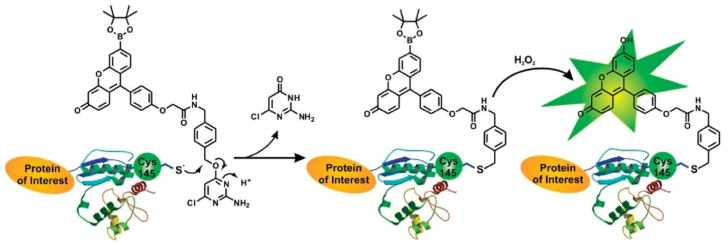
Design strategy for organelle-specific hydrogen peroxide reporters using the SNAP-tag methodology. Reprinted with permission from [164]. Copyright 2010, American Chemical Society.

**Figure 51 molecules-27-02139-f051:**
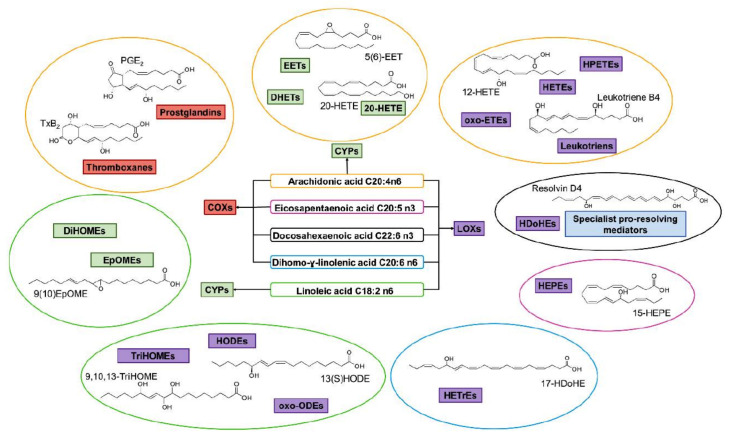
Oxylipin network and their fatty-acid precursors. Reprinted with permission from [179]. Copyright 2019, American Chemical Society.

**Figure 52 molecules-27-02139-f052:**
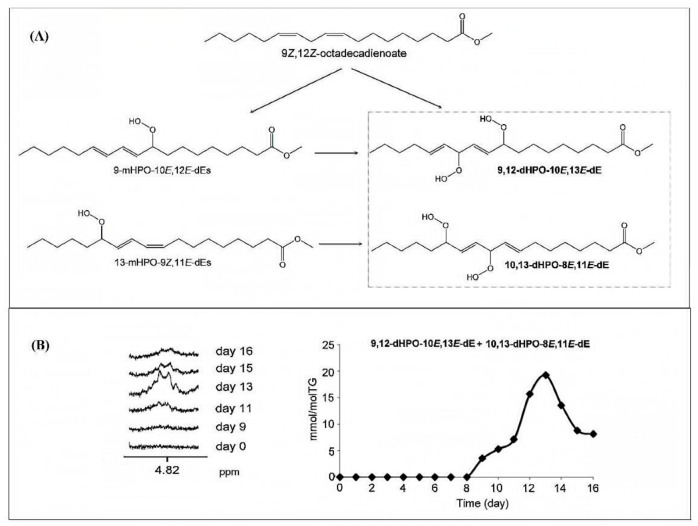
(**A**) Pathways of formation of dihydroperoxy-nonconjugated dienes (dHPO-nc-dEs). (**B**) Selected spectral region where changes occur during the accelerated storage process and their evolution with time, together with the graphical representation of the evolution of the concentration of 9,12-dHPO-10*E*,13*E*-dE + 10,13-dHPO-8*E*,11*E*-dE, expressed as mmol/mol TG, versus time [186].

**Table 1 molecules-27-02139-t001:** Dissociation energies of C-H, O-H and O-O bonds in various compounds [8].

Compounds	Bond	Δ*E* (kcal/mol)
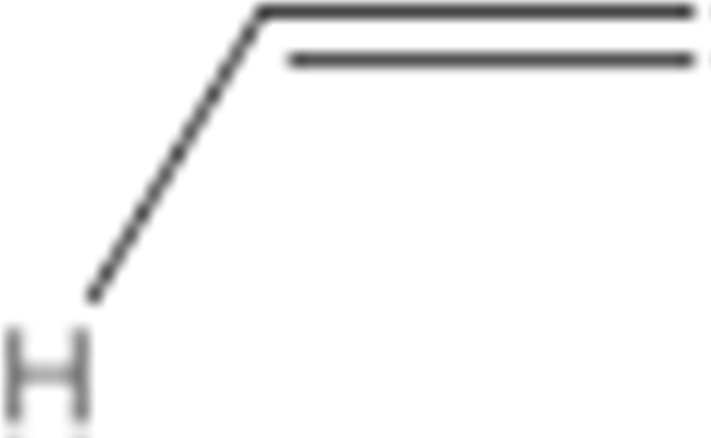	C-H	103
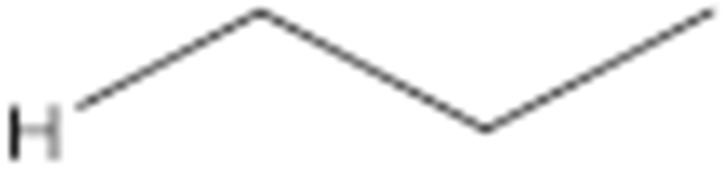	C-H	100
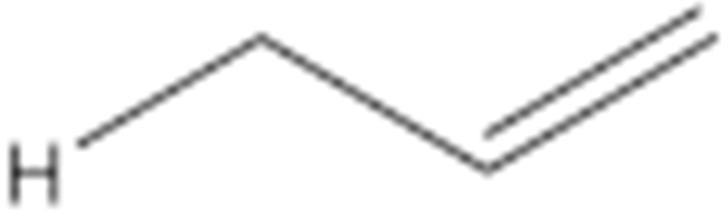	C-H	85
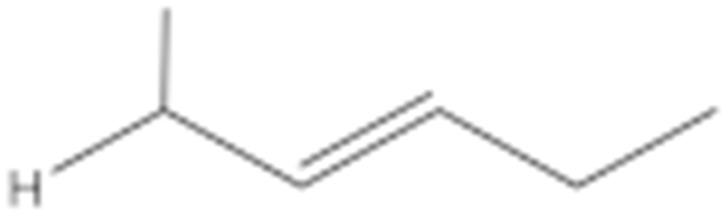	C-H	77
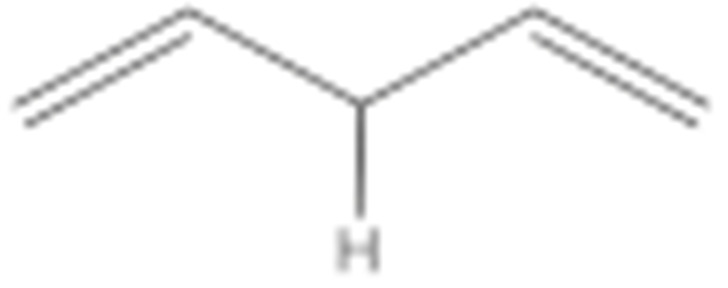	C-H	65
H–O-O-R	O-H	90
H–O-O-R	O-O	44

**Table 2 molecules-27-02139-t002:** ^13^C-NMR chemical shifts (ppm) of geometric dienol isomers from autoxidized methyl linoleate. Reprinted, with permission, from [99]. Copyright 1990, Elsevier.

Assignment	13-OH *cis*-9, *trans*-11	13-OH *trans*-9, *trans*-11		9-OH *trans*-10, *cis*-12	9-OH *trans*-10, *trans*-12
CH_3_			51.4		
C-1			174.2		
C-2			34.2		
C-3			25.0–25.4		
C-8	27.8	32.6		37.6	37.5
C-9	132.4	135.0		72.9	72.9
C-10	127.9	129.6		125.8	131.0
C-11	136.1	133.9		136.1	133.8
C-12	125.5	130.7		128.0	129.6
C-13	72.9	72.9		132.7	135.9
C-14	37.4	37.4		27.8	32.7
C-16			31.5–31.9		
-CH_2_-			29.0–29.5		
C-17			22.6		
C-18			14.0		

**Table 3 molecules-27-02139-t003:** ^1^H-NMR chemical shifts and spin multiplicities of oxidation products of edible lipids. Adopted with permission from [102]. Copyright 2014, Institute of Food Technologists.

Signal	Chemical Shift (ppm)	Multiplicity	Functional Group
Primary Oxidation Compounds			
−CH=CH−CH=CH−	6.58	dddd	(*Z*,*E*)-conjugated double
	6.00	ddtd	bonds associated with
	5.56	ddm	hydroperoxides (OOH)
	5.51	dtm	
−CH=CH−CH=CH−	6.45	ddd	*(Z*,*E*)-conjugated double
	5.94	dd	bonds associated with
	5.64	dd	hydroxides (OH)
	5.40	ddt	
−CH=CH−CH=CH−	6.27	ddm	(*E*,*E*)-conjugated double
	6.06	ddtd	bonds associated with
	5.76	dtm	hydroperoxides (OOH)
	5.47	ddm	
−CHOOH−CH = CH−	5.72	m	Double bond associated with
			hydroperoxides (OOH)
−OOH	8.3 to 8.9	—	Hydroperoxide group
**Secondary or Further Oxidation Compounds**			
Aldehydes			
−CHO	9.49	d	(*E*)-2-alkenals
−CHO	9.52	d	(*E*,*E*)-2,4-alkadienals
−CHO	9.55	d	4,5-epoxy-2-alkenals
−CHO	9.57	d	4-hydroxy-(*E*)-2-alkenals
−CHO	9.58	d	4-hydroperoxy-(*E*)-2-alkenals
−CHO	9.60	d	(*Z*,*E*)-2,4-alkadienals
−CHO	9.75	t	*n*-alkanals
−CHO	9.78	t	4-oxo-alkanals
−CHO	9.79	t	*n*-alkanals of low molecular
			weight (ethanal and propanal)
Alcohols			
−CHOH−CHOH−	3.43	m	9,10-dihydroxy-12- octadecenoate
			(leukotoxindiol)
−CHOH−	3.54–3.59	m	secondary alcohols
−CH_2_OH−	3.62	t	primary alcohols
Epoxides			
−CHOHC−	2.63	m	(*E*)-9,10-epoxystearate
−CHOHC−	2.88	m	(*Z*)-9,10-epoxystearate
−CHOHC−	2.90	m	9,10-epoxy-octadecanoate;
			9,10-epoxy-12-
			octadecenoate
			(leukotoxin); 12,13-epoxy-
			9-octadecenoate
			(isoleukotoxin)
−CHOHC−CHOHC−	2.90	m	9,10–12,13-diepoxy
			octadecanoate
−CHOHC−CH_2_−CHOHC−	3.10	m	9,10–12,13-diepoxy
			octadecanoate
Ketones and unidentified			
O=C*<*CH=CH−	6.08	dt	Double bond conjugated with a keto group
	6.82	m	
	7.50		Unidentified
	8.10		Unidentified

d, doublet; t, triplet; m, multiplet.

**Table 4 molecules-27-02139-t004:** Oxidation conditions during storage of sunflower oil samples: storage time (ST), air/oil contact surface (CS), air volume (AV), oil volume (OV), air/oil volume ratio (AOVR) together with the relative molar proportions of hydroperoxides (HY) and aldehydes (AL) [39].

Sample	ST (Months)	CS (cm^2^)	AV (cm^3^)	OV (cm^3^)	AOVR	HY	AL
S6	4	15.2	10.9	898.6	0.012	0.075	0.000
S7	63	32.2	45.3	729.6	0.062	0.270	0.000
S17 ^a^	90	38.5	697.0	192.4	3.622	0.782	0.170
S18	106	12.6	12.6	62.8	0.200	1.391	0.226
S23	106	12.6	37.7	37.7	1.000	2.091	0.751

^a^ α-Tocopherol (αT) was added at concentration ranges from 600 to 720 mg/kg of oil.

**Table 5 molecules-27-02139-t005:** Signal assignments of the peroxides and aldehydes annotated in Figure 35. Reprinted with permission from [135]. Copyright 2018, American Chemical Society.

Hydroperoxides					Aldehydes			
Peak	Chem. Shift (ppm)	FA	Annotation	Partial Chemical Structure	Chem. Shift (ppm)	FA	Annotation	Partial Chemical Structure
A	δ 11.25-11.12	αLn	Cyclized		K	δ 9,79-9,755	Small nalkanals+ unasssigned	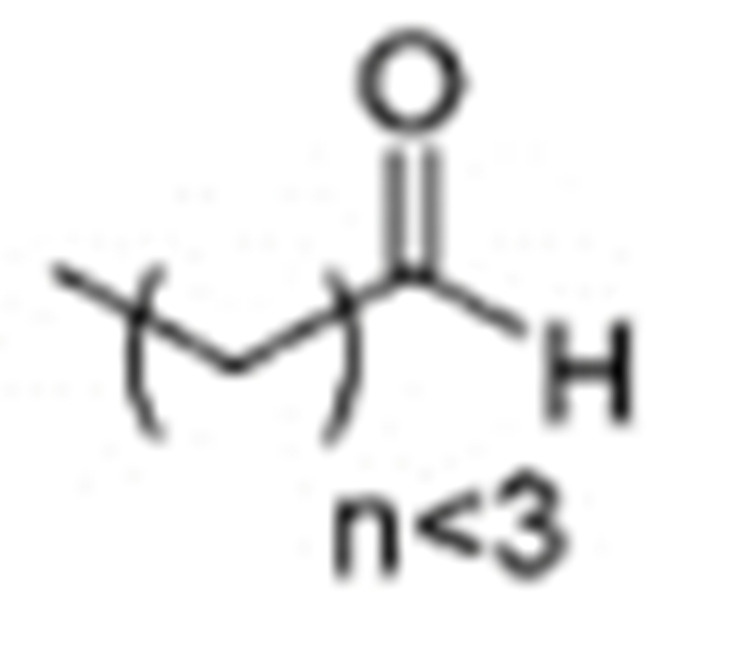
B	δ 11.10-11.05	αLn	Cyclized	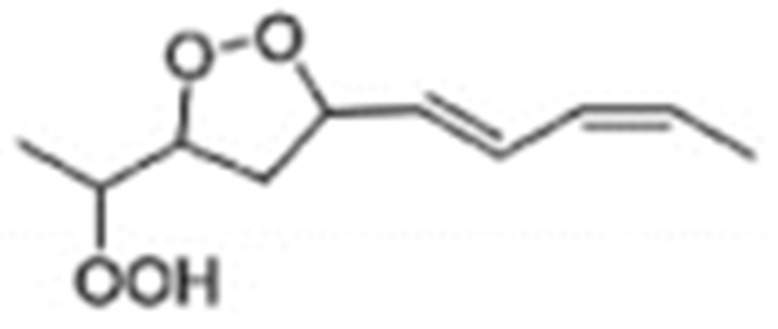	L	δ 9,75-9,73	n-alkanals (n > 5)	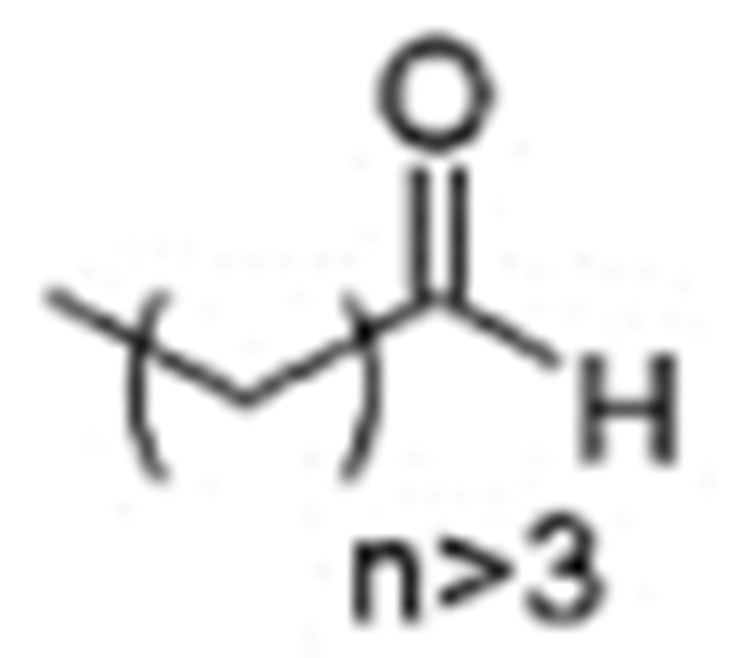
C	δ 11.05-11.00	αLn	Cyclized		M	δ 9,71-9,69	unasssigned	
D	δ 10.99-10.96	αLn	12-OOH, 13-OOH	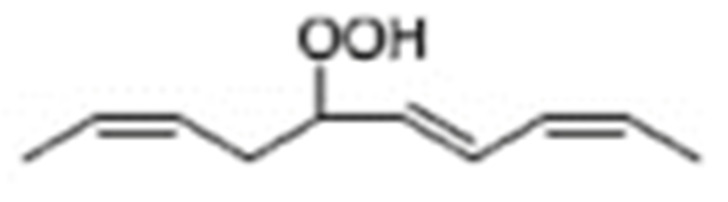	N	δ 9,65-9,64	unasssigned	
E	δ 10.95-10.91	αLn	9-OOH, 16-OOH	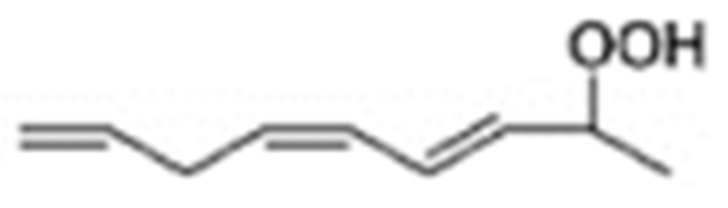	O	δ 9,635-9,62	unasssigned	
F	δ 10.91-10.87	LA	9-*trans*,*cis*-OOH 13-*cis*,*trans*-OOH	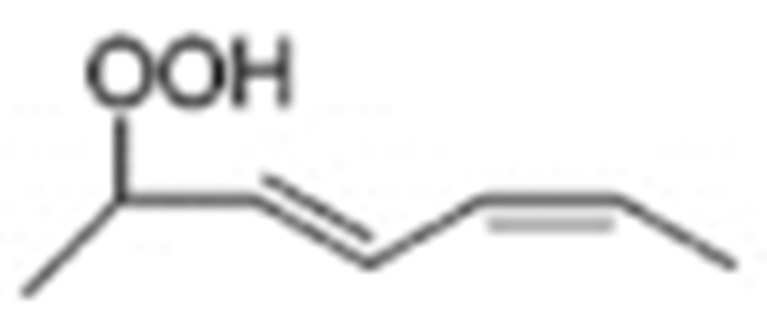	P	δ 9,62-9,61	unasssigned	
G	δ 10.87-10.82	LA	9-*trans*,*trans*-OOH 13-*trans*,*trans*-OOH	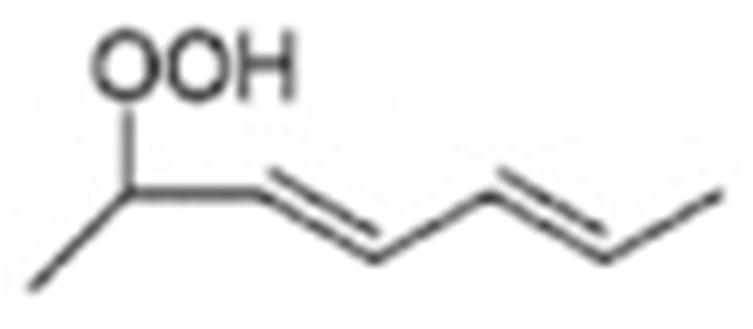	Q	δ 9,61-9,58	unasssigned	
H	δ 10.82-10.80		unassigned		R	δ 9,58-9,54	4-hydroperoxy-*trans*-2-alkenals	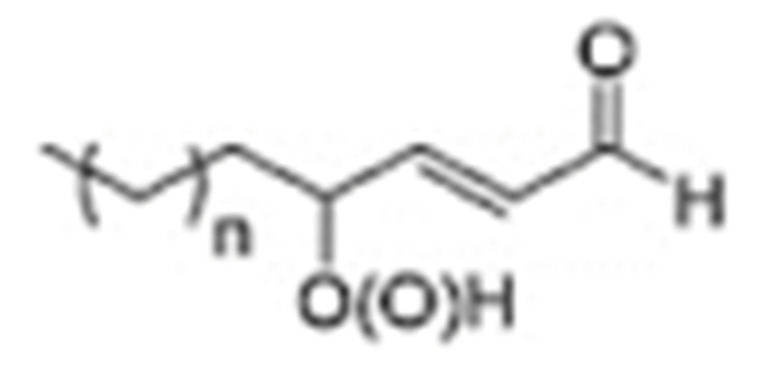
I	δ 10.80-10.77	O	9-*trans*-OOH, 13-*trans*-OOH	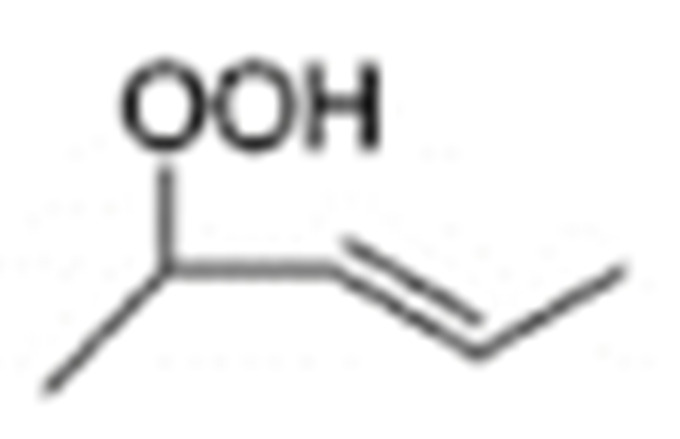	S	δ 9,53-9,50	*Trans*,*trans*-2,4-alkadienals, 4,5-epoxy-*trans*-2-alkenals	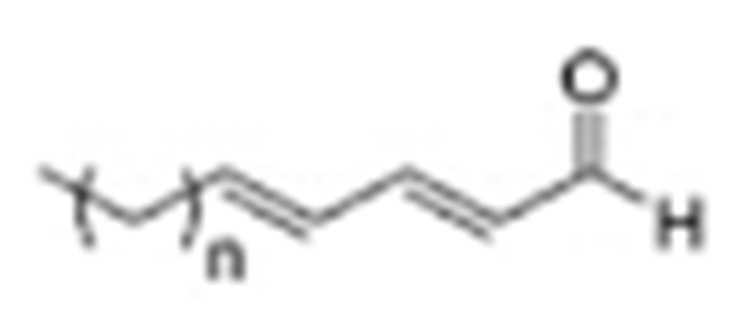
J	δ 10.74-10.71	O	9-*cis*-OOH, 13-*cis*-OOH	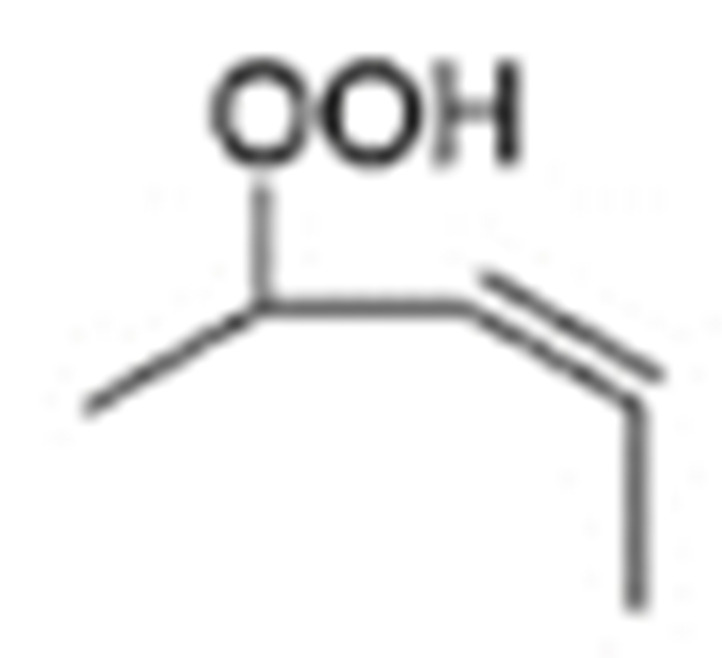	T	δ 9,50-9,475	*Trans*-2-alkenals	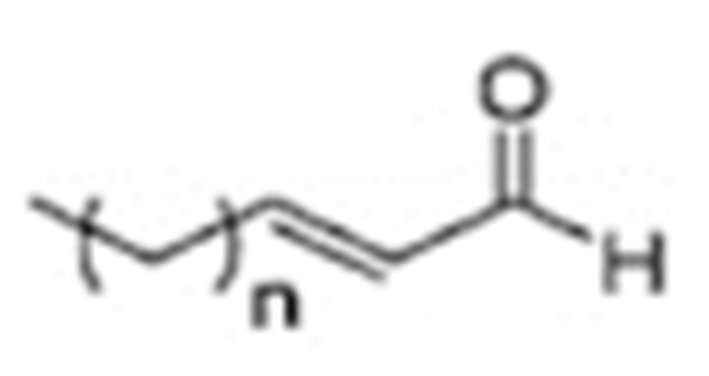

**Table 6 molecules-27-02139-t006:** Comparison of the quantification data ^a^ of hydroperoxides produced during oxidation of methyl oleate heated at 70 °C for 135 h [137] and with reported literature [10,98].

Hydroperoxide Chemical Shift (ppm)	% with Respect to Total Methyl Oleate	% of Total Hydroperoxides	Hydroperoxide Geometric Isomer	% of Total Hydroperoxide [10,98]		Assignment as per Literature
				75 °C	50 °C	
7.94 ^b^	3.19	18% ^c^	*Trans*	19.0%	17.8%	*Trans* 8-OOH
7.94 ^b^		20% ^c^	*Trans*	22.5%	22.5%	*Trans* 9-OOH
8.01	1.74	21%	*Trans*	22%	21.3%	*Trans* 10-OOH
7.97	1.72	20%	*Trans*	19.5%	17.4%	*Trans* 11-OOH
8.00	0.95	11%	*Cis*	6.1%	8.3%	*Cis* 8-OOH
8.07	0.85	10%	*Cis*	5.4%	8.3%	*Cis* 11-OOH
				2.7%	2.2%	*Cis* 9-OOH
				2.9%	2.2%	*Cis* 10-OOH

^a^ With respect to the total integral of the terminal CH_3_ group at 0.86 ppm. ^b^ Composite signal presumably due to *trans* 8-OOH and *trans* 9-OOH. ^c^ The tentative assignment was based on the increased % amount of *trans* 9-OOH according to the literature [10,98].

**Table 7 molecules-27-02139-t007:** Comparison of NMR quantification data ^a^ of hydroperoxides produced during oxidation of methyl linoleate heated at 70 °C for 8 h [137] and with the reported literature for chromatographically isolated hydroxyperoxides [4,6].

Hydroperoxide ^1^H- NMR Chemical Shifts (ppm)	% with Respect to Total Methyl Linoleate	% Integration Data of Total Hydroperoxides	Geometric Isomerism of the Adjacent Group to the C-O-O-H Group	% Integration Data from [10,99] ^b^	Assignment as per Literature [10,99]
7.92	2.45	34%	*Trans*	30.4%	9-OOH *trans-*10, *trans-*12
7.94	2.48	35%	*Trans*	33.5%	13-OOH *trans-*9, *trans-*11
7.95	0.97	14%	*Trans*	17.5%	9-OOH *trans-*10, *cis-*12
8.04	1.08	15%	*Trans*	18.6%	13-OOH *Cis-*9, *trans-*11

^a^ With respect to the total integral of the terminal CH_3_ group at 0.97 ppm [137]. ^b^ Oxidation of methyl linoleate at 65 °C [10,99].

**Table 8 molecules-27-02139-t008:** ^1^H-NMR quantification data of hydroperoxides and *endo*-hydroperoxides produced during the oxidation of methyl linolenate [138] ^a^.

Hydroperoxide *δ* (^1^H), ppm	Integration Data (%)	Integration Data (%) of Total Hydroperoxides	Assignment
7.92	5.78%	29% (33.0%) ^b^	*10-Trans*, *12-cis*, *15-cis*-9-OOH
8.04	2.89%	14% (10.8%) ^b^	*9-Cis*, *13-trans*, *15-cis*-12-OOH
8.05	8.23%	41% (43.9%) ^b^	*10-Cis*, *13-cis*, *15-trans*-16-OOH
8.06	3.08%	15% (12.3%) ^b^	*9-Cis*, *11-trans*, *15-cis*-13-OOH
9.08	1.80%		9-*Cis*, 11-*trans*, *syn erythro*, 16-OOH *endo*-hydroperoxide
9.12	1.36%		13-*T**rans*, 15-*cis*, *syn erythro*, 9-OOH *endo*-hydroperoxide
9.50	2.40%		9*-Cis*, 11-*trans*, *syn threo*, 16-OOH *endo*-hydroperoxide
9.55	1.88% ^c^		13-*Trans*, 15-*cis*, *syn threo*, 9-OOH *endo*-hydroperoxide^c^

^a^ Heated at 40 °C for 48 h. ^b^ % Integration data from the literature at 40 °C [10]. ^c^ The concentration of the *threo endo*-hydroperoxide was estimated to be 1.68 mM in the NMR tube, which can be compared with the detection limit of 3 μM of the 800 MHz NMR instrument.

**Table 9 molecules-27-02139-t009:** Comparison of experimental ^1^H- and ^13^C-NMR chemical shifts of the 9-*cis*, 11*-trans*-16-OOH diastereomeric *endo*-hydroperoxides of methyl linolenate [138] with the reported literature data with the use of chromatographically isolated *endo*-hydroperoxides [139,140].

	9-*Cis*, 11-*Trans-*16-OOH *Endo*-Peroxy *(Threo)*			9-*Cis*, 11-*Trans-*16-OOH *Endo-*Peroxy *(Erythro)*		9-*cis*, 11-*Trans-*16-OOH *Endo-*Peroxy *(Threo)*	
Proton No.	*δ* (^1^H), ppm	*δ* (^1^H), ppm ^a^	*δ* (^1^H), ppm ^b^	*δ* (^1^H), ppm	*δ* (^1^H), ppm ^a^	*δ* (^13^C), ppm	*δ* (^13^C), ppm ^b^
18	1.05	1.04	1.03	1.07	1.05	10.0	10.2
17	1.49	---	1.89–1.20	1.66	---	22.2	22.8
16	4.13	4.14	4.15	3.87	3.86	87.2	87.4
15	4.49	4.49	4.49	4.49	4.48	83.3	83.5
14(a)	2.84	2.86	2.84	2.88	2.88	40.7	41.3
14(b)	2.48	2.46	2.47	2.23	2.23		
13	4.81	4.80	4.80	4.79	4.78	82.8	82.9
12	5.63	5.62	5.62	5.58	5.57	126.2	126.3
11	6.65	6.67	6.67	6.67	6.64	131.6	131.8
10	6.00	6.00	6.01	6.01	5.99	127.3	127.3
9	5.55	5.54	5.55	5.54	5.54	135.4	135.2
OOH	9.50	9.52	9.38	9.08	9.04		

^a^ Ref [139]; ^b^ Ref [140].

**Table 10 molecules-27-02139-t010:** *δ*(C-O-O-^1^H), *δ*(^13^C-O-O-H), *δ*(^13^C) of C_3_ and C_5_, and ^3^*J*(^13^C-O-O-^1^H) coupling constants of low-energy conformers of Figure 49. Energy minimization was performed at the DFT/B3LYL/6-31+G(d) level and calculation of NMR parameters at the GIAO/B3LYP/6-311+G(2d,p) CPCM (CHCl_3_) level [141].

Stereoisomers	*δ*(O-O-^1^H) (ppm)	*δ*(C^1^H-O-O-H) (ppm)	*δ*(^1^H, H_5_) (ppm)	*δ*(^1^H, H_3_) (ppm)	*δ*(^13^C-O-O-^1^H) (ppm)	*δ*(^13^C, C_3_) (ppm)	*δ*(^13^C, C_5_) (ppm)	^3^*J*(^13^C-O-O-^1^H) (Hz)
*Erythro anti* A	9.47	4.32	4.57	5.09	90.86	88.76	94.38	−0.11
*Erythro anti* B	9.35	4.32	4.59	5.06	90.62	87.32	93.79	−0.18
*Erythro syn* A	8.78	4.32	4.43	5.16	89.44	86.15	92.94	−0.42
*Erythro syn* B	9.14	4.40	4.36	5.06	90.67	85.60	92.63	−0.21
*Threo anti* A	10.14	3.91	4.18	5.14	93.79	86.10	91.99	−0.27
*Threo anti* B	10.09	3.89	4.13	5.13	94.12	86.66	91.30	−0.26
*Threo syn* A	10.12	3.96	4.29	4.98	94.34	87.11	94.26	−0.29
*Threo syn* B	10.10	3.97	4.32	4.76	94.18	90.60	94.50	−0.26

**Table 11 molecules-27-02139-t011:** Results of the ^13^C chemical-shift calculations of several model compounds of the isolated photoinduced squalene cyclic *endo* hydroperoxide [156].

Model Structures	Notation	RMS	Max Absolute	Mean Absolute
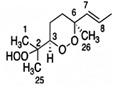	*Cis* 1A	0.96	1.80	0.75
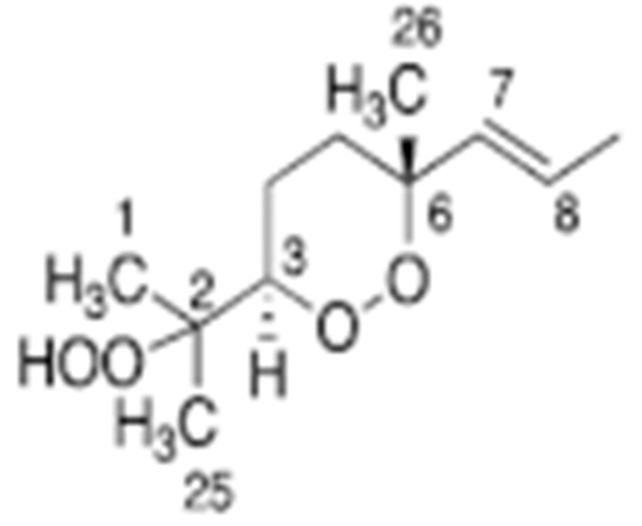	*Trans* 1A	1.85	3.40	1.58
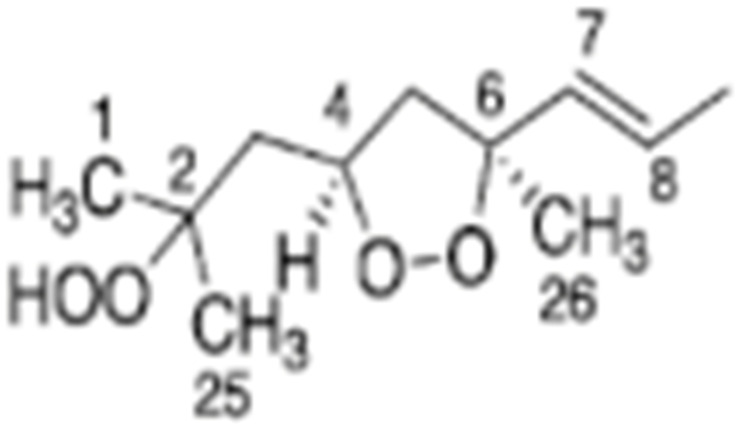	*Cis* 2A	9.55	21.00	6.90
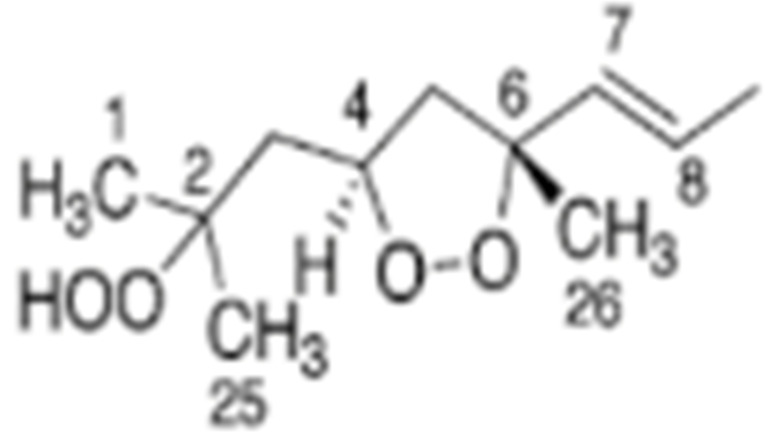	*Trans* 2A	10.05	22.00	7.51
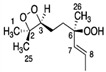	R 1B	4.23	7.20	3.65
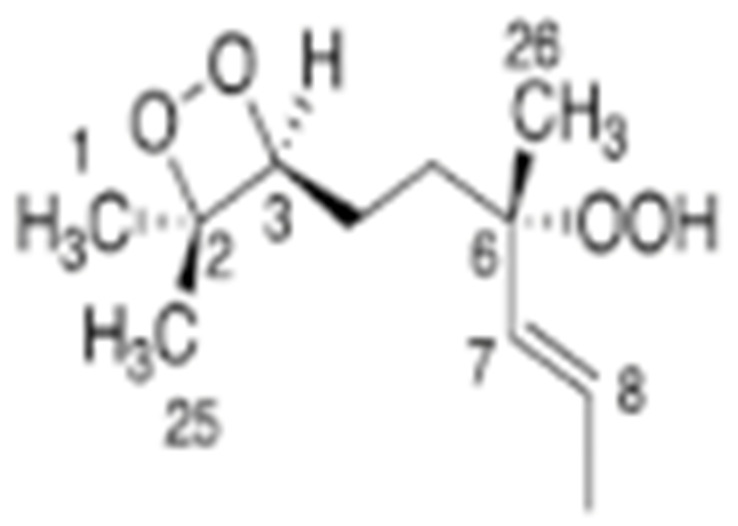	S 1B	4.31	5.90	3.61
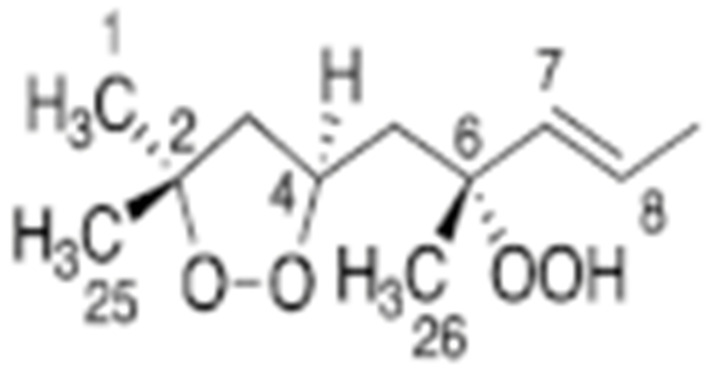	R 2B	10.72	31.60	6.62
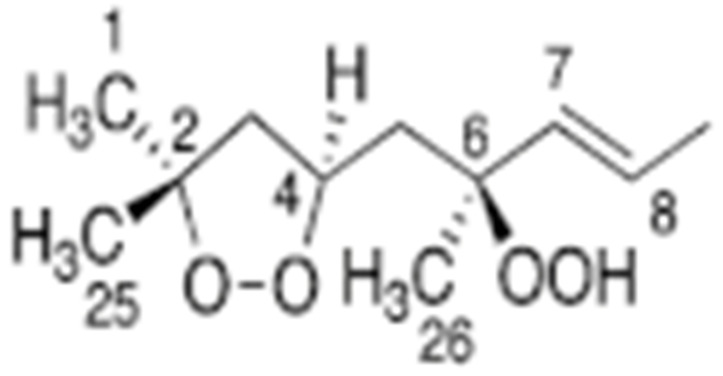	S 2B	10.35	30.20	6.32

## Data Availability

Not applicable.

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
