# Peer review of "Analytical and Structural Tools of Lipid Hydroperoxides: Present State and Future Perspectives"

_molecules, 2022, doi:10.3390/molecules27072139_

Round 1

Reviewer 1 Report

I recommend the publication of this review as it is. It is a good quality paper extensively developed, and well documented, especially from NMR point of view.

Very well chosen References and good topic cover overall.

Main contributions of the review: A clear and extended view about lipid hydroperoxides. In my opinion, are two components of this review: the actual status, which is extensively treated from analytical point of view using an impressive number of complex methods from chemical physics to biology : volumetry, UV-vis spectroscopy, IR, Raman/surface-enhanced Raman, fluorescence and chemiluminescence, chromatographic methods, mass spectrometry techniques, NMR and chromatographic methods including NMR spectroscopy in mixture analysis, structural investigations based on quantum chemical calculations of these  parameters, tests in living cells metabolomics, and, the second part (in spite of the fact that it is restraint compared with the rest) perspectives of the future in this field.

Insignificant coherence errors: analytical methods must be named in the same way: UV-Vis, IR, NMR, MS etc. and/or detailed and explained once at the beginning in parenthesis (like NIR at line 199).

The NMR section is very well developed (and excellent illustrated) comparing with the rest of methods.

More accurate assertions needed, avoiding approximately evaluations, like in “Lipid perroxidation is perhaps the most damaging effect since cell membranes are composed of poly-unsaturated fatty acids which are primary targets of reactive oxygen species”.

At Fig. 7 for Absorbance on the axis can be indicate the arbitrary units (a.u.) if is possible (or similar missing as parenthesis on Fig.10).

All these errors are insignificant and have not altered the good quality of the review.

The final Chapter is very strong and conclusive. The part containing References is not exhaustive but this part contain the most representatives works in the field and in displayed in great number.

Author Response

Reviewer #1

(1) “I recommend the publication of this review as it is. It is a good quality paper extensively developed, and well documented, especially from NMR point of view.

We are very pleased with this comment of the Reviewer.

(2) “Very well chosen References and good topic cover overall.”

We are very pleased with this comment of the Reviewer.

(3) “Main contributions of the review: A clear and extended view about lipid hydroperoxides. In my opinion, are two components of this review: the actual status, which is extensively treated from analytical point of view using an impressive number of complex methods from chemical physics to biology : volumetry, UV-vis spectroscopy, IR, Raman/surface-enhanced Raman, fluorescence and chemiluminescence, chromatographic methods, mass spectrometry techniques, NMR and chromatographic methods including NMR spectroscopy in mixture analysis, structural investigations based on quantum chemical calculations of these  parameters, tests in living cells metabolomics, and, the second part (in spite of the fact that it is restraint compared with the rest) perspectives of the future in this field.

We are very pleased with the above favorable comments of the Reviewer.

 (4) “Insignificant coherence errors: analytical methods must be named in the same way: UV-Vis, IR, NMR, MS etc. and/or detailed and explained once at the beginning in parenthesis (like NIR at line 199).”

We agree with the Reviewer, therefore, the appropriate shorthand rotations are introduced in the last paragraph of the introduction and in the titles of the sub-sections. Furthermore, corrections have also been made in the text.

(5) “The NMR section is very well developed (and excellent illustrated) comparing with the rest of methods.”

We are very pleased with this comment of the Reviewer.

(6) “More accurate assertions needed, avoiding approximately evaluations, like in “Lipid perroxidation is perhaps the most damaging effect since cell membranes are composed of poly-unsaturated fatty acids which are primary targets of reactive oxygen species”.

 We agree with the Reviewer, therefore the appropriate corrections have been made in the revised version of our manuscript.

(7) “At Fig. 7 for Absorbance on the axis can be indicate the arbitrary units (a.u.) if is possible (or similar missing as parenthesis on Fig.10).”

 We agree with the Reviewer, therefore, the appropriate corrections have been made on Figs 7 and 10 of the revised version of our manuscript.

(8) “All these errors are insignificant and have not altered the good quality of the review.”

We are very pleased with this comment of the Reviewer.

(9) “The final Chapter is very strong and conclusive. The part containing References is not exhaustive but this part contain the most representatives works in the field and in displayed in great number.”

We are very pleased with this comment of the Reviewer.

Reviewer 2 Report

Review of the manuscript titled: Analytical and Structural Tools of Lipid Hydroperoxides: Present State and Future Perspectives.

General comments
The review is well written and well organized. The text is easy to read and the contents on the analysis of lipid hydroperoxides with different methods and instruments are very complete. For these reasons I suggest to accept the paper in this form after the following minor revisions:

Table 1: I suggest spacing the third and fourth formulas because it looks like the hydrogen atom makes two bonds!
Figure 1 and throughout the text: Authors must specify what the R and R' residues are otherwise write the complete formula for oleate (linoleate and linolenate).

Author Response

Reviewer #2

(1) “General comments The review is well written and well organized. The text is easy to read and the contents on the analysis of lipid hydroperoxides with different methods and instruments are very complete. For these reasons I suggest to accept the paper in this form after the following minor revisions:”

We are very pleased with the above comment of the Reviewer.

(2) “Table 1: I suggest spacing the third and fourth formulas because it looks like the hydrogen atom makes two bonds!”

 We agree with the Reviewer, therefore, the appropriate correction has been made on Table 1 in the revised version of our manuscript.

(3) “Figure 1 and throughout the text: Authors must specify what the R and R' residues are otherwise write the complete formula for oleate (linoleate and linolenate).”

 We agree with the Reviewer, therefore, the definition of R and R’ is given in the figure caption.
